# Molecular basis of the PIP$_2$-dependent regulation of Ca$_V$2.2 channel and its modulation by Ca$_V$ β subunits

**Cheon-Gyu Park, Wookyung Yu, Byung-Chang Suh\***

Department of Brain Sciences, Daegu Gyeongbuk Institute of Science and Technology (DGIST), Daegu, Republic of Korea

**Abstract** High-voltage-activated Ca$^{2+}$ (Ca$_V$) channels that adjust Ca$^{2+}$ influx upon membrane depolarization are differentially regulated by phosphatidylinositol 4,5-bisphosphate (PIP$_2$) in an auxiliary Ca$_V$ β subunit-dependent manner. However, the molecular mechanism by which the β subunits control the PIP$_2$ sensitivity of Ca$_V$ channels remains unclear. By engineering various α1B and β constructs in tsA-201 cells, we reported that at least two PIP$_2$-binding sites, including the polybasic residues at the C-terminal end of I–II loop and the binding pocket in S4$_{II}$ domain, exist in the Ca$_V$2.2 channels. Moreover, they were distinctly engaged in the regulation of channel gating depending on the coupled Ca$_V$ β2 subunits. The membrane-anchored β subunit abolished the PIP$_2$ interaction of the phospholipid-binding site in the I–II loop, leading to lower PIP$_2$ sensitivity of Ca$_V$2.2 channels. By contrast, PIP$_2$ interacted with the basic residues in the S4$_{II}$ domain of Ca$_V$2.2 channels regardless of β2 isotype. Our data demonstrated that the anchoring properties of Ca$_V$ β2 subunits to the plasma membrane determine the biophysical states of Ca$_V$2.2 channels by regulating PIP$_2$ coupling to the nonspecific phospholipid-binding site in the I–II loop.

## Editor's evaluation

This manuscript describes experiments using heterologous expression to achieve molecular dissection of the effects of PIP2 and CaVβ2 auxiliary subunits on CaV2.1 (P/Q-type) calcium channels. The experiments also probe interplay between lipid effects and other modulatory pathways. Understanding the functional regulation of this channel is important because CaV2.1 channels play significant roles in neuronal plasticity.

**\*For correspondence:**
bcsuh@dgist.ac.kr

**Competing interest:** The authors declare that no competing interests exist.

## Introduction

Voltage-gated Ca$^{2+}$ (Ca$_V$) channels that mediate Ca$^{2+}$ influx upon membrane depolarization contribute to various physiological events, including synaptic transmission, hormone secretion, excitation–contraction coupling, and gene transcription (*Berridge et al., 2000*; *Catterall, 2011*; *Clapham, 2007*; *Li et al., 2016*). Ca$_V$ channels can be divided into high-voltage-activated (HVA) and low-voltage-activated (LVA) channels based on their activation voltage threshold. The HVA Ca$^{2+}$ channels, which consist of the Ca$_V$1 and Ca$_V$2 families, are multiprotein complexes with a pore-forming α1 subunit and auxiliary α2δ and β subunits. Diverse cellular factors regulate Ca$_V$ channel activity (*Felix, 2005*; *Huang and Zamponi, 2017*).

Among the various intracellular regulatory signals of Ca$_V$ channels, we focus on the membrane phospholipid phosphatidylinositol 4,5-bisphosphate (PIP$_2$). Previous studies have shown that PIP$_2$ activates several types of HVA Ca$_V$ channels in recombinant systems and native tissue cells (*Hille et al., 2015*; *Rodríguez-Menchaca et al., 2012*; *Suh and Hille, 2008*; *Wu et al., 2002*; *Xie et al., 2016*).

Dr-VSP, a voltage-sensing lipid phosphatase from zebrafish, can be used to examine the effects of $PIP_2$ on $Ca_V$ channels without the involvement of other downstream second messengers generated from $G_q$-coupled receptors (*Murata et al., 2005*; *Okamura et al., 2009*; *Suh et al., 2010*). In vitro experiments using Dr-VSP have shown that most HVA $Ca^{2+}$ channels are suppressed by membrane $PIP_2$ depletion without influencing LVA $Ca^{2+}$ channels (*Jeong et al., 2016*; *Suh et al., 2010*). $PIP_2$ induces two distinct and opposing regulatory effects on $Ca_V2.1$ channels (*Wu et al., 2002*). Thus, the $Ca_V2.1$ channel protein was suggested to contain two distinct $PIP_2$-interaction sites with different binding affinity (*Wu et al., 2002*). A more recent study showed that four arginine residues within the C-terminal end of the I–II loop of L-type $Ca_V1.2$ channels are involved in nonspecific phospholipid interactions; therefore, the substitution of these basic residues for alanine decreases current inhibition via $PIP_2$ breakdown and increases the open probability of $Ca_V1.2$ channels (*Kaur et al., 2015*). The precise $PIP_2$-binding sites have not been fully determined in $Ca_V$ channels yet.

Among the auxiliary subunits, $Ca_V$ β subunits directly bind to an α-interacting domain (AID) within the N-terminal region of the I–II loop. They play key roles in regulating membrane trafficking and fine-tuning the gating of $Ca_V$ channels (*Buraei and Yang, 2010*; *Buraei and Yang, 2013*). A single β subunit can be divided into five distinct regions: conserved src homology-3 (SH3) and guanylate kinase (GK) domains, a flexible HOOK region connecting the two domains, and variable N- and C-terminus. The GK domain contains an α-binding pocket (ABP), which is a site for interaction with the AID of the I–II loop (*Buraei and Yang, 2010*; *Buraei and Yang, 2013*; *Chen et al., 2004*; *Opatowsky et al., 2004*; *Van Petegem et al., 2004*). Additionally, the HOOK region, a flexible linker composed of around 70 amino acids, is important in determining the inactivation kinetics, current density, and $PIP_2$ regulation of $Ca_V2.2$ channels via electrostatic interaction with the plasma membrane (PM) (*Miranda-Laferte et al., 2012*; *Park et al., 2017*; *Park and Suh, 2017*; *Richards et al., 2007*). Several studies have shown that subcellular localization of the β subunits is primarily involved in the modulation of $Ca_V$ channel gating, including inactivation kinetics, current density, and $PIP_2$ sensitivity (*Keum et al., 2014*; *Kim et al., 2015a, Kim et al., 2016*; *Suh et al., 2012*; *Takahashi et al., 2003*). For example, N-type $Ca_V2.2$ channels coexpressed with membrane-anchored β subunits, such as β2a or β2e, show relatively slower inactivation kinetics, higher current density, and lower $PIP_2$ sensitivity than channels with the cytosolic β subunit, such as β2b, β2c, or β3 (*Keum et al., 2014*; *Kim et al., 2015a, Kim et al., 2015b*; *Kim et al., 2016*; *Suh et al., 2012*). However, the underlying mechanisms for the differential regulation of $Ca_V2.2$ channel gating depending on the subcellular localization of $Ca_V$ β subunits has not been clearly resolved.

Previous studies have proposed a bidentate model where two palmitoyl chains of the $Ca_V$ β2a subunit compete with the interaction of the two fatty acyl chains of $PIP_2$. Subsequently, this dislodges the $PIP_2$ molecule from its binding site on the N-type $Ca_V2.2$ channels, decreasing the requirement for $PIP_2$ (*Heneghan et al., 2009*; *Hille et al., 2015*; *Mitra-Ganguli et al., 2009*; *Roberts-Crowley and Rittenhouse, 2009*). Using cryo-electron microscopy (cryo-EM), *Dong et al., 2021* and *Gao et al., 2021* have recently shown that human $Ca_V2.2$ channels possess a $PIP_2$-binding pocket within the $S4_{II}$ domain of α1B subunit. $PIP_2$ interaction to this site is required for a minor shift of the $S4_{II}$ domain to the I–II loop. The functional role of the $PIP_2$-binding site in $Ca_V2.2$ channel gating and the modulatory effects of $Ca_V$ β subunits on the $PIP_2$ interaction are yet to be defined. In this study, we developed diverse engineered α1B and β constructs and found that the $Ca_V2.2$ channels were regulated by $PIP_2$ through at least two distinct interacting sites, including a nonspecific phospholipid-binding motif in the distal I–II loop and the binding pocket in the $S4_{II}$ domain. Our results revealed that the PM-anchored β2a subunit selectively disrupted $PIP_2$ interaction with the phospholipid-binding site in the I–II loop, leading to a channel state less sensitive to Dr-VSP-induced $PIP_2$ depletion. However, the $S4_{II}$-binding pocket of $Ca_V2.2$ channels interacted with $PIP_2$ regardless of the coupled β2 isotype. The present study provides new insights into the reciprocal roles of the $Ca_V$ β subunits and membrane $PIP_2$ in HVA $Ca_V$ channel regulation.

## Results

### N-terminal length of PM-tethering Ca$_V$ β subunit is important in determining current inactivation and PIP$_2$ sensitivity of Ca$_V$2.2 channels

We have previously reported that subcellular localization of the Ca$_V$ β subunit plays an important role in determining the inactivation kinetics and PIP$_2$ sensitivity of Ca$_V$2.2 channels (*Keum et al., 2014*; *Kim et al., 2015a*, *Kim et al., 2016*). By manipulating the β2 constructs, we further examined how Ca$_V$ β subunits determine the gating properties of the Ca$_V$2.2 channel depending on their subcellular localization. First, we used a palmitoylation-resistant cytosolic mutant form of β2a, β2a(C3,4S), where two palmitoylation sites (C3 and C4) in the N-terminus of the β2a subunit were mutated to serine residues (*Chien et al., 1996*; *Hurley et al., 2000*; *Olcese et al., 1994*; *Qin et al., 1998*; *Figure 1A* and *Figure 1—figure supplement 1A*). Additionally, we constructed two more membrane-recruited β2c analogs by adding membrane-targeting Lyn$_{11}$ (N-terminal G2-myristoylation and C3-palmitoylation modification sequence from Lyn kinase; *Resh, 1994*) or Lyn$_{11}$ plus a flexible 48 amino acid linker (Lyn-48aa) to the N-terminus of β2c. When these Ca$_V$ β constructs were expressed in cells without the pore-forming α1B, β2a(C3,4S) was distributed through the cytosol similar to β2c. By contrast, the engineered Lyn-β2c and Lyn-48aa-β2C were localized at the PM like the membrane-anchored β2a subunit (*Figure 1B, C*). However, in the presence of α1B and α2δ1, all the β2 constructs were mainly distributed at the PM, probably via binding to α1B subunits (*Figure 1—figure supplement 1B, C*). This suggested that amino acid mutation or chimeric modification of the β2 subunit does not affect the formation of the Ca$_V$2.2 channel multicomplex. Next, we tested the effects of the β2 constructs on current inactivation and PIP$_2$ sensitivity of the Ca$_V$2.2 channels. PIP$_2$ regulation of Ca$_V$2.2 channel gating was measured as the difference before and after a + 120 mV depolarizing pulse using Dr-VSP (see *Figure 1—figure supplement 2A*). Coexpression of β2a(C3,4S) accelerated current inactivation and increased the PIP$_2$ sensitivity of Ca$_V$2.2 channels, such as those with the cytosolic β2c subunit. Expression of the chimeric Lyn-β2c slowed down the inactivation rate and decreased PIP$_2$ sensitivity, like the channels with the PM-anchored β2a subunit (*Figure 1D, E*). Interestingly, cells co-transfected with the PM-tethered chimeric Lyn-48aa-β2c showed faster current inactivation and higher PIP$_2$ sensitivity in Ca$_V$2.2 channels, which were similar to the responses of channels with the cytosolic β2c subunit. In control experiments without Dr-VSP, we confirmed that the current amplitudes of Ca$_V$2.2 channels with the developed β2 constructs were not significantly different before and after the depolarizing pulse (*Figure 1—figure supplement 2A, B*). Additionally, we verified that the effects of Dr-VSP were not due to relieving the Gβγ-mediated tonic inhibition from the Ca$_V$2.2 channels. As shown in *Figure 1—figure supplement 2C*, prepulse depolarization did not change the current amplitudes in cells intracellularly perfused with 1 mM of the G protein inhibitor GDP-β-S instead of GTP in the absence of Dr-VSP. Moreover, the Ca$_V$2.2 channels with GDP-β-S showed very similar PI(4,5)P$_2$ sensitivities to those in experiments with GTP in cells expressing Dr-VSP (*Figure 1—figure supplement 2D*). This suggested that 0.1 mM GTP concentration in the pipette solution was not sufficient to trigger spontaneous G protein activation or suppress Ca$_V$2.2 channels through Gβγ binding.

We further examined the effects of the length of the flexible linker between Lyn and the β2c subunit on the inactivation kinetics and PIP$_2$ sensitivity of Ca$_V$2.2 channels. As shown in *Figure 1—figure supplement 3*, when the inserted linkers were longer than 24 aa, current inactivation was faster and current inhibition by PIP$_2$ depletion was stronger. Together, these data suggest that the N-terminal length of the PM-tethering Ca$_V$ β subunit is critical in determining the inactivation kinetics and PIP$_2$ sensitivity of Ca$_V$2.2 channels.

### Proximal interaction of the fatty acyl chains with channel complex underlies the β subunit-dependent regulation of Ca$_V$2.2 channel gating

It has been previously reported that disruption of the SH3–GK interaction in the membrane-anchored β2a subunit accelerates the channel inactivation of Ca$_V$2.1 channels (*Chen et al., 2009*). The GK domain of the Ca$_V$ β subunit interacts directly with the AID domain in the I–II loop of Ca$_V$ α1 subunits (*Buraei and Yang, 2010*; *Buraei and Yang, 2013*; *Chen et al., 2004*; *Opatowsky et al., 2004*; *Van Petegem et al., 2004*); therefore, disruption of the SH3–GK interaction in the Ca$_V$ β subunit may increase the length between the N-terminus and the GK–AID complex through the flexible HOOK region. To test the possible effects of increased N-terminal length from the AID–GK complex on Ca$_V$

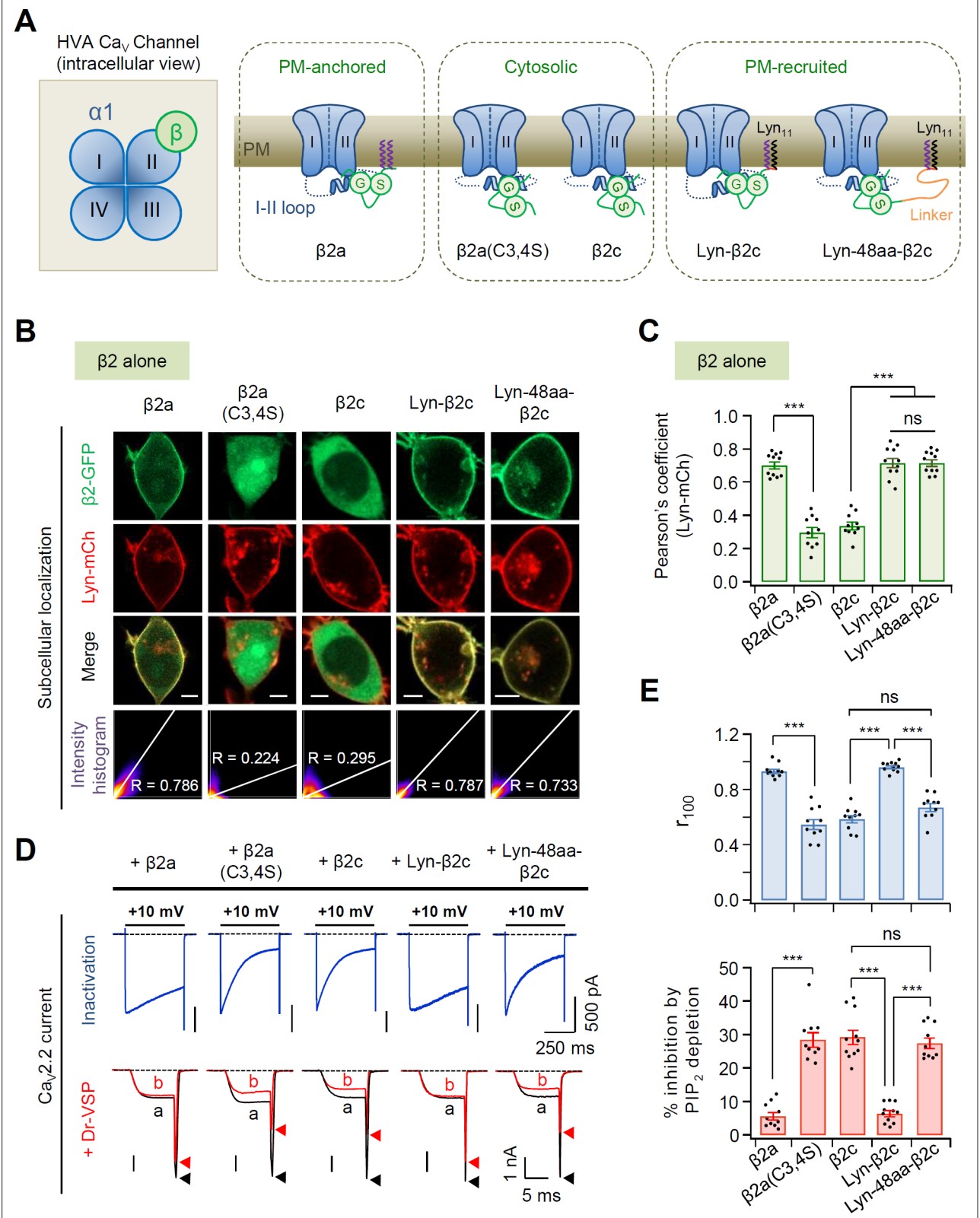

**Figure 1.** Current inactivation and PIP$_2$ sensitivity in N-type Ca$_V$2.2 channels with different subtypes of the β2 subunit. (**A**) Schematic diagram of high-voltage-activated (HVA) calcium channel complex viewed from the intracellular side (left). Ca$_V$ β subunit is located beside the domain II of α1B in the cytosolic side while Ca$_V$ α2δ subunit is mostly localized at the extracellular surface of the channel protein (*Gao et al., 2021*). Schematic model of Ca$_V$2.2 channels with plasma membrane (PM)-anchored β2a, cytosolic β2a(C3,4S) and β2c, or N-terminus engineered PM-recruited β2c (right). (**B**)

*Figure 1 continued*

Representative confocal images of tsA-201 cells expressing the PM marker Lyn-mCh and β2 isoforms or its derivatives fused to GFP without the α1 and α2δ1 subunits. Scale bar, 5 µm. The scatter plot shows a 2D intensity histogram of the red (Lyn-mCh) and green (β2-GFP) pixels in the confocal image. The value indicates the Pearson's correlation coefficient ($R$) that is obtained by the Colocalization Threshold plugin of Fiji software (Image J). (**C**) Summary of Pearson's coefficient between Lyn-mCh and the β2 construct ($n$ = 10–11). (**D**) Current inactivation of $Ca_V2.2$ channels with β2 isoforms or its derivatives was measured during 500-ms test pulses to +10 mV (top). Current inhibition of $Ca_V2.2$ channels by Dr-VSP-mediated $PIP_2$ depletion (bottom). The current traces before (**a**) and after (**b**) the strong depolarizing pulse to +120 mV were superimposed. Peak tail current is indicated by arrowheads (trace a, black head; trace b, red head). (**E**) Summary of current inactivation (top; $n$ = 10–11) and inhibition (%) by $PIP_2$ depletion (bottom; $n$ = 10–11) in $Ca_V2.2$ channels with the β2 constructs. $r_{100}$ indicates the fraction of current remaining after 100-ms depolarization to +10 mV (top). Dots indicate the individual data points for each cell. Data are mean ± standard error of the mean (SEM). ***$p < 0.001$, using one-way analysis of variance (ANOVA) followed by Tukey post hoc test.

The online version of this article includes the following source data and figure supplement(s) for figure 1:

**Source data 1.** Current inactivation ($r_{100}$) and current inhibition (%) by $PIP_2$ depletion in N-type $Ca_V2.2$ channels with different subtypes of the β2 subunit.

**Figure supplement 1.** Subcellular localization of N-terminus engineered constructs of β2 subunit in the presence of α1 and α2δ1 subunits.

**Figure supplement 1—source data 1.** Pearson's coefficient between Lyn-mCh and the β2 construct in the presence of α1 and α2δ1 subunit.

**Figure supplement 2.** Summary of the $Ca_V2.2$ current inhibition by a 120-mV-depolarizing pulse in cells without or with Dr-VSP.

**Figure supplement 2—source data 1.** Current inhibition (%) by a depolarizing pulse in cells in the absence of Dr-VSP.

**Figure supplement 2—source data 2.** Summary of the $Ca_V2.2$ current inhibition by Dr-VSP-mediated $PIP_2$ depletion in cells were recorded with pipette solution containing GDP-β-S.

**Figure supplement 3.** Effects of inserting a flexible linker between Lyn and β2c subunit on current inactivation and $PIP_2$ sensitivity of $Ca_V2.2$ channels.

**Figure supplement 3—source data 1.** Current inactivation ($r_{100}$) and current inhibition (%) by $PIP_2$ depletion in $Ca_V2.2$ channels with chimeric Lyn-linker-β2c derivatives.

channel gating, we constructed mutant β2a subunits in which the SH3–GK intramolecular interaction was disrupted by mutating seven amino acids in the SH3 and GK domains to alanine residues (*Figure 2A*). Additionally, the N-terminus was deleted to abolish membrane targeting of the β2a subunit by itself, and $Lyn_{11}$ was inserted to the N-terminus for membrane recruitment. Without α1B and α2δ1, both N-terminus-deleted (ΔN)β2 WT and (ΔN)β2 Mut, in which the SH3–GK interaction was disrupted, were expressed in the cytosol. Conversely, Lyn-(ΔN)β2 WT and Lyn-(ΔN)β2 Mut constructs were localized to the PM (*Figure 2A*, inset images). In $Ca_V2.2$ channels with the N-terminus-deleted mutant (ΔN)β2 WT, the current exhibited fast inactivation and high $PIP_2$ sensitivity (*Figure 2B–D*). These phenomena similarly appeared in channels with the (ΔN)β2 Mut. In contrast, $Ca_V2.2$ channels with Lyn-(ΔN)β2 WT exhibited slow inactivation and weak $PIP_2$ sensitivity. However, the channels with Lyn-(ΔN)β2 Mut exhibited fast inactivation and strong $PIP_2$ sensitivity, like channels with cytosolic (ΔN)β2 WT and (ΔN)β2 Mut (*Figure 2B–D*). We also confirmed that disruption of the SH3–GK interaction did not shift the current–voltage (*I–V*) curve of $Ca_V2.2$ currents (*Figure 2—figure supplement 1*). These data suggested that the length from the N-terminal lipid anchor to the GK domain of β subunit is crucial in determining the inactivation rate and $PIP_2$ sensitivity of $Ca_V2.2$ channels.

To further examine the functional role of length between lipid anchor and GK domain on $Ca_V$ channel gating in live cells, we developed new chimeric β2 constructs by applying the rapamycin-induced dimerizing system FK506-binding protein (FKBP) and FKBP–rapamycin-binding (FRB) protein (*Banaszynski et al., 2005*; *Inoue et al., 2005*; *Suh et al., 2006*). As shown in *Figure 3A*, FKBP and FRB proteins irreversibly assembled to form a ternary complex upon application of rapamycin, which led to shortening of the length between the lipid anchor $Lyn_{11}$ and GK–AID domains. We fused a Förster resonance energy transfer (FRET) probe YFP to the C-terminus of all β2 chimera to investigate whether the FKBP domain was really translocated to the PM to make a $Lyn_{11}$-FRB and FKBP complex closely after rapamycin addition (*Figure 3A*, right diagram). In experiments with the β chimera without the FKBP domain (Control: Lyn-FRB-HOOK-GK), both FRETr and the current amplitude of $Ca_V2.2$ channels were not changed by rapamycin addition (*Figure 3B*). Consistently, rapamycin treatment did not affect current inactivation and the $PIP_2$ sensitivity of $Ca_V2.2$ channels in these cells (*Figure 3C–F*). In contrast, in $Ca_V2.2$ channels with Lyn-FRB-HOOK-GK-FKBP (RF), rapamycin treatment irreversibly enhanced the FRETr signal and increased the current amplitude of $Ca_V2.2$ channels (*Figure 3B*, middle and *Figure 3—figure supplement 1*). Moreover, rapamycin treatment reduced the current inactivation and $PIP_2$ sensitivity of $Ca_V2.2$ channels (*Figure 3C–F*). However, in $Ca_V2.2$ channels with

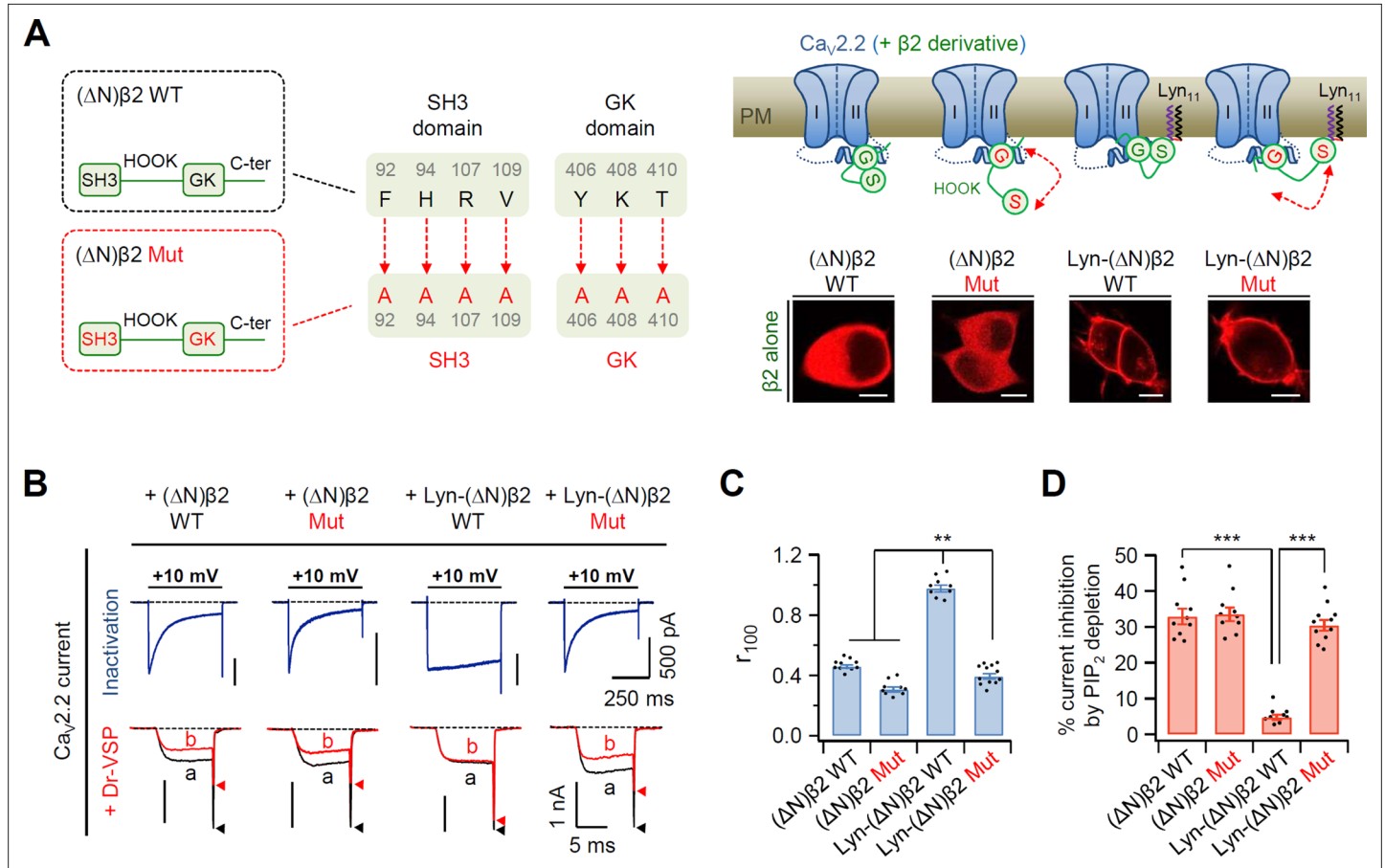

**Figure 2.** Disruption of SH3–GK interaction in the plasma membrane (PM)-recruited Ca$_V$ β2 subunit leads to an increase in both current inactivation and PIP$_2$ sensitivity of Ca$_V$2.2 channels. (**A**) Left, a diagram showing how the SH3–GK intramolecular interaction is disrupted in β2 constructs (top). Phenylalanine 92, histidine 94, arginine 107, and valine 109 residues in the SH3 domain and tyrosine 406, lysine 408, and threonine 410 residues in the GK domain are replaced with alanine. Schematic model of Ca$_V$2.2 channels with engineered β2 constructs in which the SH3–GK intramolecular interaction is disrupted. Lyn-(ΔN)β2: Lyn-labeled N-terminus-deleted β2 construct. Lyn-(ΔN)β2 Mut: Lyn-(ΔN)β2 construct with a disrupted SH3–GK intramolecular interaction. Inset: confocal images of tsA-201 cells expressing engineered β2 constructs labeled with mCherry without α1B and α2δ1 subunits. Scale bar, 5 μm. (**B**) Representative currents of Ca$_V$2.2 channels with engineered β2 constructs. The currents were measured during 500-ms test pulses to +10 mV (top). Current traces before (**a**) and after (**b**) a + 120-mV depolarizing pulse in cells expressing Ca$_V$2.2 channels with engineered β2 constructs and Dr-VSP (bottom). Peak tail current is indicated by arrowheads (trace a, black head; trace b, red head). (**C**) Summary of Ca$_V$2.2 current inactivation (n = 9–12). $r_{100}$ indicates the fraction of current remaining after 100-ms depolarization to +10 mV. (**D**) Summary of Ca$_V$2.2 current inhibitions (%) by PIP$_2$ depletion in Dr-VSP-expressing cells (n = 9–11). Dots indicate the individual data points for each cell. Data are mean ± standard error of the mean (SEM). **p < 0.01, ***p < 0.001, using one-way analysis of variance (ANOVA) followed by Tukey post hoc test.

The online version of this article includes the following source data and figure supplement(s) for figure 2:

**Source data 1.** Current inactivation ($r_{100}$) and current inhibition (%) by PIP$_2$ depletion in N-type Ca$_V$2.2 channels with the engineered β2 construct.

**Figure supplement 1.** Disruption of the SH3–GK intramolecular interaction of β2 subunit does not shift current–voltage (I–V) curve of Ca$_V$2.2 current.

**Figure supplement 1—source data 1.** Current–voltage (I–V) curve of Ca$_V$2.2 current.

Lyn-FRB-HOOK-GK-Linker-FKBP (RCF), where a 194-aa linker was inserted between GK and FKBP, rapamycin enhanced the FRETr signal without causing significant changes in the current amplitude (*Figure 3B*, right and *Figure 3—figure supplement 1*). The effects of rapamycin on inactivation kinetics and PIP$_2$ sensitivity were much weaker in Ca$_V$2.2 channels with RCF when compared with those in channels with RF (*Figure 3C–F*). This suggested that rapamycin-induced dimerization may be insufficient to shorten the length between the lipid anchor and isolated GK domain of β subunit in channels with RCF.

Next, we measured the effects of the N-terminal length of PM-tethered β subunit on Ca$_V$2.2 channel activity by inserting flexible linkers of various lengths between Lyn$_{11}$ and the GK domain of β2

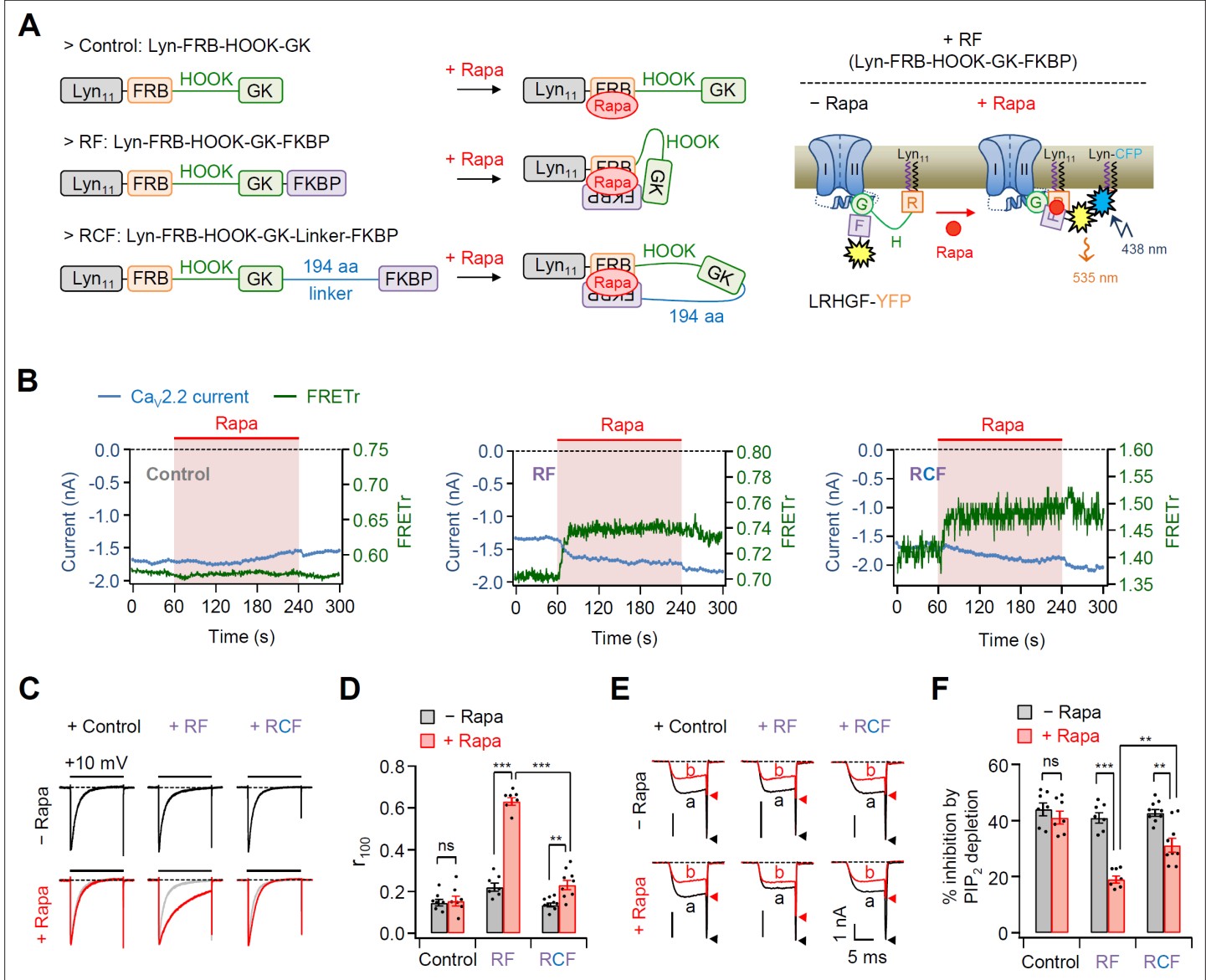

**Figure 3.** Effects of the real-time translocation of the GK domain to the plasma membrane (PM) on Ca$_V$2.2 channel gating. (**A**) Left, a schematic diagram showing rapamycin-induced translocatable β2 chimeric constructs. Translocatable β2 chimeric constructs were invented by fusing FRB or FKBP to the N- and C-termini of the GK domain, respectively. The new constructs were tagged with Lyn$_{11}$ (RF or Lyn-FRB-Hook-GK-FKBP) to be tethered to the PM. Rapamycin (Rapa) addition triggers the formation of a tripartite FRB–rapamycin–FKBP complex, resulting in the movement of the FKBP domain to the PM (right). For Förster resonance energy transfer (FRET) imaging, chimeric β constructs labeled with YFP in the C-terminus and PM-targeting Lyn-CFP were coexpressed. Right, schematic model of Ca$_V$2.2 channels with RF before and after rapamycin application. Rapamycin induces the formation of the tripartite complex, resulting in a shift of the FKBP domain to the PM and an enhanced FRET signal. (**B**) Time courses of Ca$_V$2.2 currents (blue traces) and FRET ratio (green traces) were measured simultaneously in single cells expressing Ca$_V$2.2 channels with Cont (left), RF (middle), or RCF (right) and the membrane marker Lyn-CFP. (**C**) Current inactivation of Ca$_V$2.2 channels with Cont (left), RF (middle), and RCF (right) was measured during 500-ms test pulses to +10 mV before (black traces) and after (red traces) rapamycin addition. (**D**) Summary of inactivation of Ca$_V$2.2 currents before (black bars) and after (red bars) rapamycin application (*n* = 7–9). The fraction of the current remaining after 100-ms depolarization ($r_{100}$) to +10 mV. (**E**) Current inhibition of Dr-VSP-mediated PIP$_2$ depletion on Ca$_V$2.2 channels with Cont (left), RF (middle), and RCF (right) before and after rapamycin addition. The traces before (**a**) and after (**b**) the depolarizing pulse to +120 mV were superimposed. Peak tail current is indicated by arrowheads (trace a, black head; trace b, red head). (**F**) Summary of Dr-VSP-induced Ca$_V$2.2 current inhibition before (black bars) and after (red bars) rapamycin addition (*n* = 7–9). Dots indicate the individual data points for each cell. Data are mean ± standard error of the mean (SEM). **$p < 0.01$, ***$p < 0.001$, using two-way analysis of variance (ANOVA) followed by Sidak post hoc test.

The online version of this article includes the following source data and figure supplement(s) for figure 3:

**Source data 1.** Time courses of Ca$_V$2.2 currents and Förster resonance energy transfer (FRET) ratio.

*Figure 3 continued on next page*

Figure 3 continued

**Source data 2.** Current inactivation ($r_{100}$) and current inhibition (%) by PIP$_2$ depletion in Ca$_V$2.2 channels with rapamycin-induced translocatable β2 chimeric constructs before and after rapamycin.

**Figure supplement 1.** The real-time translocation of the GK domain to the plasma membrane increased the current amplitude of Ca$_V$2.2 channels.

**Figure supplement 1—source data 1.** Relative peak current amplitudes of Ca$_V$2.2 channels with chimeric Lyn-linker-β2c derivatives.

(**Figure 4A**). The inserted linkers were unstructured flexible peptides (see **Figure 4—figure supplement 1**); therefore, the length of the linkers was calculated using the worm-like chain (WLC) model (see Methods). Our results showed that both the current inactivation and PIP$_2$ sensitivity of Ca$_V$2.2 channels became gradually stronger as the inserted flexible linkers became longer (**Figure 4B–D**). Consistently, the current activation was gradually accelerated by the increase in linker length (**Figure 4—figure supplement 2**). However, no additional difference was detected in channels with the membrane-tethered Lyn-43aa-GK subunit when compared with the cytosolic GK subunit. This indicated that the GK domain with the length of the inserted 43-aa linker is sufficient to act like the cytosolic Ca$_V$ β subunit (**Figure 4B–D**). Interestingly, the PIP$_2$ sensitivity and inactivation kinetics of Ca$_V$ channels were differentially regulated by the length between the lipid anchor and the GK domain: the channels with Lyn-43aa-GK showed faster inactivation than the channels with Lyn-22aa-GK, whereas the PIP$_2$ sensitivity of the two channels was not significantly different (**Figure 4B–E**). Additionally, our data analysis indicated that the biophysical gating properties of Ca$_V$2.2 channels with a membrane-anchored β2a subunit were similar to those of channels with Lyn-9aa-GK. Furthermore, the gating properties of Ca$_V$2.2 channels coupled with cytosolic β2c were similar to those of channels with Lyn-20aa-GK (**Figure 4E**).

Previous studies have reported that subcellular localization of the Ca$_V$ β subunit is important in determining the current density of Ca$_V$ channels, where Ca$_V$ channels with the membrane-anchored β subunit show relatively higher current density than channels with the cytosolic β subunit (**Suh et al., 2012**). In line with this, we found that the current density of Ca$_V$2.2 channels with β2a was significantly higher than that of channels with β2c (**Figure 4—figure supplement 3A–C**). Therefore, we tested whether the current density of Ca$_V$ channels was dependent on the N-terminal length. Ca$_V$2.2 channels showed slightly decreased current density that was dependent on the expansion of the flexible linker length between Lyn and the GK domain alone (**Figure 4—figure supplement 4A, C**). This phenomenon was observed in channels with the whole β2c subunit with Lyn (**Figure 4—figure supplement 3D–F**). We tested whether the length between N-terminal lipid anchor and GK domain affected the voltage-dependent gating of Ca$_V$ channels. Voltage-dependent activation of Ca$_V$2.2 channels with Lyn-linker-GK derivatives showed a greater shift to positive voltage as the inserted flexible linkers increased in length (**Figure 4—figure supplement 4B, D**). This suggested that incremental increases in linker length lead to a decreased voltage sensitivity. There was no difference in the current density and voltage-dependent activation between Ca$_V$ channels with the Lyn-43aa-GK and GK subunit. Together, these results suggested that differential regulation of Ca$_V$2.2 channel gating by β subunits is mainly determined by the anchoring properties of the β subunits to PM.

## Polybasic motif at the C-terminal end of I–II loop plays an important role in the PIP$_2$ regulation of Ca$_V$2.2 channels

How does the N-terminal length of the PM-tethering Ca$_V$ β subunit regulate Ca$_V$ channel gating? Recently, **Kaur et al., 2015** have reported that a polybasic motif consisting of four basic amino acids within the C-terminal end of the I–II loop of L-type Ca$_V$1.2 channels interacts with membrane phospholipids, including PIP$_2$. Additionally, the putative PIP$_2$-binding site is conserved in the I–II loop of N-type Ca$_V$2.2 channels (**Figure 5—figure supplement 1**). We examined whether the polybasic motif affects the PIP$_2$ sensitivity of Ca$_V$2.2 channels. First, we eliminated the potential phospholipid-binding motif from the Ca$_V$2.2 channel I–II loop by mutating the four polybasic residues to alanine (4A α1B) (**Figure 5A**). In Ca$_V$2.2 channels with the β2a subunit, the inactivation kinetics of the current did not differ between WT α1B and 4A α1B (**Figure 5B, C**, left). However, in Ca$_V$2.2 with β2c, the inactivation rate was slower in 4A α1B channels (**Figure 5B, C**, right). The effects of PIP$_2$ depletion on current amplitude were also measured in these channels. In control experiments without Dr-VSP, the current of WT or 4A-mutant Ca$_V$2.2 channels did not significantly differ before and after a + 120-mV depolarizing

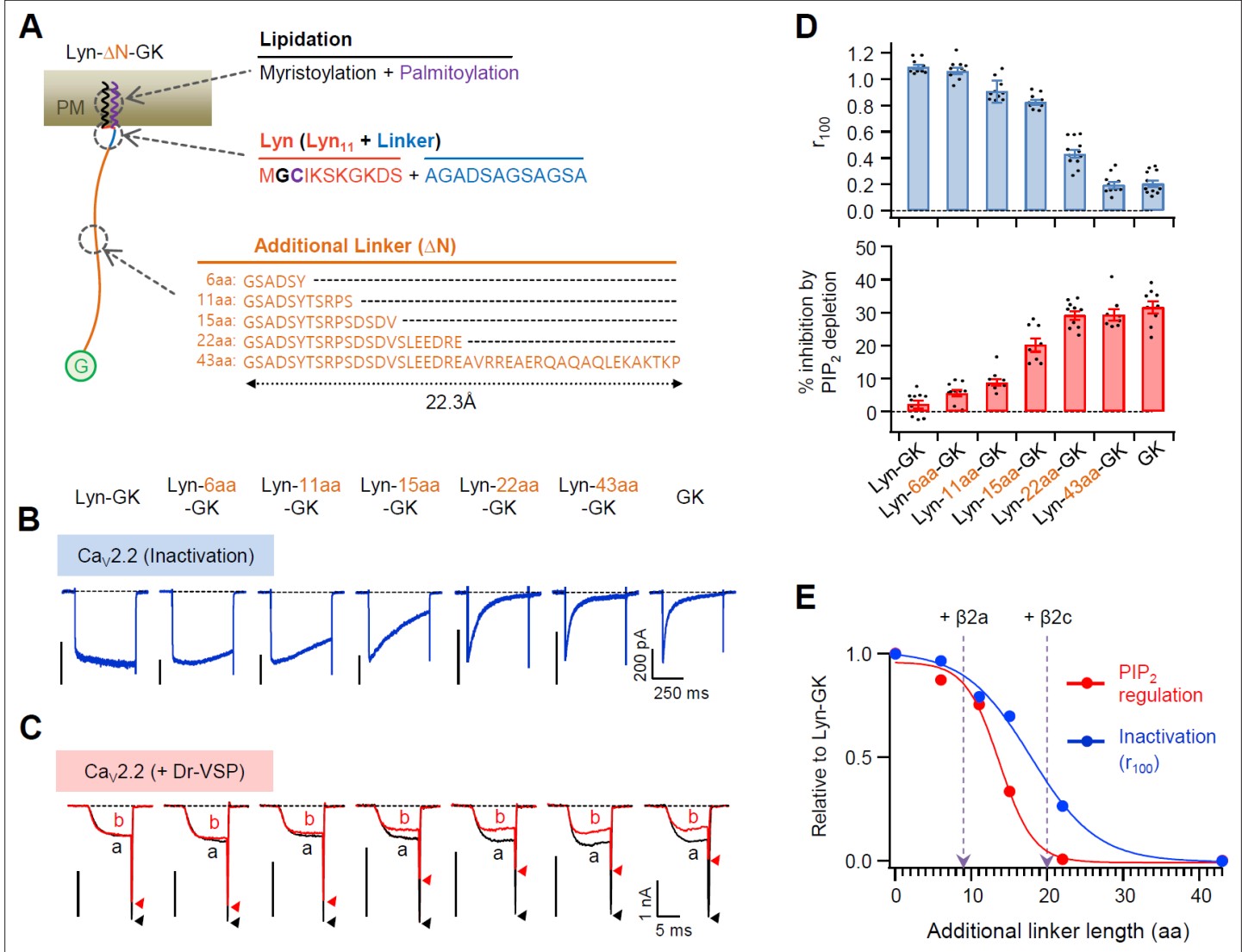

**Figure 4.** Flexible linker length between Lyn and the GK domain of the β subunit performs a key role in determining both the current inactivation and the PIP$_2$ sensitivity of Ca$_V$2.2 channels. (**A**) Schematic diagram of diverse flexible linkers (ΔN) inserted between Lyn and GK (G) domain. The length of each linker is calculated by the worm-like chain (WLC) model (see Methods). Amino acid sequences of Lyn (Lyn$_{11}$ plus 12 aa linker) and the additional linkers are listed. (**B**) Current inactivation of Ca$_V$2.2 channels with diverse Ca$_V$ β-GK derivatives was measured during 500 ms test pulses to +10 mV. (**C**) Effects of Dr-VSP-mediated PIP$_2$ depletion on Ca$_V$2.2 channels with GK domain derivatives. Peak tail current is indicated by arrowheads (trace a, black head; trace b, red head). (**D**) Summary of current inactivation (blue bars; $n$ = 9–12) and inhibition (%) by PIP$_2$ depletion (red bars; $n$ = 8–10) in Ca$_V$2.2 channels with Ca$_V$ β GK derivatives. Data are mean ± standard error of the mean (SEM). Dots indicate the individual data points for each cell. (**E**) Normalized mean current inactivation and mean current inhibition by PIP$_2$ depletion versus additional linker length (aa) of Ca$_V$ β GK derivatives measured in Ca$_V$2.2 channels. The normalized current regulation in cells expressing Ca$_V$2.2 with β2a and β2c is indicated with dashed arrows.

The online version of this article includes the following source data and figure supplement(s) for figure 4:

**Source data 1.** Current inactivation ($r_{100}$) and current inhibition (%) by PIP$_2$ depletion in Ca$_V$2.2 channels with the engineered β2 GK derivatives.

**Figure supplement 1.** IUPRED web-server result of inserted linker.

**Figure supplement 2.** Summary of time constants for Ca$_V$2.2 current activation.

**Figure supplement 2—source data 1.** Time constants of current activation in Ca$_V$2.2 channels with diverse Ca$_V$ β-GK derivatives.

**Figure supplement 3.** Current density in N-type Ca$_V$2.2 channels with β2 variants.

**Figure supplement 3—source data 1.** Population current density versus voltage relations for Ca$_V$2.2 channels with β2 variants.

**Figure supplement 4.** Flexible linker length between Lyn and GK domain of β subunit is important in determining the current density and the voltage-dependent gating of Ca$_V$2.2 channels.

*Figure 4 continued on next page*

*Figure 4 continued*

**Figure supplement 4—source data 1.** Population current density versus voltage relations and the voltage dependence of normalized steady-state activation for $Ca_V2.2$ channels with the engineered β2 GK derivatives.

pulse in cells with either β2a or β2c subunits (*Figure 5D*). By contrast, $PIP_2$ depletion by Dr-VSP activation similarly inhibited the $Ca_V$ current by approximately 5% in cells expressing either WT or 4A $Ca_V2.2$ channels with a PM-anchored β2a subunit (*Figure 5E*, left). This indicated the presence of another $PIP_2$-binding site in the α1B subunit other than this polybasic motif in I–II loop. On the other hand, the $PIP_2$ sensitivity in channels with β2c was dramatically reduced in 4A channels, indicating that the polybasic motif in the I–II loop plays a key role in $PIP_2$ regulation of $Ca_V2.2$ channels with the cytosolic β subunit (*Figure 5D, E*). However, in cells expressing 4A $Ca_V2.2$ channels with β2c, we observed another ~5% current inhibition by $PIP_2$ depletion. This was similar to the $Ca_V2.2$ channels with β2a.

Next, we investigated whether the polybasic motif affects the $PIP_2$ sensitivity of $Ca_V2.2$ channels with Lyn-β2c and Lyn-48aa-β2c (*Figure 5—figure supplement 2*). Similar to β2a, we did not detect any significant differences in current inactivation and $PIP_2$ sensitivity between WT and 4A mutant $Ca_V2.2$ with Lyn-β2c (*Figure 5—figure supplement 2*). Conversely, WT $Ca_V2.2$ channels with Lyn-48aa-β2c exhibited faster inactivation and higher $PIP_2$ sensitivity, which was similar to the responses of $Ca_V2.2$ channels with cytosolic β2c. However, in cells expressing 4A mutant $Ca_V2.2$ channels with Lyn-48aa-β2c, the current inactivation was slowed and the $PIP_2$ sensitivity was decreased to ~5% (*Figure 5—figure supplement 2*). The $PIP_2$ sensitivities of 4A $Ca_V2.2$ channels with Lyn-β2c and Lyn-48aa-β2c did not significantly differ and were similar to that of WT channels with Lyn-β2c. Consistent with the data in *Figure 5*, these results suggested that the polybasic motif within the I–II loop interacts with membrane $PIP_2$ in $Ca_V2.2$ channels with β2c-like Lyn-48aa-β2c, but not with β2a-like Lyn-β2c subunits. On the other hand, in channels with the β2a subunit, there was no significant difference in the voltage-dependent activation between WT α1B and 4A α1B (*Figure 5F, G*). However, the activation of 4A α1B with the β2c subunit was significantly shifted toward the hyperpolarization direction when compared with WT α1B channels with β2c (*Figure 5F, G*). In addition, the activation curve of 4A α1B with β2c was similar to the curves of WT and 4A α1B with β2a (*Figure 5F, G*). Together, our data suggested that two different $PIP_2$-interacting sites with differential $PIP_2$ sensitivities exist in $Ca_V2.2$ channels. More importantly, our data indicate that $PIP_2$ interacts with the polybasic motif when $Ca_V2.2$ is expressed with cytosolic β subunits but not when expressed with lipidated membrane-anchored β subunit.

Finally, we determined whether other arginine residues in the distal region of polybasic motif also affected the $PIP_2$ sensitivity of $Ca_V2.2$ channels (*Figure 5—figure supplement 3*). For this, two arginine residues (R476 and R477) near the polybasic motif were replaced with alanine (α1B R476,477A) (*Figure 5—figure supplement 3A*). We also constructed a α1B R465,466A by mutating only two arginine residues (R465 and R466) in the polybasic motif (R465, R466, K469, and R472) (*Figure 5—figure supplement 3A*). In $Ca_V2.2$ channels with the β2a subunit, we did not detect any significant differences in current inactivation and $PIP_2$ sensitivity among WT α1B, α1B R465,466A, and α1B R476,477A (*Figure 5—figure supplement 3B–E*). However, in $Ca_V2.2$ with β2c, the inactivation rate was slower and the $PIP_2$ sensitivity was weaker in both α1B R465,466A and α1B R476,477A compared to WT α1B (*Figure 5—figure supplement 3B–E*).

## Differential modulation of $Ca_V2.2$ channels by muscarinic receptor stimulation in cells expressing PM-anchored or cytosolic β subunit

To examine whether the polybasic motif influenced the $G_q$-coupled modulation of $Ca_V2.2$ channels, we applied the muscarinic acetylcholine receptor agonist, oxotremorine-M (Oxo-M), to cells co-transfected with the $M_1$ muscarinic receptor ($M_1$R) (*Figure 6*). Since the $M_1$R stimulation suppressed $Ca_V2.2$ channels through both Gβγ binding to channels and $PIP_2$ depletion (*Keum et al., 2014*), we then used a Gβγ-insensitive chimeric $Ca_V2.2$ channel construct, α1C-1B, to examine the effect of $PIP_2$ depletion alone on channel regulation (*Figure 6*). In this chimera construct, the N-terminus of $Ca_V2.2$ (α1B), which contains one of the Gβγ interaction sites, is replaced by the N-terminus of $Ca_V1.2$ (α1C) (*Agler et al., 2005*). $M_1$R activation inhibited the current by approximately 5% in cells expressing either α1C-1B WT or 4A channels with β2a subunit, which were similar to the responses of regulation by Dr-VSP-mediated $PIP_2$ depletion in those channels (*Figure 6B, C*). However, consistent with the results

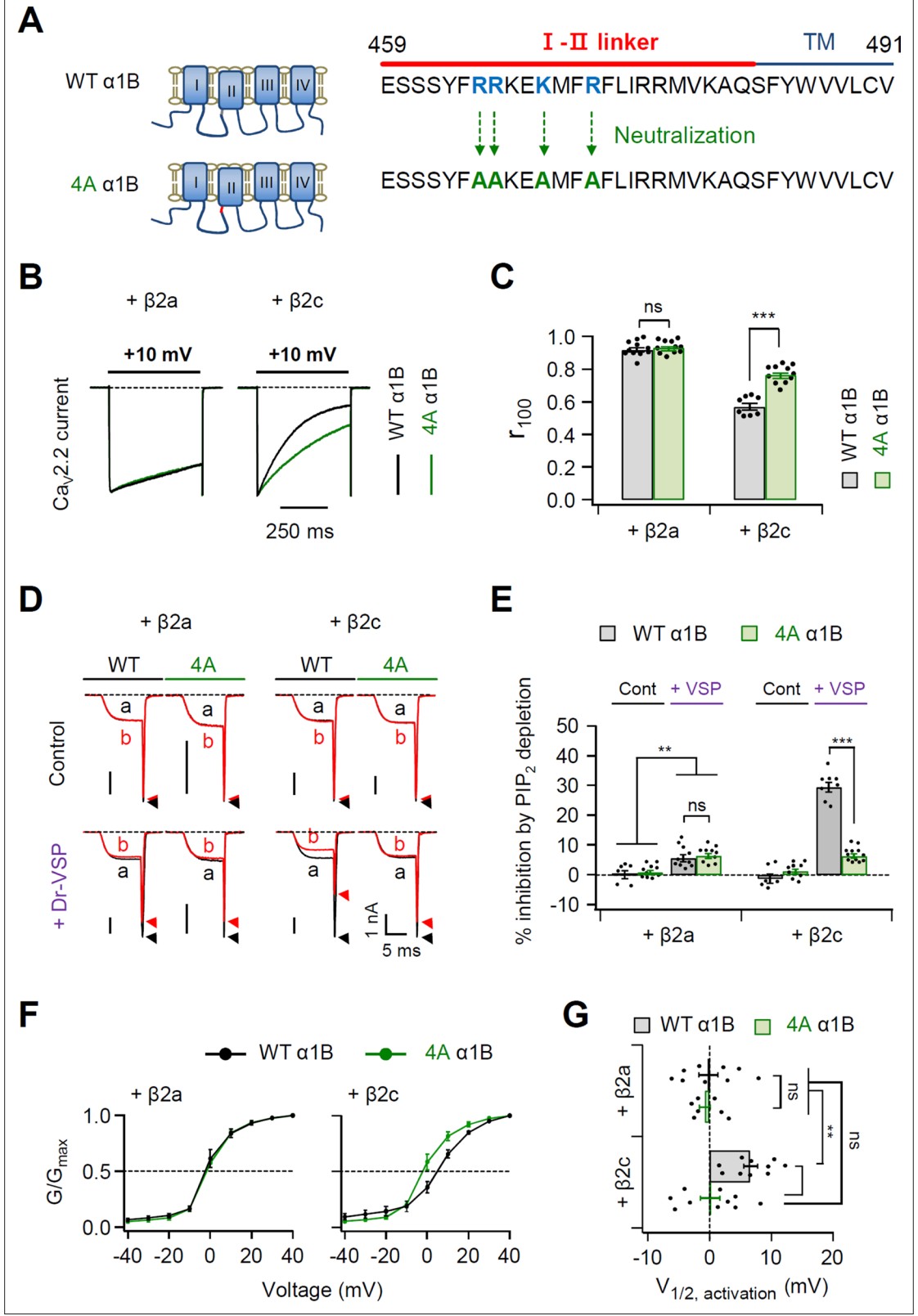

**Figure 5.** Polybasic motif at the C-terminal end of the I–II loop influences determination of steady-state activation, current inactivation, and PIP$_2$ sensitivity of Ca$_V$2.2 channels. (**A**) Schematic diagram of phospholipid-binding residue-neutralizing mutations within the C-terminal end of the I–II loop in the α1B subunit. The phospholipid-binding residues (R465, R466, K469, and R472) highlighted in blue were mutated to alanine (4A). (**B**) Current inactivation was measured during 500-ms test pulses to +10 mV in cells expressing α1B WT (black traces) and 4A mutants (green traces) with β2a (left) or

*Figure 5 continued on next page*

*Figure 5 continued*

β2c (right) subunits. (**C**) Summary of current inactivation of $Ca_V2.2$ WT (black bars) and 4A (red bars) with β2 subunits ($n$ = 8–11). $r_{100}$ indicates the fraction of current remaining after 100-ms depolarization to +10 mV. (**D**) Current inhibition by Dr-VSP-mediated $PIP_2$ depletion in cells expressing $Ca_V2.2$ WT and 4A with the β2a (left) or β2c subunit (right). $Ca_V2.2$ currents before (**a**) and after (**b**) the depolarizing pulse to +120 mV are superimposed in control (top) and Dr-VSP-expressing (bottom) cells. Peak tail current is indicated by arrowheads (trace a, black head; trace b, red head). (**E**) Summary of current inhibition (%) of $Ca_V2.2$ WT (black bars) and 4A (red bars) by $PIP_2$ depletion in control ($n$ = 10) and Dr-VSP-transfected cells ($n$ = 8–12). (**F**) The voltage dependence of normalized steady-state activation ($G/G_{max}$) for α1B WT (black) and 4A mutants (green) with β2a (left) or β2c (right) subunits. Tail currents elicited between −40 and +40 mV in 10 mV steps, from a holding potential of −80 mV were normalized to the largest tail current in each series of test pulse. The curves were fitted by a Boltzmann function. Dashed line indicates the $V_{1/2}$ of normalized steady-state activation. (**G**) Summary of the $V_{1/2}$ of normalized steady-state activation in cells expressing α1B WT (black bars) and 4A mutants (green bars) with β2a (upper) or β2c (bottom) subunits ($n$ = 7–10). Dots indicate the individual data points for each cell. Data are mean ± standard error of the mean (SEM). \*\*p < 0.01, \*\*\*p < 0.001, using two-way analysis of variance (ANOVA) followed by Sidak post hoc test.

The online version of this article includes the following source data and figure supplement(s) for figure 5:

**Source data 1.** Current inactivation ($r_{100}$) and current inhibition (%) by $PIP_2$ depletion in cells expressing α1B WT and 4A mutants with β2a and β2c.

**Figure supplement 1.** Sequence alignment of the C-terminal end of the I–II loop in $Ca_V$ α1 subunits.

**Figure supplement 2.** Current inactivation and $PIP_2$ sensitivity of mutant $Ca_V2.2$ channels with Lyn-β2c and Lyn-48aa-β2c.

**Figure supplement 2—source data 1.** Current inactivation ($r_{100}$) and current inhibition (%) by $PIP_2$ depletion in $Ca_V2.2$ channels with Lyn-β2c and Lyn-48aa-β2c.

**Figure supplement 3.** Neutralization of polybasic residues in the distal end of the I–II loop domain plays a crucial role in determining current inactivation and $PIP_2$ sensitivity of $Ca_V2.2$ channels with β2c.

**Figure supplement 3—source data 1.** Current inactivation ($r_{100}$) and current inhibition (%) by $PIP_2$ depletion in cells expressing WT α1B, α1B R465,466A, and α1B R476,477A with β2a and β2c.

for Dr-VSP-induced channel modulation, current suppression was much weaker in α1C-1B 4A channels with β2c than in α1C-1B WT with β2c (*Figure 6B, C*). We confirmed that the current suppression by $M_1R$ activation were not recovered in both α1C-1B WT and α1C-1B 4A channels by a prepulse regardless of the coupled β2 isotypes (*Figure 6D, E*). We additionally used $G_i$-coupled $M_2$ muscarinic receptor ($M_2R$) to further examine whether the polybasic motif in I–II loop affects the Gβγ-mediated modulation of $Ca_V2.2$ channels (*Figure 6—figure supplement 1*). $M_2R$ activation inhibited the currents evoked by a + 10-mV test pulse without significant difference between WT and 4A α1B with β2a or β2c (*Figure 6—figure supplement 1B, C*). $M_2R$ activation commonly slowed down the activation kinetics of $Ca_V2.2$ currents (*Figure 6—figure supplement 1D*). We have previously reported that subcellular localization of the $Ca_V$ β subunit plays important roles in determining the Gβγ-dependent inhibition of $Ca_V2.2$ channels; membrane-anchored β2a subunit changes $Ca_V2.2$ channels are more sensitive to Gβγ-mediated voltage-dependent inhibition, whereas cytosolic β2b and β3 subunit changes channels are less sensitive to Gβγ-mediated voltage-dependent inhibition (*Keum et al., 2014*). In $Ca_V2.2$ channels with the β2a subunit, the recoveries from Gβγ-mediated inhibition did not significantly differ between WT α1B and 4A α1B (*Figure 6—figure supplement 1E, F*). However, in $Ca_V2.2$ with the β2c subunit, there was less recovery from Gβγ-mediated inhibition in α1B WT than in α1B 4A (*Figure 6—figure supplement 1E, F*). Recovery from $M_2R$-mediated inhibition in 4A α1B with β2c was similar to the values of WT and 4A α1B with β2a (*Figure 6—figure supplement 1E, F*).

## $PIP_2$-binding site in $S4_{II}$ domain is important in maintaining the $Ca_V2.2$ channel activity regardless of the coupled β2 isotype

Recently, the cryo-electron microscopic structure of human $Ca_V2.2$ complex composed of α1B, α2δ1, and β3 subunits was revealed at a resolution of 3.0 Å (*Dong et al., 2021*; *Gao et al., 2021*). These studies have shown that the 5-phosphate group of membrane $PIP_2$ interacts with two basic residues (R584 and K587) within $S4_{II}$ domain of α1B. We examined whether the two basic residues affect the $PIP_2$ sensitivity of $Ca_V2.2$ channels. First, we constructed neutralized mutant α1B subunits in which the two basic residues in $S4_{II}$ were replaced by alanine residues (α1B RA/KA) (*Figure 7A*). In $Ca_V2.2$ channels with β2a, the inactivation kinetics of the current were not changed in α1B and α1B RA/KA, regardless of the 4A mutation (*Figure 7B, C*). In $Ca_V2.2$ with β2c, WT α1B RA/KA showed faster inactivation than WT α1B, whereas 4A α1B RA/KA showed much slower but similar inactivation to those of α1B and α1B RA/KA with β2a (*Figure 7B, C*). Additionally, the effects of $PIP_2$ depletion on current amplitude were

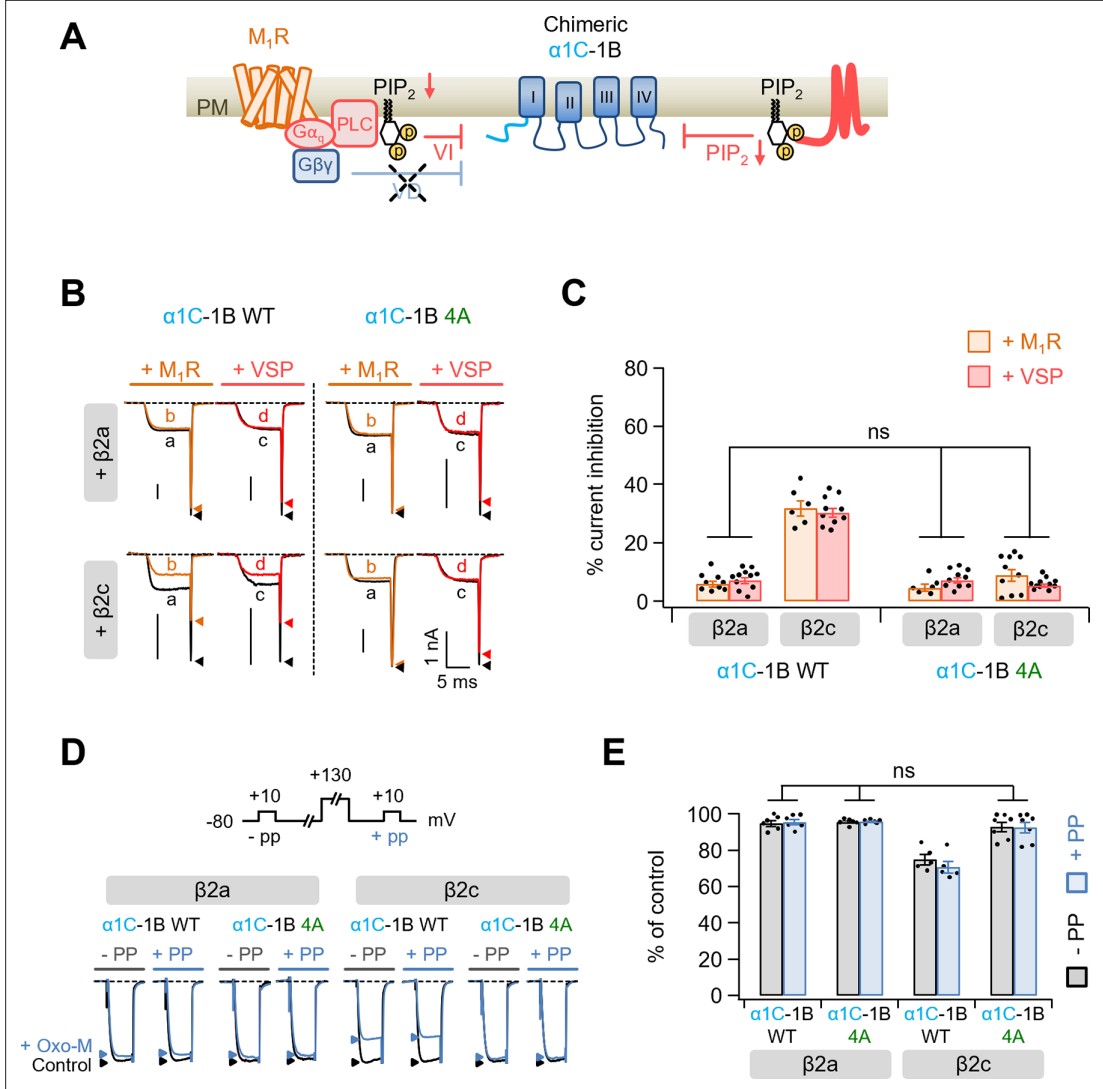

**Figure 6.** Modulation by M₁ muscarinic stimulation and Dr-VSP activation in Gβγ-insensitive chimeric α1C-1B Ca$_V$2.2 channel. (**A**) Schematic diagram showing the inhibitory signaling from M₁ muscarinic acetylcholine receptor (M₁R) and Dr-VSP to Gβγ-insensitive chimeric α1C-1B channel. VI, voltage-independent inhibition; VD, voltage-dependent inhibition. (**B**) Current traces before (a, black) and during (b, orange) the 10 µM Oxo-M application or before (c, black) and after (d, red) the Dr-VSP activation in cells expressing the α1C-1B WT and α1C-1B 4A with β2a or β2c subunits. Peak tail current is indicated by arrowheads (trace a, black head; trace b, orange head; trace c, black head; trace d, red head). (**C**) Summary of current inhibition (%) of α1C-1B WT and α1C-1B 4A by M₁R stimulation (orange bars) or Dr-VSP activation (red bars) in cells with β2a or β2c subunits (n = 6–11). (**D**) Current traces before (control, black) and during the Oxo-M application (+Oxo-M, blue) were superimposed. Cells were given a test pulse (−PP) and then depolarized to +130 mV, followed by the second test pulse after 20 ms (+PP). Peak current is indicated by arrowheads (control, black head; +Oxo-M, blue head). (**E**) Summary of the prepulse experiments in before and Oxo-M perfused cells with α1C-1B WT and α1C-1B 4A with β2a or β2c subunits (n = 5–7). The current amplitude after Oxo-M application is given as percentage of the initial control. Dots indicate the individual data points for each cell. Data are mean ± standard error of the mean (SEM).

The online version of this article includes the following source data and figure supplement(s) for figure 6:

**Source data 1.** Current inhibition (%) of α1C-1B WT and 4A mutants by M₁R or Dr-VSP activation in cells expressing with β2a and β2c.

**Figure supplement 1.** Polybasic motif at the C-terminal end of the I–II loop affects in determining the M₂ muscarinic modulation of Ca$_V$2.2 channels.

**Figure supplement 1—source data 1.** Current inhibition (%) of α1B WT and 4A mutants by M₂R activation in cells expressing with β2a and β2c.

measured in these mutant channels. Mutation of the two basic residues in S4$_{II}$ completely abolished the Dr-VSP-mediated current inhibition in cells expressing WT α1B RA/KA or 4A α1B RA/KA with the β2a subunit, while there was approximately 5% inhibition in cells expressing WT and 4A α1B with β2a (***Figure 7D, E***). Importantly, PIP₂ depletion significantly inhibited the currents in cells expressing WT

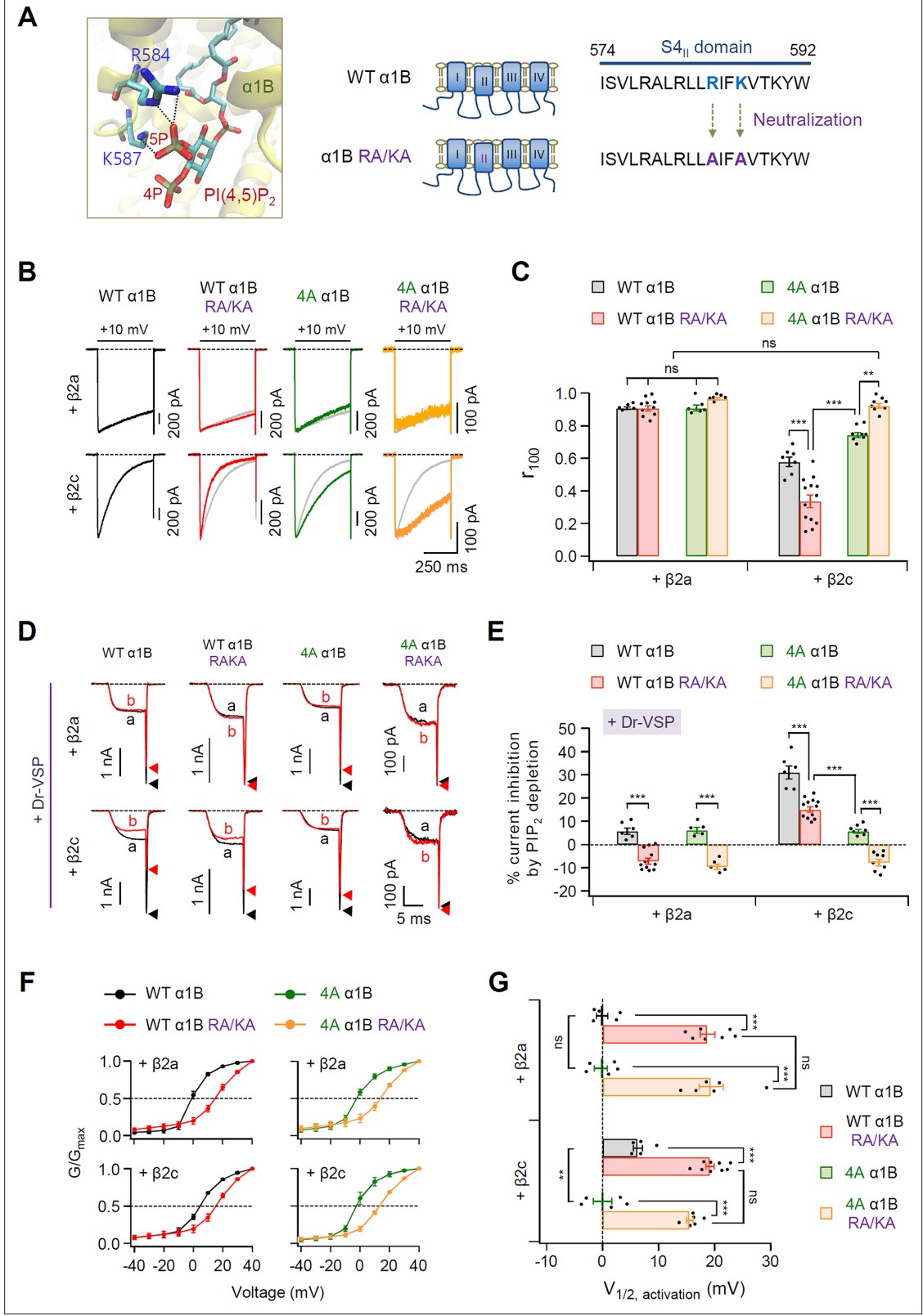

**Figure 7.** PIP$_2$-binding residues within the S4$_{II}$ domain plays an important role in determining steady-state activation and PIP$_2$ sensitivity of Ca$_V$2.2 channels. (**A**) Distance analysis of PIP$_2$-binding site in the S4$_{II}$ domain of α1B subunit. Two amino acids (R584 and K587) interacting with the 5-phosphate of PIP$_2$ were neutralized to alanine residues (RA/KA). (**B**) Current inactivation was measured during 500-ms test pulses to +10 mV in cells expressing WT α1B (black traces), WT α1B RA/KA (red traces), 4A α1B (green traces), and 4A α1B RA/KA (orange traces) with β2a (upper) or β2c (bottom). Gray traces

*Figure 7 continued*

present the curve of WT α1B for comparison. (**C**) Summary of current inactivation of $Ca_V2.2$ channel in cells expressing indicated α1B with β2a ($n$ = 5–10) or β2c ($n$ = 7–13). The $r_{100}$ indicates the fraction of current remaining after 100-ms depolarization to +10 mV. (**D**) Current inhibition by Dr-VSP-mediated $PIP_2$ depletion in cells expressing WT α1B, WT α1B RA/KA, 4A α1B, and 4A α1B RA/KA with β2a (upper) or β2c (bottom) subunits. $Ca_V2.2$ currents before (**a**) and after (**b**) the depolarizing pulse to +120 mV are superimposed in Dr-VSP-expressing cells. Peak tail current is indicated by arrowheads (trace a, black head; trace b, red head). (**E**) Summary of the $Ca_V2.2$ current inhibition (%) by $PIP_2$ depletion in cells expressing indicated α1B with β2a ($n$ = 6–12) or β2c ($n$ = 5–11). (**F**) The voltage dependence of normalized steady-state activation ($G/G_{max}$) for WT α1B (black), WT α1B RA/KA (red), 4A α1B (green), and 4A α1B RA/KA (orange) with β2a (left) or β2c (right). Tail currents elicited between −40 and +40 mV in 10 mV steps, from a holding potential of −80 mV were normalized to the largest tail current in each series of test pulse. The curves were fitted by a Boltzmann function. Dashed line indicates the $V_{1/2}$ of normalized steady-state activation. (**G**) Summary of the $V_{1/2}$ of normalized steady-state activation in F ($n$ = 5–9). Dots indicate the individual data points for each cell. Data are mean ± standard error of the mean (SEM). **$p < 0.01$, ***$P < 0.001$, using two-way analysis of variance (ANOVA) followed by Sidak post-hoc test.

The online version of this article includes the following source data and figure supplement(s) for figure 7:

**Source data 1.** Current inactivation ($r_{100}$), current inhibition (%) by $PIP_2$ depletion and the $V_{1/2}$ of normalized steady-state activation in cells expressing WT α1B, WT α1B RA/KA, 4A α1B, and 4A α1B RA/KA with β2a or β2c.

**Figure supplement 1.** $PIP_2$ sensitivity of α1B R578,581A with β2a or β2c subunits.

**Figure supplement 1—source data 1.** Current inhibition (%) by $PIP_2$ depletion in cells expressing WT α1B and α1B R578,581A with β2a or β2c.

α1B RA/KA and β2c, whereas 4A α1B RA/KA exhibited no current inhibition, like α1B RA/KA with β2a (*Figure 7D, E*). Since the mutation of two basic residues changes the gating charges of $S4_{II}$ voltage-sensor domain, we additionally tested if other charge residues within the $S4_{II}$ similarly affects $PIP_2$ sensitivity of $Ca_V2.2$ channels. We eliminated two adjacent arginine residues (R578 and R581) in $S4_{II}$ by replacing with alanine (α1B R578,581A) (*Figure 7—figure supplement 1A*). In both WT α1B and α1B R578,581A channels with β2a or β2c, there was no significant changes in the current inhibition by $PIP_2$ depletion (*Figure 7—figure supplement 1B, C*), suggesting that the R578 and R581 charge residues near the $PIP_2$-binding pocket were not involved in the $PIP_2$ interaction. Next, we examined the functional role of the $PIP_2$-binding site within $S4_{II}$ in the voltage-dependent activation of $Ca_V2.2$ channels. Regardless of β2 isotype, the activation curves were significantly shifted toward the depolarization direction in both WT α1B RA/KA and 4A α1B RA/KA (*Figure 7F, G*). Together, our results suggest that the two basic residues within the $S4_{II}$ domain consistently interact with $PIP_2$ regardless of the coupled β2 isotype. Additionally, $PIP_2$-binding to the $S4_{II}$-binding pocket is important in maintaining stable $Ca_V2.2$ channel gating.

## Discussion

This study has expanded our understanding of the inter-regulatory actions of the $Ca_V$ β subunit and membrane $PIP_2$ on $Ca_V$ channel gating properties, including inactivation kinetics, current density, and voltage dependency. Our data predict that $Ca_V2.2$ channels complexed with any β isotype can interact with membrane $PIP_2$ through the binding pocket in the $S4_{II}$ domain (*Figure 8*). However, in $Ca_V2.2$ channels with cytosolic β2c, there seems to be another interaction with $PIP_2$ through the nonspecific phospholipid-binding site at the distal end of the α1B I–II loop. This leads to the channel becoming highly sensitive to Dr-VSP-mediated $PIP_2$ depletion (*Figure 8*, lower panel). In channels with β2a, the membrane anchoring of the subunit may interfere with the interaction between the phospholipid-binding site and $PIP_2$. This converts the channels to a less $PIP_2$-sensitive state (*Figure 8*, upper panel). Additionally, the neutralization of polybasic residues in the I–II loop to alanine abolished $PIP_2$ binding on the phospholipid-binding site regardless of β isotype, which led to the less $PIP_2$-sensitive state (*Figure 8*, 4A α1B). By contrast, the neutralization of two basic residues in the $S4_{II}$-binding pocket slightly reduced $PIP_2$ sensitivity in channels with cytosolic β2c subunits and completely abolished the response in channels with a β2a subunit (*Figure 8*, α1B RA/KA). Taken together, these data showed that when $PIP_2$ molecules were depleted at the $VSD_{II}$ $PIP_2$ and polybasic phospholipid-binding sites or both sites were mutated to neutralized amino acid residues, the channels move to a nonconducting state (*Figure 8*, 4A α1B RA/KA).

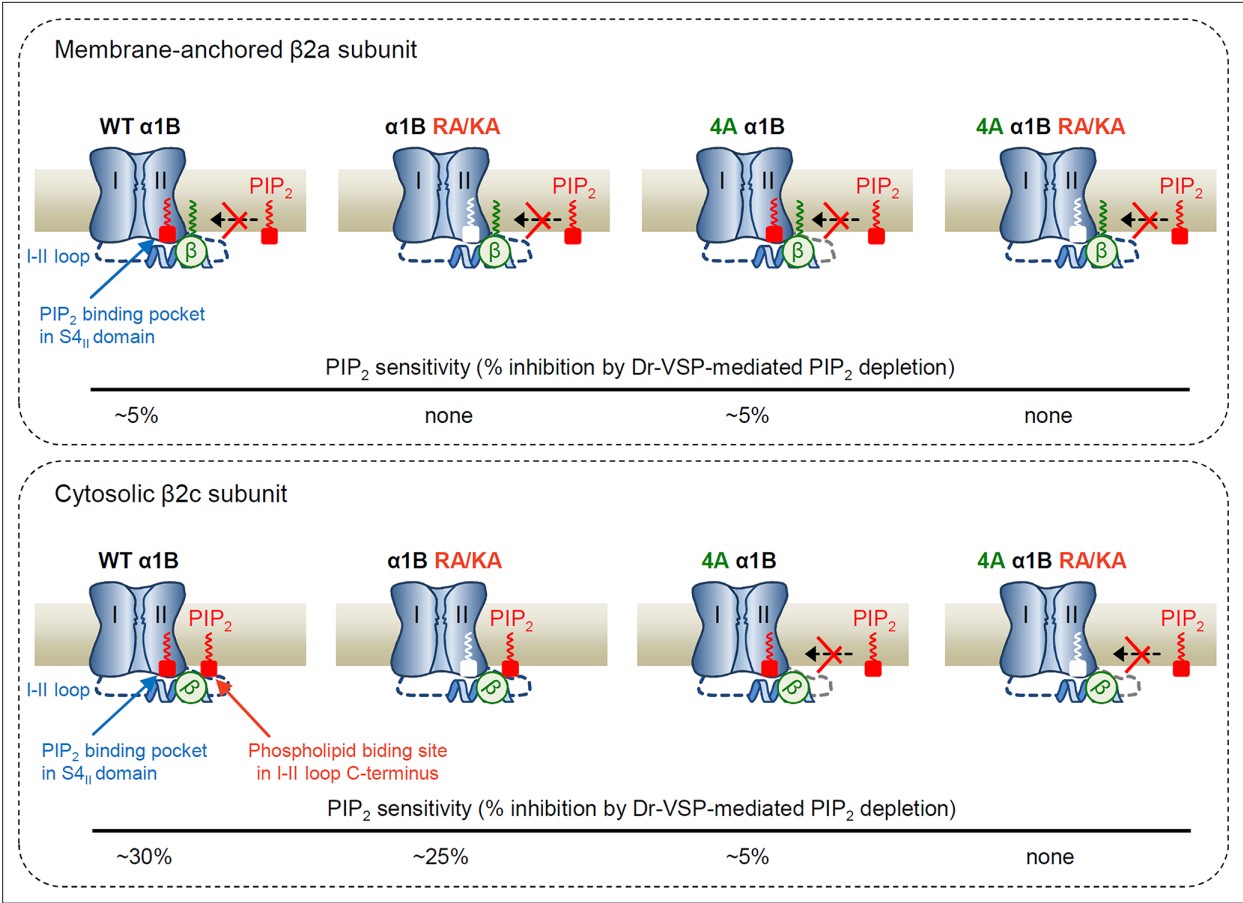

**Figure 8.** Schematic model showing the differential regulation of Ca$_V$2.2 channels with membrane-anchored and cytosolic β subunits by PIP$_2$. The channel possesses two distinct PIP$_2$-interacting sites: the PIP$_2$-binding pocket in the S4$_{II}$ domain and the nonspecific phospholipid-biding site in the I–II loop C-terminus. When the Ca$_V$2.2 channel is coupled with membrane-anchored β2a (upper panel), the proximal interaction of N-terminus of β2a with plasma membrane (PM) via its lipid anchor eliminates the binding of PIP$_2$ to the polybasic phospholipid site on I–II loop, leading to the state less sensitive to PIP$_2$ (upper left). In this condition, mutation of the PIP$_2$-interacting phospholipid site in the I–II loop does not change the PIP$_2$ sensitivity (upper 4A α1B). In contrast, when the Ca$_V$2.2 channel is coupled with cytosolic β2c (lower panel), there is no interaction of β subunit with the PM, leading to the higher PIP$_2$-sensitive state through the association of the polybasic phospholipid-binding site with acid phospholipids in the PM (lower left). In 4A mutant channels, PIP$_2$ interaction with the phospholipid-binding site is abolished, changing the channels to a state that shows only PIP$_2$ binding to the binding pocket in S4$_{II}$ domain. PIP$_2$ depletion in the PM or mutations of both PIP$_2$-interacting sites alter channels to the nonconducting state by shifting the voltage-dependent activation to the depolarization direction (lower right). The approximate PIP$_2$ sensitivity of each channel state in response to Dr-VSP activation is indicated as % inhibition at the bottom of each panel.

## Ca$_V$ β subunits regulate bidentate PIP$_2$ binding to Ca$_V$2.2 channels

Previous studies have proposed a bidentate model for the PIP$_2$ modulation of N-type Ca$_V$2.2 channel regulation (*Heneghan et al., 2009*; *Hille et al., 2015*; *Mitra-Ganguli et al., 2009*; *Roberts-Crowley and Rittenhouse, 2009*). In this model, lipidation on the N-terminus of Ca$_V$ β subunits disrupts the hydrophobic interaction between the two fatty acyl chains of PIP$_2$ and Ca$_V$2.2 channels, and thus reduces current inhibition by PIP$_2$ depletion. For example, β2a subunits interact with the PM through two palmitoyl fatty acyl chains in the N-terminus, leading to competition in binding to Ca$_V$ channels with the fatty side chains of PIP$_2$. This competition removes PIP$_2$ from the channel-binding site. Thus, Ca$_V$ channels with β2a are uncoupled from the membrane PIP$_2$ and show lower PIP$_2$ sensitivity to PIP$_2$ depletion. By contrast, non-lipidated β3 subunits expressed in the cytosol do not interrupt the interaction between the fatty acyl chains of PIP$_2$ and Ca$_V$2.2 channels, and show high PIP$_2$ sensitivity of channels (*Heneghan et al., 2009*; *Hille et al., 2015*; *Suh et al., 2012*). Consistently, we found that when the β3 subunits were anchored to the PM by adding the lipidation signal of Lyn to the N-terminus, the engineered Lyn-β3 construct decreased the PIP$_2$ sensitivity of Ca$_V$2.2 channels, similar to β2a. The Lyn$_{11}$ domain incorporates into the PM through the G2-myristoylated and C3-palmitoylated lipid anchors;

therefore, Lyn-β3 mimics β2a in competing with PIP$_2$ for the hydrophobic Ca$_V$2.2 channel interaction. Conversely, the lipid anchor of Lyn-48aa-β2c may be localized far from the channel complex because of its long N-terminal flexible linker, suggesting that these mutant subunits cannot disrupt the hydrophobic interaction between PIP$_2$ and channels.

Our results provide advance information about the bidentate model. First, we confirmed that two distinct PIP$_2$-interacting sites were preserved in the Ca$_V$2.2 channel: the binding pocket in VSD$_{II}$ and phospholipid-binding site in the I–II loop. Our data are consistent with that the 5-phosphate group of membrane PIP$_2$ interacts with the two basic residues within the S4$_{II}$ domain of Ca$_V$2.2 channels regardless of β2 isotype. The additional interaction of PIP$_2$ with the nonspecific phospholipid biding site in the distal I–II loop of Ca$_V$ channels was mainly observed in Ca$_V$2.2 channels with the cytosolic Ca$_V$ β2c subunit. Our data indicate that PIP$_2$-binding to the I–II loop phospholipid-binding site is selectively disrupted by the lipid anchor of membrane-anchored β2a. The hydrophobic interaction of the palmitoyl or myristoyl groups of Ca$_V$ β2a or Lyn-β constructs with channel complex may be the cause of PIP$_2$ release from the lower-affinity I–II loop phospholipid-binding site (*Roberts-Crowley and Rittenhouse, 2009*). When PIP$_2$ interacts with the VSD$_{II}$ PIP$_2$-binding site of Ca$_V$2.2 channels complexed with β2a, the PIP$_2$ sensitivity of the channels dramatically decreased to approximately 5%. Our results suggested that this minimal PIP$_2$ sensitivity specifically caused by PIP$_2$ degradation on VSD$_{II}$-binding pocket by Dr-VSP activation.

This work suggests that the PIP$_2$ sensitivity of the Ca$_V$2.2 channel is mainly affected by the length between the lipid anchor and GK domain of the Ca$_V$ β subunit. Although both Lyn-β2c and Lyn-48aa-β2c are localized at the PM, the PIP$_2$ sensitivity and inactivation kinetics of Ca$_V$2.2 channels are significantly different from each other: Ca$_V$2.2 channels with Lyn-β2c subunits exhibited relatively slower inactivation kinetics and lower PIP$_2$ sensitivity, similar to channels with the membrane-anchored β2a subunit. By contrast, Ca$_V$2.2 channels with Lyn-48aa-β2c subunits exhibited faster inactivation kinetics and higher PIP$_2$ sensitivity, similar to channels with the cytosolic β2c subunit. Similarly, disruption of the SH3-GK interaction in the membrane-anchored β2a subunit accelerated current inactivation and increased the current inhibition by PIP$_2$ depletion. Moreover, real-time translocation of the lipid anchor, Lyn$_{11}$, to the channel complex by rapamycin-inducible dimerization systems slowed the inactivation and decreased the PIP$_2$ sensitivity of Ca$_V$2.2 channels. Inversely, incremental increases in flexible linker length between the lipid anchor and GK domain of Ca$_V$ β2 subunits gradually accelerated the inactivation kinetics and increased the PIP$_2$ sensitivity of Ca$_V$2.2 channels. However, the mechanism by which the physical distance from the PM lipid to GK domain of the Ca$_V$ β subunit affects the PIP$_2$ sensitivity of the Ca$_V$2.2 channel is not fully understood yet. Another possibility is that torsional rigidity of the linker domain may be different depending on the length and thus differently restrict the cytoplasmic movement of Ca$_V$ β subunit as well as the gating of Ca$_V$2.2 channels.

Colecraft et al. have reported that chemically induced anchoring of intracellular loops of the channels to the PM can modulate the gating of the HVA Ca$^{2+}$ channel (*Subramanyam and Colecraft, 2015*; *Yang et al., 2013*). They have shown that PdBu-induced translocation to the PM of chimeric β3-C1$_{PKCγ}$, which is assembled by fusing the C1 domain of PKCγ to the C-terminus of the β3 subunit, leads to the inhibition of the Ca$_V$2.2 current. Conversely, the C1$_{PKCγ}$-β3 subunit, which is assembled by adding C1$_{PKCγ}$ to the N-terminus of the β3 subunit, has no effect on the current (*Yang et al., 2013*). These studies suggest that the polarity of the PM-targeting domain may play an important role in determining the Ca$_V$2.2 channel gating; however, the molecular basis of the differential regulation mechanism remain unclear. On the basis of our results, we speculate that the C1$_{PKCγ}$-β3 form may be insufficient to disrupt the interaction with between phospholipid-binding site and PIP$_2$ in Ca$_V$2.2 channels because the length from the C1$_{PKCγ}$ and the GK domain of the β3 subunit is 175 aa. This could be too long to interfere the interaction between PIP$_2$ and Ca$_V$2.2 channels.

Recently, *Gao et al., 2021* have shown that two basic gating charge residues (R584 and K587) within the S4$_{II}$ domain of human Ca$_V$2.2 channel interact with the 5-phosphate group of membrane PIP$_2$. In our present work, we found that mutation of the two residues (RA/KA) in the S4$_{II}$ domain completely blocked the Dr-VSP-induced current suppression in channels with β2a and shifted the voltage-dependent activation curve toward the depolarization direction regardless of Ca$_V$ β2 isotype. The cryo-EM structure does not show the nonspecific PIP$_2$-binding site in the channels probably because it is located in the flexible I–II loop. We hypothesize that the polybasic residues in the I–II loop tether to the anionic phospholipids through the electrostatic interaction and this dipole–dipole

interaction may contribute to the low-affinity phospholipid-binding site (*Yeon et al., 2018*). In contrast, the $VSD_{II}$ $PIP_2$-binding site forms a pocket-like structure inside the $S4_{II}$ domain and covered by the AID domain in the cytosolic side (*Dong et al., 2021*; *Gao et al., 2021*), which could stabilize the domain in a high-affinity $PIP_2$ interacting site. Thus, it is possible that the $PIP_2$ molecule inside the $VSD_{II}$ $PIP_2$-binding pocket is relatively less accessible to the degradation by phospholipase C or Dr-VSP, leading to the lower $PIP_2$ sensitivity in $Ca_V2.2$ channels.

In conclusion, our findings provide new insights on the regulatory mechanism of $Ca_V2.2$ channel gating by $Ca_V$ β subunits. Our recent study has reported that when intracellular $Ca^{2+}$ is increased by depolarizing the cells or activating $G_q$-coupled receptors, the high intracellular $Ca^{2+}$ concentration induces a dissociation of the N-terminus of the $Ca_V$ β2e subunit from the PM. This increases both the inactivation kinetics and $PIP_2$ sensitivity of $Ca_V2.2$ channels (*Kim et al., 2016*). The N-terminus of the β2e subunit is anchored to the PM *via* electrostatic interaction with the anionic phospholipids of these PM. These studies suggest that dissociation of the β2e subunit from the membrane leads to an interaction between the I–II loop phospholipid-binding site and $PIP_2$, which changes the gating properties of $Ca_V$ channels in physiological conditions. The interaction of $Ca_V$ α1B with β subunits can be dynamically exchanged by other free β isoforms in intact cells (*Yeon et al., 2018*); therefore, the displacement of cytosolic β subunits by membrane-tethered β subunits on $Ca_V$ channels will abolish the interaction with between $PIP_2$ and the I–II loop phospholipid-binding site via lipid anchor of membrane-tethered β subunits, which alters the $Ca_V$ channel gating properties. Further studies are needed to investigate whether the conformational shift of the I–II loop to the membrane or cytosolic face by endogenous β subunit combinations determines $Ca_V$ channel gating in neurons and other excitable cells.

# Materials and methods
## Cell culture and transfection
Human embryonic kidney tsA-201 cells (large T-antigen transformed HEK293 cells; RRID:CVCL_2737) were a kind gift from Dr Bertil Hille at University of Washington. The identity of this cell line has been authenticated by STR analysis and has recurrently tested negative for mycoplasma contamination using PCR (Cosmogenetech, Daejeon, South Korea). Cells were maintained in Dulbecco modified Eagle medium (Invitrogen, CA) supplemented with 10% fetal bovine serum (Invitrogen, CA) and 0.2% penicillin/streptomycin (Invitrogen, CA) in 100 mm culture dishes at 37°C with 5% $CO_2$. The cells were transiently transfected with Lipofectamine 2000 (Invitrogen, CA) when the confluency of the cells reached 50–70%. For assessment of $Ca_V$ channel expression, the cells were co-transfected with α1 of $Ca_V$, α2δ1, and various β2 chimera constructs in a 1:1:1 molar ratio. The transfected cells were plated onto a coverslip chip coated with poly-L-lysine (0.1 mg/ml, Sigma-Aldrich, MO) 24–36 hr after transfection. Plated cells were used for electrophysiological and confocal experiment within 24 hr after plating, as described previously (*Park et al., 2017*).

## Plasmids
The following plasmids were used: The calcium channel subunits α1B of rat $Ca_V2.2e$[37b] (GenBank Sequence accession number AF055477) and rat α2δ1 (AF286488) were from Diane Lipscombe, Brown University, Providence, RI. Chimeric α1C-1B was generously donated by David T. Yue, Johns Hopkins University, Baltimore, MD. Mouse cDNAs of β2a and β2c were generously donated by Veit Flockerzi, Saarland University, Homburg, Germany. The Dr-VSP (AB308476) was obtained from Yasushi Okamura, Osaka University, Osaka, Japan.

## Molecular cloning
Cloning of β2a-GFP, β2a(C3,4S)-GFP, and β2c-GFP was performed as previously described (*Park et al., 2017*). For the generation of various β2 chimera constructs, we used the one-step sequence- and ligation-independent cloning (SLIC) as a time-saving and cost-effective cloning strategy (*Jeong et al., 2012*). First, pEGFP-N1, pEYFP-N1, and mCherry-N1 vectors (Clontech) were linearized by KpnI restriction enzyme digestion. The cDNAs encoding β2a, β2c, Lyn, FRB, or FKBP were amplified by PCR using primers with an 18-bp homologous sequence attached to each end of the linearized vector. Primers used for β2 chimera constructs are listed in *Supplementary file 1*. Second, the linearized vector and PCR fragments were blended and incubated at room temperature for 2.5 min with T4

DNA polymerase (NEB, The Netherlands). Third, the DNA mixture was kept on ice for 10 min, after which competent *Escherichia coli* cells were transformed directly. For the deletion and point mutation of GK-SH3 interaction sites of the β2 subunit and the potential PIP$_2$-interaction sites of α1B, first, the α1B or β2 subunits were amplified by inverse PCR using nPfu-special DNA polymerase (Enzynomics, Daejeon, South Korea). Second, the PCR product was 5'-phosphorylated by T4 polynucleotide kinase (Enzynomics, Daejeon, South Korea) and plasmid DNA was digested by Dpn I (Agilent Technologies, Santa Clara, CA). Finally, the PCR product was ligated by T4 DNA ligase (NEB, The Netherlands). The primers used for mutagenesis are listed in *Supplementary file 2*. All the chimera and mutant constructs were verified by DNA sequencing (Macrogen, South Korea).

## Electrophysiology

The whole-cell configuration of the patch-clamp technique was used to record Ba$^{2+}$ currents using HEKA EPC-10 patch-clamp amplifier with pulse software (HEKA Elektronik). Electrodes pulled from glass micropipette capillaries (Sutter Instrument) had resistances of 2–4 MΩ. The whole-cell access resistance was of 2–6 MΩ, and series resistance errors were compensated by 60%. For all recordings, cells were maintained at −80 mV. The external solution contained 10 mM BaCl$_2$, 150 mM NaCl, 1 mM MgCl$_2$, 10 mM HEPES, and 8 mM glucose, adjusted to pH 7.4 with NaOH and an osmolarity of 321–350 mOsm. The internal solution of the pipette consisted of 175 mM CsCl, 5 mM MgCl$_2$, 5 mM HEPES, 0.1 mM 1,2-bis(2-aminophenocy)ethane *N,N,N',N'*-tetraacetic acid (BAPTA), 3 mM Na$_2$ATP, and 0.1 mM Na$_3$GTP, adjusted to pH 7.4 with CsOH and an osmolarity of 321–350 mOsm.

## Confocal imaging

All imaging examinations were performed with an LSM 700 confocal microscope (Carl Zeiss AG) at room temperature (22–25°C). The external solution for confocal imaging contained 160 mM NaCl, 2.5 mM KCl, 2 mM CaCl$_2$, 1 mM MgCl$_2$, 10 mM HEPES, and 8 mM glucose, adjusted to pH 7.4 with NaOH and an osmolarity of 321–350 mOsm. For live-cell imaging, images were obtained by scanning cells with a ×40 (water) apochromatic objective lens at 1024 × 1024 pixels using digital zoom. Analysis of line scanning of fluorescence images was performed using the 'profile' tool in Zen 2012 lite imaging software (Carl Zeiss Microimaging). To analyze colocalization, we performed quantitative colocalization analysis using Fiji software with the Colocalization Threshold plugin to determine the Pearson's correlation coefficient (*R*). Pixel intensities were presented as 2D intensity histograms with a linear regression line and as bar graphs with mean *R* values. All images were transferred from LSM4 to JPEG format.

## Förster resonance energy transfer

FRET experiments were performed using a monochromator (Polychrome V; TILL Photonics) with a ×40, NA 0.95 dry immersion objective lens (Olympus). Regular pulses of indigo light (438 ± 12 nm) excited the fluorescent proteins. Emission was separated into short (460–500 nm) and long (520–550 nm) wavelengths by appropriate filters and then acquired by two photomultipliers. Donor and acceptor signals obtained by photometry (TILL Photonics) were transferred to the data acquisition board (PCI-6221; National Instruments). Signal acquisition and real-time calculation of the FRET ratio were conducted by a custom program. The FRET ratio was calculated as follows:

$$\text{FRETr} = \left(\text{YFP}_\text{C} - \text{cFactor} \times \text{CFP}_\text{C}\right)/\text{CFP}_\text{C}$$

CFP$_\text{C}$ is the CFP emission detected by the short-wavelength photomultiplier, and YFP$_\text{C}$ is the YFP emission detected by the long-wavelength photomultiplier, as described previously (*Keum et al., 2014*).

## Calculation of distance with a WLC model

The Lyn-Linker-(additional Link) structure was suggested as an unstructured structure from the IUPRed Web-server (http://iupred.elte.hu/) (*Dosztányi et al., 2005*) to predict disorder tendency. To calculate the distance between the GK domain and the inner surface of the PM, the WLC model was used. This model is usually used to describe the behavior of polymers that are semi-flexible: quite stiff with successive segments pointing in roughly the same direction, and with persistence length within a few orders of magnitude of the polymer length. This model is also used to describe unstructured proteins

like this linker structure (*Zhou, 2001*). In the WLC, the mean square end-to-end distance $<R^2>$ is written as:

$$<R^2> = 2PL_0 \left[ 1 - \frac{P}{L_0} \exp \left( -\frac{L_0}{P} \right) \right]$$

where $P$ is the polymer's characteristic persistence length and  is the maximum length. We used $P$ = 0.6 and  as (N − 1)*3.8, where $N$ is number of amino acids in the unstructured protein (*Lapidus et al., 2002*). We then removed three amino acids in Lyn(MGC), which is directly connected to the membrane via palmitoylation and myristoylation. The root mean square end-to-end distance $\sqrt{<R^2>}$, which can be suggested as the average distance, was calculated. $\sqrt{<R^2>}$ was 32.7 Å for six additional linkers, 36.0 Å for 11 aa, 38.4 Å for 15 aa, 42.4 Å for 22 aa, 52.5 Å for 43 aa, and 28.2 Å for no additional linker.

## Statistical analysis

Patch clamp data acquisition and analysis used Pulse/Pulse Fit 8.11 software with the EPC-10 patch clamp amplifier (HEKA Elektronik). Further data processing was performed with Igor Pro 6.2 (Wave-Metrics, Inc), Excel office 365 (Microsoft), and GraphPad Prism 7.0 (GraphPad Software, Inc). All quantitative data were presented as mean ± standard error of the mean values. Comparisons between groups were analyzed by Student's two-tailed unpaired $t$-test. Comparisons among more than two groups were analyzed using one-way analysis of variance (ANOVA) followed by Tukey post hoc test. Comparisons among more than two groups with two independent variables were analyzed using two-way ANOVA followed by Sidak post hoc test. Differences were considered significant at the *$p < 0.05$, **$p < 0.01$, and ***$p < 0.001$, as appropriate.

## Acknowledgements

We thank many laboratories for providing the plasmids.

## Additional information

### Funding

| Funder | Grant reference number | Author |
| --- | --- | --- |
| National Research Foundation of Korea | 2020R1A6A3A01100500 | Cheon-Gyu Park |
| National Research Foundation of Korea | 2019R1A2B5B01070546 | Byung-Chang Suh |
| National Research Foundation of Korea | 2020R1A4A1019436 | Byung-Chang Suh |

The funders had no role in study design, data collection, and interpretation, or the decision to submit the work for publication.

### Author contributions

Cheon-Gyu Park, Conceptualization, Data curation, Formal analysis, Funding acquisition, Investigation, Visualization, Methodology, Writing - original draft; Wookyung Yu, Formal analysis; Byung-Chang Suh, Conceptualization, Resources, Supervision, Visualization, Project administration, Writing - review and editing

### Author ORCIDs

Cheon-Gyu Park http://orcid.org/0000-0002-4739-1913
Wookyung Yu http://orcid.org/0000-0001-9835-930X
Byung-Chang Suh http://orcid.org/0000-0003-0278-2459

### Decision letter and Author response

Decision letter https://doi.org/10.7554/eLife.69500.sa1
Author response https://doi.org/10.7554/eLife.69500.sa2

## Additional files

### Supplementary files
- Supplementary file 1. Primers for β2 chimera constructs.
- Supplementary file 2. Primers for deletion or mutagenesis of Ca$_V$ α1B and β2 constructs.
- Transparent reporting form

### Data availability
All data generated or analyzed during this study are included in the manuscript and supporting files.

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
