## [Editor Report]

This manuscript describes experiments using heterologous expression to achieve molecular dissection of the effects of PIP2 and CaVβ2 auxiliary subunits on CaV2.1 (P/Q-type) calcium channels. The experiments also probe interplay between lipid effects and other modulatory pathways. Understanding the functional regulation of this channel is important because CaV2.1 channels play significant roles in neuronal plasticity.

---

## [Decision Letter]

**Decision letter after peer review:**

Thank you for submitting your article "Molecular basis of CaV β-and PIP2-dependent regulation of the CaV2.2 voltage-gated Ca 2+ channel" for consideration by *eLife*. Your article has been reviewed by 3 peer reviewers, and the evaluation has been overseen by a Reviewing Editor and Richard Aldrich as the Senior Editor. The following individuals involved in review of your submission have agreed to reveal their identity: Han Chow Chua (Reviewer #1); Ann Rittenhouse (Reviewer #3).

The work described in the manuscript provides new information about the roles of PIP2/phospholipids and the β subunit in regulating the gating of the calcium channel CaV2.2 and is likely to be of broad interest because CaV2.2 is an important pharmacological target. In particular, the work identifies a previously unknown site in the CaV2.2 loop at which phospholipids regulate channel gating. However, there was consensus among the reviewers that important conclusions of the manuscript are not well supported by the experimental results.

In your consideration of a revised version, please take note of the following.

1. The title or abstract should explicitly state that the experiments were conducted on transfected tsA201 cells.

2. The Discussion needs to be better organized and more succinct. The Reviewers have made specific suggestions in that regard.

3. Because GTP in the pipette should have promoted G-protein mediated stabilization of the reluctant state, the prepulse depolarization used to activate DR-VSP could have caused the G-protein to dissociate and relieved this tonic inhibition. Thus, the authors need to provide more methodological detail about the prepulse application and describe how the effects of activation of DR-VSP could be experimentally distinguished from G-protein dissociation. Otherwise, the possible effect of G-protein dissociation complicates the interpretation of the experiments with DR-VSP.

4. It should also be acknowledged that DR-VSP may not have equivalent access to the various pools of PIP2 that are important for regulating channel gating.

5. Features of the model presented by the authors to explain PIP2 actions on channel gating are not constrained by the experimental results and thus too speculative. These include statements about distance of the β subunit from the membrane and movement of the I-II loop, statements about the occupancy of the R and S states in relationship to the experimental results, and statements about PIP2 binding rather than phospholipid binding more generally.

*Reviewer #1 (Recommendations for the authors):*

In this manuscript, Park et al. investigated the mechanism of CaV β and PIP2 regulation of the voltage-gated calcium channel CaV2.2. Using electrophysiological and fluorescent techniques, combined with protein engineering, the authors presented evidence to show that the membrane-anchoring or cytosolic localisation of a CaV β subunit induces distinct conformational changes in the I-II loop of CaV2.2, which may explain the differential modulation of CaV2.2 inactivation kinetics and PIP2 sensitivity when co-expressed with different CaV β subunits. The authors also showed the mutation of a putative PIP2-interacting site on the I-II loop only affected the modulation by cytosolic, but not membrane-anchored CaV β subunit, leading them to conclude that the subcellular localisations of CaV β subunits also determine the coupling of the channel to PIP2. The authors performed extensive, careful experiments to support their claims, and the data were analysed appropriately. However, some aspects of their interpretation or hypotheses need to be clarified, and the writing of the manuscript needs improvement.

Based on the characterisation of the quadruple Ala mutant (4A α1B) alone, the authors made several claims including (1) CaV β subunit controls PIP2 interaction with CaV2.2, and in turn, determines the conformational state of the channel, and (2) there is an additional PIP2 interaction site on the channel that mediates the remaining 5 % current inhibition by PIP2 depletion. These claims are not convincing as the authors assumed that Ala substitution of the four Arg residues is sufficient to abolish PIP2 interaction in the region. It is entirely conceivable that there are additional residues in the I-II loop that are able to interact with PIP2 (e.g., K467). As such, it is difficult to agree with the speculations of the authors in the absence of stronger evidence. The characterisation of different mutants (e.g., Arg-to-Gln, Arg-to-Glu or Arg-to-Lys), or evidence demonstrating that the I-II loop of the 4A α1B mutant is incapable of membrane binding when co-expressed with cytosolic CaV β subunits (e.g., showing isolated WT I-ll loop being membrane-anchored in the presence of β2c, whereas the charge-neutralised version is found in the cytoplasm) may help strengthen the authors' argument.

The authors repeatedly state that the subcellular localisation of CaV β subunits is not as important a factor in determining the differential effects these subunits exert on the inactivation kinetics and PIP2 sensitivity of CaV2.2 as their distances from the PM or the conformational changes of the I-II loop of CaV2.2 (pp.6, 13 and 14). This interpretation is puzzling as the three factors are interrelated, but the authors appear to treat them as separate entities. The observation that the artificial linker length has a direct correlation with the modulatory effects of a cytosolic CaV β subunit when it is membrane anchored (e.g., Lyn-β2c vs. Lyn-48aa-β2c) does not mean that subcellular localisation of the auxiliary subunit is not important. A longer linker length simply offers flexibility for the membrane-anchored β2c subunit to induce similar conformational change in the I-II loop the same way as the cytosolic WT β2c subunit does. In fact, the extensive work performed on CaV β constructs bearing different linker lengths just further supports the established fact that the subcellular localisations of these auxiliary subunits are crucial for their distinct modulatory effects.

While the figures of this manuscript are generally clear and relatively easy to follow, the writing needs to be more succinct to convey a clear message to the readers. This issue is particularly noticeable in the superfluous Discussion section, which not only suffers from lengthy description of previous findings, but also lacks clear organisation of ideas. The use of subheadings may help compose a more coherent, easy-to-follow story.

*Reviewer #2 (Recommendations for the authors):*

Park and colleagues present some very interesting data that adds to our understanding of the mechanism by which G protein coupled receptors inhibit voltage-gated calcium ion channels of the CaV2 family. In particular they study CaV2.2, a channel that regulates calcium entry and transmitter release from neurons including sympathetic neurons. This G protein signaling pathway is of major importance and thus the work should be of interest to neuroscientists broadly. PIP2 is a major modulator of ion channel activity controlled by G protein activation, and the authors explore the role of calcium channel β subunits on PIP2 mediated effects on CaV2.2.

The authors propose a model to explain how calcium channel β subunits (which associated with the main α 1 subunits that encode CaV2 channels) control both PIP2 depletion mediated channel inhibition and channel inactivation kinetics. They propose a model in which the I-II loop region of the CaV2.2 channel (known to be the main binding site of β subunits) moves between two conformations that represent previously proposed willing and reluctant modes. These modes correspond to channels with low and high sensitivity to PIP2 inhibition. In this study the authors explore the impact of altering the relative proximity of the β subunit to the plasma membrane and presumably the I-II loop of CaV2.2. Critically, different β subunits differ in being anchored (β 2a) or not (β 2c) to the plasma membrane. By generating various chimeric β subunit proteins, the authors propose is that beta2a moves the I-II linker, destabilizing PIP2 interaction with the regulatory domain of the CaV2.2 channel and stabilizing the channel in a willing state.

Strengths

The experiments presented are strong and the data show convincingly by a number of different approaches the critical importance of membrane anchoring in determining the influence of β subunits on the state of the CaV2.2 channel. In particular, β subunit influences on the propensity of the CaV2.2 channel to inactivate, and its sensitivity to PIP2 depletion-mediated inhibition. The authors also show very nicely a graded impact of the β subunit depending on the length of linker separating the plasma membrane anchor motifs and the β subunit GK domain (e.g, Fig, 4 show a strong correlation), and they also demonstrate the importance of the poly-basic motif in the I-II loop of CaV2.2 in mediating the effects of for example β 2c (Figure 5).

The use of the voltage-sensing lipid phosphatase to deplete PIP2 in the plasma membrane using readily controlled voltage is very powerful and the authors use it here to great effect.

Weaknesses

The conclusions veer too far from the data presented and the approaches used. This detracts from the strong and provides interesting insights into an important signaling pathway.

For example, referring to "distance" from the plasma membrane with regard to the position of the β subunit, but distance is not being measured. The proposed states of the channel that involve movement of the I-II loop is possible but not directly addressed experimentally.

The authors might consider the following comments/suggestions to help clarify and strengthen the manuscript.

– The introduction could be strengthened by starting with a statement about the question that is being addressed. As it is now, the introduction is used to introduce each key component of the a1, β complex and PIP modulation but a clearer direct leading paragraph would help provide context and rational for the background information.

– Similarly, the Results section would benefit from a leading sentence or two about the logic of the experiments and their connection to the proposed model.

– The authors refer to "distance" from the plasma membrane to the Cavb subunit (e.g. first subheading of the results and throughout) which implies a measurement which is not the case. Why not say that the linker disrupts the effect of PM anchoring on CaV channel inactivation and inhibition by PIP2 depletion? The intermediate linkers 12, 24, 36 aa do not show progressive changes in current inhibition (following PIP2 depletion; supplemental figure 1-3) but do show a transition between the 24aa and 36aa linkers. Thus, there isn't a clear progressive correlation between length/distance and inhibition.

– Define "Lyn" when first appears.

– "However, the actions of the chimeric Lyn-48aa-β2c in current regulation were similar to those of the channels with the cytosolic β2c subunit, although the chimeric analogue was targeted to the PM." This is not clear, be more explicit. It's an important result.

– "…these data suggest that the distance from the PM to the CaV β subunit may be more important than PM-anchoring of CaV β in determining the inactivation kinetics and PIP2 sensitivity of CaV2.2 channels." This statement is not clear. The linker disrupts the effect of membrane anchoring.

– Figure 2. The authors generate alanine substituted CaVb subunits in two key regions shown to be required to interact with CaVa1 subunit – SH3 and GK domains. In the untethered version of CaVb2, there is strong inactivation and PIP2 depletion inhibition consistent with data shown in Figure 1. Similarly, membrane targeting of b2 changes the phenotype to slow inactivation and weak inhibition by PIP2 depletion, but the mutant version of the membrane targeted b2 subunit does not exhibit the WT phenotype suggesting that the SH3 and or GK domains are required to prevent PIP2 depletion mediated inhibition. The authors suggest that this is evidence that there is a conformational shift in the I-II linker of Cava1 subunit to decrease both the inactivation rate and PIP2 sensitivity of CaV2.2 channels. This is an attractive model but they do not directly test this hypothesis and all manipulations are of the β subunit.

– "In CaV2.2 channels with Lyn-FRB-HOOK-GK-Linker-FKBP (RCF), where a 194-aa linker was inserted between GK and FKBP, however, rapamycin enhanced the FRETr signal only without causing significant changes in the current amplitude (Figure 3B, right and Figure 3—figure supplement 1)." Rapamycin did affect current amplitude and PIP2 inhibition significantly according to the data shown in figure C-F (right). As above, it's not clear how these experiments "confirm that the inactivation kinetics and PIP2 sensitivity of CaV2.2 channels are mainly determined by a β subunit-dependent conformational shift of the I-II helix."

– "The effects of rapamycin on inactivation kinetics and PIP2 sensitivity were much weaker in CaV2.2 channels with RCF in comparison to those in channels with RF (Figure 3C-F), suggesting that rapamycin-induced dimerization was not sufficient to shift the I-II helix to the PM in channels with RCF". As per above, this is one possibility but there are many others. The data are not sufficient to make this claim. Furthermore, as noted above, there are measurable differences induced by rapamycin with the RCF construct.

– Is Fig, 3-supplemental 1 mislabeled? In A the largest currents are labelled RCF in panel A but labelled RF in panel B?

– The experiments in Figure 4 show a strong correlation between linker length and faster inactivation- higher PIP2 sensitivity. However, it is also clear from the recordings that there are effects on the overall rate of activation. This should be mentioned in the Results section.

– The experiments using muscarinic receptor are an important addition but they raise a general question. Why is there strong inhibition by M1 activation in the presence of b2a, and in the presence of b2c with the alanine substituted CaVa1 subunit, when under these conditions, there is very little or no inhibition in response to PIP2 depletion? What component of the receptor-mediated inhibition of the calcium current depends on PIP2 depletion?

– Figure 5F-G. Shows a comparison of Qon and tails currents measured by stepping to the reversal protntal to isolated Qon and then stepping back to a negative membrane potential (probably -80 mV) to measure the instantaneous current. They show that the ratio of Itail to Qon is reduced in WT CaV channels in the presence of b2c relative to b2a and b2c in the presence of the CaValpha alanine mutant. They suggest that this provides evidence that neutralizing the basic amino acids in CaV α within the I-II influences the channel open probability. They conclude that b2a supports channels with higher Po compared to b2c but there is a simpler explanation. Cav channels in the presence of b2c inactivate compared to those expressed with b2a. The authors use a 100 ms depolarization to drive activation which is long enough for CaV channels to inactivate, especially given that the deodorization is to +60 mV. The authors could generate scatter plots comparing Qon vs Itail and compared the slopes and/or use much shorter step depolarizations to limit the degree of inactivation.

*Reviewer #3 (Recommendations for the authors):*

Ion channels interact with transmembrane lipids for proper functioning. The voltage-gated ca^2+^ channel pore subunit CaV2.2 appears to have two phospholipid binding sites that regulate channel gating. These two sites were first characterized in CaV2.1 by Wu et al. (2002) and appear similar in CaV2.2. Phospholipid binding to an "S" site stabilizes channels so that they open upon moderately positive voltage steps. Binding to an "R" site results in reluctant gating, which requires very positive voltage steps to open the channels due to a rightward shift in the voltage-activation curve. Loss of PIP2 binding to the R site converts channels to "willing" gating where channel activation shifts to more negative voltages. The reluctant and willing terminology, first applied by Bean (1989) to describe voltage-dependent inhibitory actions of Go/iPCRs, continues in use to describe the actions of phospholipids on CaV2.2 gating. Which phospholipids bind to the two sites is unclear though the working assumption is that both sites bind PIP2 but additional phospholipids may also be capable of binding to the sites. Ca^2+^, Na+, divalent ion chelators, G-proteins as well as other proteins have complicated interactions that affect the willing and reluctant gating patterns. Despite these observations, the molecular mechanisms underlying CaV2.2's complicated gating behavior by phospholipids remains unclear. Physiologically, modulation of CaV2.2 gating is critical for CNS, peripheral autonomic and sensory functioning. These channels are a major therapeutic site for managing neuropathic pain. Thus, having a clearer understanding of what mechanisms regulate gating should provide new strategies for controlling pain and possibly other diseases.

In the current study, Park et al. have carried out a biophysical study that interrogates the role of different CaVbeta subunits in translocating an α-helix region, located near the C-terminus end of the intracellular loop between domains I and II of CaV2.2, to the inner plasma membrane. There, multiple basic residues of the α helix appear to face and interact with membrane phospholipids. The authors use mutagenesis of the polybasic segment along with imaging and biophysical strategies to manipulate channel conformations to test the importance of the site. Use of the DR-VSP and Rapamycin techniques continue to probe lipid binding to channels and CaV2.2 and beta2 subunit interactions, respectively, have yielded important findings. There is a large amount of data to think through; overall the quality of the data is excellent. Numerous diagrams help to explain various experiments. The authors make a strong case that a PIP2 binds to the polybasic segment of the α-helix. Their findings are complimentary to Kaur et al. (2015) who identified a homologous polybasic α-helix in CaV1.2 as having a phospholipid binding site. This shared polybasic sequence of CaV1 and Cav2 subunits identifies this segment as a critical phospholipid binding site for proper channel gating.

Park et al. go on to probe how the plasma membrane anchoring properties of CaVβ2 subunits regulate PIP2 binding to the polybasic α-helix region of CaV2.2. They created CaVbeta2c subunit constructs that are attached to a membrane targeting lyn11 sequence containing an amino acid peptide of varying lengths. They then tested whether the distance (or closeness) a CaVβ2 subunit is from the inner plasma membrane will determine whether PIP2 is able to bind to CaV2.2's polybasic α-helix. They added an mCherry tag to the Lynn11 tether and a GFP to the N-terminus of β subunit constructs to observe their cellular vs membrane association. They found that independent of tether length, all beta2c subunits and Lyn11 tether localize to the plasma membrane. However only beta2-subunits with a short tether result in CaV2.2 currents resistant to PIP2 depletion. They contend that changes rapid current kinetics occur when β 2c is not tightly tethered to the membrane.

They corroborate this by coexpressing Dr-VSP, a voltage sensitive 5 π phosphatase, which removes the 5 phosphate group from the inositol ring, and testing for PIP2 sensitivity. The authors propose CaVbeta2c subunits with a short tether hold the α-helix so tightly against the inner membrane that no PIP2 can bind and consequently, no PIP2 breakdown by the voltage-activated π 5-phosphatase is observed. It is also possible that the α-helix is pushed against the membrane such that PIP2 is protected from metabolism. Alternatively, the lipid tail of the Lyn11 construct associates with a hydrophobic transmembrane segment of CaV2.2 in a similar manner as the 2 palmitoylated tails of CaVbeta2a compete with PIP2 for a transmembrane binding site. When the tether becomes too long, the CaVbeta2c protein cannot dock the tail of the Lyn11 tether to the PIP2 binding site efficiently just as is shown in Figure 1A. This would explain why the imaging data (Figure 1) shows both Lyn-48aa-tether (labeled red) and beta2-subunit (labeled green) co-localizing to the membrane, yet unable to confer resistance to +Dr-VSP. This model also is consistent with the experimental results from Figure 2 showing that when the interactions of the SH + GK domains of a CaVbeta2 subunit are disrupted, the CaVbeta2 loses its ability to dock the Lyn11 fatty acid tail in the correct position (whether within the channel complex or outside the channel) to block PIP2 binding. An additional alternative is that the fatty acid tail of the Lyn11 construct does not compete with PIP2 binding but rather binds to the channel elsewhere causing conformational changes that confer resistance to PIP2 breakdown. Thus, while the data are convincing that a phospholipid, probably PIP2, binds the polybasic segment of the α-helix, additional studies are needed in order to determine the exact mechanism of action by which CaV2.2, coexpressed with CaVbeta2 subunits, become independent of PIP2.

In identifying a phospholipid binding site in the polybasic segment within the α-helix of the I-II loop, the authors conclude that loss of PIP2 binding at this site favors willing over reluctant gating described by Wu et al. (2002) for PIP2. The kinetic analyses are challenging to interpret since lipid-mediated gating changes remain insufficiently characterized (despite their work) for Park et al. to assign "R" and "S" binding sites based on gating. Also complicating the analysis is that Go/iPCR-mediated current inhibition appears to interact with the actions of PIP2 (see Rousset et al., 2004). A well-documented Gbeta/γ binding site overlaps with the β-subunit AID site on the I-II linker. Park et al. have included GTP in their pipette and therefore their recording conditions are predicted to have both phospholipid and tonic G protein subunit interactions with the I-II linker and potentially with one another. Lastly, both proteins bind just distal to a rigid α-helix that begins as the I-II linker leaves domain I (see Vitko et al., 2008). Thus, both C- and N-terminus α-helical regions of the I-II loop may influence channel interactions with PIP2 and vice versa; the rigid α-helices may confer changes in the binding sites for Gbeta/γ and CaVbeta subunits following PIP2 binding.

Thus, though the authors assign this site as the R site, the findings here are just a first step-though an important one-in characterizing the functional consequences of phospholipid binding to the polybasic α-helix. Determining the mechanism of action and whether this site is the R and/or S site will require additional experiments to bring clarity to the model. Nevertheless, these results identify the polybasic segment of the α-helix of the I-II loop as a critical phospholipid binding site regulating CaV2.2's function and thus may be a future therapeutic site for regulating activity. There is still much to understand about lipid interactions with ca^2+^ channel subunits and these findings are a major contribution to identifying one phospholipid binding site important for proper channel functioning.

1) Introduction. The Introduction does not give credit for "first findings" from other labs, which inadvertently gives misdirected credit for earlier findings.

2) Issue of R and S gating: Figure 1. The imaging data nicely show that the tethered β-subunits do indeed associate with the plasma membrane. However, a major issue with the current tracings is that the working definitions of R and S putative PIP2 binding sites and their relationship to consequential gating patterns is not well defined in the literature. In Figure 1D, the red traces do not appear to show kinetic changes normal associated with R S transitions. The +DVSP experiment should metabolize PIP2 at both the R and S sites. However, no change in either current amplitude or kinetics is observed with palmitoylated beta2a, which is thought to block at least one PIP2 binding site but leave the other site available for PIP2 binding. The authors need to provide an explanation for this if they are going to pursue the R and S model of PIP2's actions.

3) GTP in the pipette. GTP was included in the pipette solution, which should tonically promote G-protein-mediated reluctant gating. However, the prepulse given in the DR-VSP experiment (Figure 1D) should knock off any tonic G-protein interaction with CaV2.2 converting them to willing gating. (The authors need to state how long the delay is between the prepulse to 120 mV and the following current recorded at +10 mV.) Consequently, it's possible that all the +Dr-VSP experiments are testing changes in PIP2 binding to the S site because the prepulse will overcome any PIP2 effects at the R site as Wu et al. 2002 has documented. The authors need to address the confounding issue of GTP in the pipette and/or provide data that could rule out some of these confounding issues. The authors should repeat the expt in Figure 1D with GDP-β-S in the pipette. This critical control should reveal if there is a G-protein effect contaminating the data. In revisiting this experiment, the authors should also plot G-V curves to determine whether a shift in activation occurs following prepulses when GDP-β-S is included in the pipette.

Surprisingly, no prepulse facilitation occurs for beta2a (Figure 1-suppl 2A); previously the authors did observe facilitation with a prepulse using the same concentration of GTP in the pipette as used in the current study (Keum et al. 2014; Figure 2+3). The authors need to address this inconsistency with their previous data.

4) Experimental design for testing M1Rs. Figure 5 suppl2. In a number of the experiments, the currents are sampled 10 sec after Oxo-M application. Previously, the authors (Keum et al. 2014) showed that at 10 sec, inhibition could be overcome by prepulses, suggesting that Gbeta/γ was binding to the channels. The authors need to include traces from the time point when the current amplitude was measured. This experiment should be repeated with prepulses to determine how much of the inhibition is voltage-independent.

Lastly, the authors should show the current-voltage relationships of control vs Oxo-M for the 4 conditions. This will also reveal the consequences of mutating the polybasic residues on the voltage-sensitivity of the channel gating.

Also, in Figure 5 suppl2 Oxo-M inhibits currents of beta2a-WT and beta2a-4A-mutant channels. From their paper Keum et al. (2014) in Figure 2 using the same conditions of collecting traces 10 sec after Oxo-M application, they found remarkable slowed activation kinetics as if they are looking at Gbeta/gaamma mediated inhibition. They found also that VSP does not decrease the current, suggesting that the phospholipid metabolized by Oxo-M stimulated signaling must somehow be inaccessible to the 5-phosphatase or it is not PIP2. In this manuscript to be consistent, the authors should include individual traces from the time courses shown in Figure 5-suppl 2.

5) Data Interpretation. The interpretation of the data seem premature, in particular insufficient data are presented to assign locations for R vs S phospholipid binding sites. It is not obvious that all the experiments are looking at the same PIP2 binding site. For example, the authors contend that neutralizing 4 of the polybasic residues in the I-II loop removes a PIP2 binding site to stabilize CaV2.2 in a willing state. The authors also document the role of SH3-GK interaction and of lipid tethers of CaVbeta2 subunits in promoting lower sensitivity to PIP2 at the putative R site. There are additional basic residues in the α-helix that potentially participate in the binding of a second phospholipid. There are also polybasic regions in the CaVbeta2 N-terminus and HOOK regions that may be disturbed by mutating the 4 basic residues. The authors are best served by focusing on making the case for a phospholipid binding to the polybasic region critical for channel gating.

Page 10-line 1. The authors suggest that the membrane-anchored β-subunits somehow disrupt phospholipid binding to the polybasic binding motif in the α-helix of CaV2.2's I-II loop. They do not entertain the possibility that in certain subunit configurations, the α-helix may bind PIP2 (or other phospholipids) protecting it from metabolism- in other words unavailable to the voltage-activated membrane associated 5 π phosphatase (Dr-VSP). Lastly, in the discussion the authors suggest that the fatty acid tails of the lipidated Lyn11 constructs may insert into the membrane to compete with PIP2 binding to CaV2.2, similar to the proposed actions of palmitoylated beta2a (Heneghan et al., 2009; Suh et al., 2012). The authors should discuss whether the tether data fit this model. Lastly, the N-terminus region of the I-II loop forms a rigid α-helix which greatly influences channel activity (Vitko et al., 2008). The authors need to include these possibilities in the discussion.

Specific Comments

The authors repeatedly claim PIP2 binding when they have shown that a phospholipid is involved, but not specifically PIP2. In this paper the α-helix phospholipid binding site may indeed be selective for PIP2, however, Kaur et al. (2015) presented data that phosphatidylserine also may bind to the homologous site in CaV1.2. The authors need to use the more conservative term phospholipid in more places rather than PIP2.

In the introduction the authors cite that Kaur et al. (2015) as demonstrating a homologous polybasic sequence in the I-II loop that is the site of PIP2-mediated reluctant gating. In rereading that paper, Kaur et al. conclude the site stabilizes channel gating (see abstract of their paper). They did probe gating changes but did not make any definitive conclusion about the existence of a reluctant site. Therefore, the authors should remove the last two sentences (found in lines 4-7) on page 4.

Page 3; 2nd paragraph, 1st sentence. I believe the authors mean intracellular regulatory signals, not intercellular regulatory signals.

Page 4. 2nd paragraph. Please also add Richards et al. from the Dolphin lab and Miranda-Laferte et al., 2011, to the list of citations on the function of the HOOK region.

Figure 1A. Please state in the legend that the location of the β-subunit is based on CryoEM of Cav1.1, the skeletal muscle L-type channel, and β1a locations, but not β2 isoforms or an β-subunit from the CaV2 family. This is important because CaV1.1 gates quite differently than the CaV2 channel family. This may indicate that a β-subunit localizes to a similar but somewhat different position than in CaV2.2. Also please state where the AID region is in the I-II loop, i.e., near the N-terminus end of the I-II loop.

Figure 1D. Explain the rationale for why the upper traces are measured over a 500 ms test pulse for visualizing inactivation while the lower traces are measured at ~ 10 ms to document changes in current amplitude following Dr-VSP. Why not use a long test pulse in both instances?

Figure 1-Suppl 2B looks as though the Y-axis is mislabeled. It should simply read % Inhibition by a prepulse.

Figure 2A- State which beta2 variant, beta2a, beta2c, or another were mutated and used in the results presented in Figure 2A.

Page 8. 3rd line from top of page. Together, our date further confirm….. This is an overstatement. Please change confirm to, "… are consistent with our model where…."

Combine Figure 4 suppl 2 and4 for easier comparison.

Figure 4E. The Y-axis looks like the label should be normalized current remaining rather than inactivation. The same for PIP2 depletion curve. Whether this is a correct analysis of the curve, the plot needs some clarification.

Figure 4 suppl 3D. The X-axis is mislabeled and should read V1/2 activation NOT inactivation.

Page 11. Last sentence of the 1st paragraph. This is an overinterpretation of their data. If the authors think the channels inactivate when both sites lose their phospholipid, they need to show the channels are stabilized in an inactivated state by plotting voltage-dependent inactivation graphs. If the authors don't have the data, they should state the channels are stabilized in a nonconducting state.

[Editors' note: further revisions were suggested prior to acceptance, as described below.]

Thank you for resubmitting your work entitled "Molecular basis of CaV β-and PIP2-dependent regulation of the CaV2.2 voltage-gated Ca 2+ channel" for further consideration by *eLife*. Your revised article has been evaluated by Richard Aldrich (Senior Editor) and a Reviewing Editor.

The manuscript has been improved but there are some remaining issues that need to be addressed, as outlined below:

The referees' comments on your revised manuscript are attached. All agree that there are many significant results but also felt that the manuscript did not make it clear how these results tested specific questions and thus advanced the field. That is, the manuscript should (1) pose a central set of key questions at the outset and describe how these questions relate to previous work in the field, (2) as much as possible, frame these questions in the form of testable hypotheses, (3) propose a concluding model that incorporates the new and previous results but also indicates major areas of uncertainty. Put another way, the manuscript in its current state did not make it clear how your results relate to other work in the area, exactly what gating states of the channel your work seeks to characterize, and whether you think the model presented in Figure 8 summarizes all these gating states and their modulation by PIP2/phospholipids, Gbeta-γ and phosphorylation.

To expand on these points, one of the reviewers states: Could the authors be more explicit about how previously defined channel states that they refer to in the introduction and throughout (e.g. as detailed in Wu et al., 2002, who propose 6 distinct states which include 2 PIP2 binding domains "R" and "S"). In particular, the reviewers were not clear whether you were trying to imply that the R and S states were equivalent to the reluctant and willing states. Also, Wu et al. proposed that there was phosphorylation at the R site. Furthermore, it is not clear how your results compel one to think that all the PIP2 effects on CaV2.2 are a consequence of binding to either "the" S site or "the" R site. Indeed, what is the evidence that there are exactly two discrete sites? In this regard, one of the reviewers states: There is no need for Park et al. to assign a gating term (willing or reluctant) to either of the two sites. The paper will be much clearer if they call the I-II linker polybasic phospholipid binding site as just that and the VSDII site, the VSD-II PIP2 binding site, and nothing more: not a willing or reluctant site. A point of particular importance is that the model illustrated in your Figure 8 seems to have inverted the function of PIP2 binding site found by Gao, Yao and Yan in the cryo structure. Gao et al. think PIP2 binding to IIS4 inhibits activation, whereas Figure 8 designates this as an S site. Relatedly, the interactions between the CaVbeta subunits and PIP2 effects on CaV2.2 are important and might indicate a direct occlusion of PIP2 binding by membrane-anchored β, as shown in Figure 8, but aren't other possibilities equally plausible? Because CaVbeta subunits have global effects on channel gating, couldn't they interfere with the actions of PIP2 without physically occluding PIP2 binding? In regard to the effects on channel function 10 sec after M1R stimulation, how were the actions of phospholipid metabolism and those of Gbeta-γ binding distinguished from one another? If there was no way to distinguish, then the description of these results on page 11 must be stated in a way that acknowledges this difficulty. There is also an issue about whether the mutations in the polybasic segment in the I-II linker are affecting PIP2/phospholipid binding or more generally altering gating. Additional mutations in the I-II linker would help resolve this, but are not required as long as the manuscript makes clear that such mutations would be needed in future experiments.

The bottom line is that the reviewers and I believe the manuscript still requires substantial changes. As authors ourselves, we know that requests for such changes can be irritating. However, if the reviewers have concerns/confusion over some of your results, it is likely that other readers will be in a similar situation.

*Reviewer #1 (Recommendations for the authors):*

I have read the revised manuscript and accompanying response from the authors. As a reviewer, I am satisfied in the textual changes made and the inclusion of accompanying data provided by the authors in response to the previous reviews.

*Reviewer #2 (Recommendations for the authors):*

The authors have addressed most of the comments in this revised manuscript. They have added new data, removed other more preliminary findings, and incorporated findings of recently published cryo-EM structure of hCaV2.2 (Gao et al., 2021). A few remaining comments:

– The experimental rigor of this study is strong and the authors are experts in this important field. The manuscript does still lack clarity regarding the overarching goal/specific questions being addressed by the study. The additional first paragraph in this revised manuscript is helpful, but an illustrative model in the introduction could anchor the reader and bring the significance and impact of the studies into sharper focus. This would give important context to the impressive experiments and the results in this manuscript, it would highlight current gaps in knowledge being addressed, and it would also help underscore/appreciate precisely how the authors proposed model (Figure 8) advances our understanding of "the molecular basis of Cavb and PIP2 dependent regulation of CaV2.2 voltage-gated calcium channels".

– Could the authors be more explicit about how previously defined channel states that they refer to in the introduction and throughout (e.g. as detailed Wu et al., 2002 propose 6 distinct states which include 2 PIP2 binding domains "R" and "S") are incorporated into the authors final model (Figure 8).

– The Results section is clearer and the subheadings are helpful. However, the first page of the Results section, in particular, is not effective in framing the rest of the report. Is not obvious why the Results section starts with a relatively long introduction to ".multiple types of CaV b subunit…." (lines 100-107). This could be more effective.

– As previously recommended. The authors should be careful when referring to distance between protein motifs if they are not actually measuring distance. There are still some instances of this in the Results section.

– Discussion. Consider removing the description of the concluding model in Figure 8, to the start of the Discussion section. Also, importantly, include statements about how the new model differs from, and how it is an advance over previous publications/models. More generally, the discussion could benefit from more explicit statements of what is new in this manuscript and how the author's proposed model helps to advance understanding.

– The manuscript should be read thoroughly for grammatical accuracy; this includes the abstract, introduction, results, and discussion.

*Reviewer #3 (Recommendations for the authors):*

The revised manuscript by Park et al. is much improved in several ways. The requested details from the 3 Reviewers have added clarity to the manuscript. Additionally, Park et al. gave succinct responses to all of the reviewers' concerns. The description of Lyn11 being lipidated on vicinal amino acids gives new insight to potential mechanistic explanations for some of the data. Removal of the idea of the I-II linker somehow being pressed against the membrane by a tethered β subunit also appears to simplify data interpretation. The added data in figure 6 has greatly added to clarifying Gbeta-γ effects vs phospholipid effects during M1R stimulation. They show that at 10 sec of muscarinic stimulation, the majority of the palmitoylated CaVb2a-containing channels are modulated by a VD mechanism, most likely by Gbeta-γ binding to CaV2.2. The recent publication by Gao et al. 2021 of a CryoEM structure of CaV2.2/beta3/alpha2delta1 with a PIP2 molecule bound to the VSDII of CaV2.2 has contributed structural data that was used to interpret the biophysical data presented here, which also improves the paper. Additionally, in Figure 7, Park et al. have pivoted in their focus from the polybasic segment of the I-II linker to test the phospholipid sensitivity of channels with key residues in the VSD-II identified by Gao et al. as residues that interact with a phosphoinositol headgroup. The quality of the data is excellent. The significance of the findings are timely and extremely important for understanding the unique physiology of CaV2.2 channels.

That said, a major shortcoming of the manuscript remains in the interpretation of the results, which all three Reviewers were concerned about in the first version of the manuscript. I remain concerned. These shortcomings fall under three problems and must be fixed, so that if published, the model does not create another unnecessary conflict in the field which then takes years to undo. While it is great that the Suh lab has reinterpreted their model based on their new data in Figure 7, they now have taken on not only the polybasic phospholipid binding site but also the VSDII PIP2 binding site with only scant data to support their definitive conclusions. In particular three issues stand out.

Issue #1: Willing (S) and Reluctant (R) gating vs the predicted PIP2 "R" and "S" binding sites. I have reread carefully, Wu et al. 2002; Kauer et al. 2015; Gao et al., 2021, Dong et al. 2021 as well as Heneghan et al. 2009 in addition to work from the Suh lab. What seems clear is that CaV2.2 has multiple lipid binding sites that influence channel gating. The problem for all of us is that we are trying to identify the "R" and "S" binding sites that mediate "reluctant" vs "willing" gating and channel stability, first described by Wu et al. (2002). The movement of different parts of the channel may converge to create a signature pattern of gating. In this manuscript Park et al. are calling VD gating as "Reluctant Gating" and that it is the polybasic segment on the I-II linker that is responsible for VD gating. However, from my reading, Gao et al. 2021 favor a model where the PIP2 binding site on the VSD-II regulates voltage-dependent (VD) gating, not the polybasic segment. I quote from Gao et al's discussion, "The PIP2 binding site observed in this study may account for the voltage-dependent inhibitory modulation as it stabilizes the down conformation of the VSDII. It remains to be investigated whether a separate PIP2-binding site is responsible for the voltage-independent rundown reduction, a mechanism that may involve Gbg proteins." The problem is that Park et al. have come to the OPPOSITE conclusion. If I am reading this paper correctly, Park et al. have assigned the VDII PIP2 binding site as the "S" site and the polybasic site on the I-II linker as the "R" site with the "R" site responsible for VD gating. Intuitively, the Gao et al. and the newer Dong et al. CryoEM structures showing PIP2 stabilized a down state of the VSD-II should make the channel harder to open. Therefore, if that PIP2 is metabolized by M1R signaling, the VSDII should now move more easily with depolarization, which is a key characteristic of the original R site (Wu et al., 2002); Another way to think of this is that if the VSD-II site was responsible for "R" gating, its metabolism is predicted to remove the restraint on the up-movement of the VSDII and should give rise to more rapid activation kinetics and more current – NOT inhibition. Yet other labs that have studied N-current modulation by M1Rs observe current inhibition following activated M1R signaling, suggesting that another site – perhaps the polybasic segment of the I-II linker-- is the "S" site involved in N-current inhibition by Oxo-M. These two CryoEM structures are in major conflict with Park et al's model – if I am understanding it properly.

The point I am making may seem long-winded but understanding CaV2.2's structure and gating is an extraordinarily important issue to NOT GET WRONG. What may be smartest for Park et al. to do is to back off of their conclusion and state the pros and cons of which site might be the "R" vs "S" phospholipid binding sites. Notably both the Gao et al. and Dong et al. were very careful about assigning "their PIP2" to a particular gating function, raising the issue that if CaV2.2 does have more than one PIP2 binding site, where is it? Neither CryoEM study found a phospholipid that bound to the polybasic site on the I-II linker. Moreover, the N-terminus of CaV2.2 was clipped off which may also alter channel structure and/or CaVbeta orientation and therefore stabilize/destabilize a particular channel construct. There is no need for Park et al. to assign a gating term (willing or reluctant) to either of the two sites. The paper will be much clearer if they call the I-II linker polybasic phospholipid binding site as just that and the VSDII site, the VSD-II PIP2 binding site, and nothing more: not a willing or reluctant site. At the very least, Park et al., need to clearly point out the different conclusion of Gao et al. with theirs; even better, they may want to reconsider their conclusions of their data. Indeed, my interpretation of the data in Figure 6 suggest that the 4A substitutions in the I-II linker do indeed result in a loss of M1R-mediated current inhibition due to phospholipid metabolism. If so, it seems that this is the "S" site.

Issue #2. What lipid(s) bind to the I-II polybasic binding site. Park et al. present a remarkably simplistic and simultaneously aggressive interpretation of their phospholipid data in calling the polybasic I-II linker site "THE other PIP2" binding site. I make this statement for the following conclusions. (1) Gao et al. and Dong et al. did not capture a PIP2 bound to the polybasic region of the I-II linker. (2) Kauer et al. found their data inconclusive as to whether the homologous polybasic segment on CaV1.2's I-II linker is a specific PIP2 binding site or a more general phospholipid binding site; and also were unsure whether the site regulates "S" or "R" gating. (2) Park et al. did not provide control experiments where other basic residues in this segment of the I-II linker (shown in Figure 5 Supplemental 1) are mutated to alanine to determine the specificity of their mutations vs others in controlling gating. Indeed, it's possible that disruption of any part of that segment will alter gating. (3) Furthermore, they have not shown that disrupting this site might alter a PIP2 binding site elsewhere. Imaging data only show low resolution membrane localization of membrane associated fluorescing reporters and don't serve to identify PIP2 binding sites on channel subunits. I point out that the more we learn about the cytoplasmic loops of CaV2.2, the more intimate and complicated their interactions appear to be, such that a point mutation in a charged residue at one location may have profound effects on multiple aspects of channel function far from that residue. Much more data need to be collected before we have a clear understanding of lipid effects on CaV2.2 gating. (4) We don't know even how many critical inner membrane lipid binding sites might exist in the CaV2.2 channel complex; collective findings suggest at least two, but do not rule out more. Therefore, the authors must denote the polybasic I-II linker site not as "the" PIP2 binding site but as a potential PIP2 binding site or simply a phospholipid binding site. Backing off this terminology does not detract in any way from the findings. Indeed historically, calcium channel biophysicists were very careful in interpreting DHP action on L-current with identifying DHP binding sites on L-channels and the same with G-protein interaction sites with CaV2 channels. Please replace "PIP2" binding site with "phospholipid" binding site when discussing the polybasic I-II linker.

Issue #3. The interpretation of what PIP2 molecules/their metabolism may be doing to Gbeta-γ binding following M1Rs is way overstated on page 11. There are certainly other possibilities, for example, it is known that a number of kinases that phosphorylate CaV2.2 are also activated following M1R stimulation and have notable effects on N-channels. Park et al's interpretation could be correct, but they don't have the data to back up their model. The authors need to remove their comments about PIP2 and Gbeta-γ from the end of the paragraph on page 11.

Other needed changes.

1) Please include the GDP-betaS data in Figure 1 Suppl2. They would be helpful there. I am amazed that there is no tonic G-protein modulation, but Figure 6F also clearly supports this conclusion. Just to be a stickler about conclusions, the statement that just because 0.1 mM GTP has no effect on prepulse currents doesn't mean this concentration is physiological as the authors state early on in their rebuttal. Just let the data stand in this case.

2) Figures1A, 2A, 3A, 4A diagrams show Lyn 11 monolipidated, but in the discussion the authors comment that Lyn 11 is myristoylated (presumably on G3 and palmitoylated on C4). This fact changes dramatically the interpretation of the Lyn11 construct data. Please state this when first using Lyn 11 constructs. Also the authors state that C3 and C4 are of the Lyn11 construct are lipidated. Please correct the C to a G in the discussion text.

3) For clarity, please add another tail to all Lyn11 schematics in Figures1A, 2A, 3A, 4A.

4) Also please site a reference that shows this simultaneous dual lipidation of Lyn 11.

5) Figure 3B needs a sample size.

6) The concerns raised by Reviewer 2 do not seem adequately addressed. In particular, Reviewer #2 commented there are many other explanations for the changes in currents from the RF and RFC constructs, yet the authors contend all the changes are due to the β subunit's closeness to the plasma membrane. The authors need to back off of this conclusion.

In summary, Park et al. have taken on attempting to (1) define CaV2.2's "R" and "S" PIP2/phospholipid binding sites, (2) describe gating associated with different PIP2 binding sites, (3) illustrate the functional effects of vicinal lipidation of CaVbeta subunits on PIP2 binding to CaV2.2 and consequent gating and modulation, and lastly (4) attempt to define the functional relationship between PIP2 and Gbeta-γ with limited numbers of control experiments for each of these 4 important aspects of CaV2.2. I remain uncomfortable with the conclusions in this manuscript without further major changes.

[Editors' note: further revisions were suggested prior to acceptance, as described below.]

Thank you for resubmitting your work entitled "Molecular basis of the PIP_2_-dependent regulation of Ca_V_2.2 channel and its modulation by Ca_V_ β subunits" for further consideration by *eLife*. Your revised article has been evaluated by Richard Aldrich (Senior Editor) and a Reviewing Editor.

The manuscript has been improved but there are some remaining issues that need to be addressed, as outlined below:

The reviewers agree that the revised manuscript describes beautifully performed experiments, but that it still suffers from a lack of focus and insufficient framing. Specifically, what mechanistic questions were the experiments designed to address? How do the results bear on those questions? What alternative explanations are also compatible with these results? What future experiments would help to eliminate these alternatives? Below, I have tried to give examples related to these points.

1. The authors note that they are not addressing the relationship to the R and S functional states, but the introduction still starts – as it did in previous versions, by discussing R and S states as they relate to PIP2 modulation of CaV2 channels. This sets expectations that the authors will develop a consensus new model.

2. A criticism of the previous version of the manuscript was the use of "distance" based on the number of amino acid residues. The revised version has replaced "distance" with "length" (e.g., Figure 4), but the implication is the same: that the length defined in this way is an indication of physical distance. Although this may be correct, there are also other possibilities (e.g., differences in torsional rigidity) which should be discussed.

3. The muscarinic modulation experiments do not demonstrate clear pathway separation. These experiments need to be designed to isolate the effects of PIP2 metabolism from Gbeta-γ binding to CaV2.2.

4. For the proposed I-II linker phospholipid binding site in the I-II loop C-terminus, it should be determined whether mutation of other arginine residues in the distal region does not alter phospholipid actions on CaV2.2 gating to support the idea that a specific binding site has been identified.

5. There also need to be additional control mutations in VSD-II to demonstrate that only those in the proposed binding site for the PIP2 headgroup affect the actions of PIP2.

6. The previous version of the manuscript suggested that there were two binding sites important for the actions of PIP2 on CaV2.2, whereas the current version suggests there are three. Is it possible that there are only two binding sites, and that PIP2 has two different actions at this one site depending on the state of the channel?

*Reviewer #2 (Recommendations for the authors):*

The new focus of this revised manuscript is to describe the molecular basis of β-subunit control of PIP2 inhibition of CaV2 channels.

The authors' main findings are 1. That there are 2 binding sites for PIP2 on I-II linker and in the domain II S4. The authors and others have reported previously that the β subunit can modulate channel sensitivity to PIP2 including showing that membrane-linked and non-membrane-linked β subunits have differential effects on PIP2 action. In this study, the authors extend these findings, motivated by recent cyro-EM studies of PIP2 binding to CaV2.2 channels. They conclude that Cavb subunits that are membrane-anchored prevent the action of PIP2 via I-II site. But β subunits have no effect on PiP2 action via domain II S4.

A major issue with the previous versions of this manuscript was the lack of clarity about how the authors' study and models of PIP2 action related to previous functional work that defines (among other findings) R and S channel states. In their response, the authors have removed references to this model (except in the introduction – see below). This narrows the focus of the work and limits the impact/broader relevance of the study.

The technical quality of the work is not in question, it is very high and rigorous. The authors have contributed functional data showing the role of the domain II S4 site defined in cryo-EM studies of CaV2.2 and shown its lack of CaVb subunit dependence. They have also refined our understanding of the interaction between Cavb, PIP2 and the I-II domain of CaV2.2

However, it remains difficult to extract the broader impact of this study.

The authors note that they are not addressing the relationship to the R and S functional states, but their introduction still starts – as it did in previous versions, discussing R and S states as they relate to PIP2 modulation of CaV2 channels. This sets expectations that the authors will develop a consensus new model.

Other comments

L.175. See previous review. The authors are still implying distance measurements "…functional role of length between lipid anchor and GK domain…". In this context length and distance are equivalent. Also, see L.197 and several other instances including in the conclusions.

*Reviewer #3 (Recommendations for the authors):*

This is the third version of the Park et al. manuscript entitled, "Molecular basis of the PIP2-dependent regulation of CaV2.2 channel and its modulation by CaV β subunits". Previous studies from several labs have provided data suggesting that PIP2 has 2 functional effects on CaV2.2. These effects are thought to be due to PIP2 binding to two unidentified sites on CaV2.2. The initial manuscript was submitted prior to the publications of Gao et al. (2021) and Dong et al. (2021), both of which have provided high-resolution crystal structures of a CaV2.2/ alpha2delta/ β complex with a PIP2 head group bound to Domain II's voltage-sensing domain (VSDII) and its fatty acid tails interacting with S5 and S6 pore helices. Consequently, Park et al. have pivoted their manuscript to not only clarify their original findings surrounding the role of lipidated β subunits in altering PIP2 sensitivity of CaV2.2 currents and the importance of the polybasic residues in the I-II linker but now also to probe whether a putative PIP2 head group bound to VSDII influences channel gating and/or kinetics, which would indicate that VSDII is one of the functionally important PIP2 binding sites. The authors contend their re-submission has 4 major findings:

1) The length of an amino acid tether, bound to a β subunit, alters CaV2.2 currents only if the tether is short.

2) The vicinal myristylated and palmitoylated residues of the Lyn domain of the tether mimic vicinal palmitoylation of CaVb2a. They hypothesize that in both cases the lipids may compete with PIP2 for binding to the "S" site on CaV2.2, which has not been identified.

3) Polybasic residues in the I-II linker of CaV2.2 may be a nonspecific phospholipid binding site that will bind PIP2 to affect channel gating and is the site of competition with CaVbeta2a's palmitoyl groups.

4) In this re-submission, the authors present new experiments testing whether the recently discovered PIP2 interaction site on the VSDII may be a second PIP2 binding site.

While the data in Figures1-4 provide substantial data for a model where vicinally lipidated β subunits compete with PIP2 binding to CaV2.2, the experiments supporting points 3 and 4 have experimental design problems and only superficially probe the putative PIP2 binding sites with few control experiments. For example, in their initial submission, Park et al. probed whether polybasic residues in the channel's I-II linker may be a PIP2 binding site similar to the phospholipid-binding site described for CaV1.2 by Kaur et al. (2015) and also the site disrupted by palmitoylated CaVb2a. They mutated a subset of the basic residues present in the distal region of the I-II linker and showed changed gating. However, no other mutations of basic residues were made, so it is not clear how specific these residues are for PIP2 binding, nor whether PIP2 may still have been bound, but the I-II linker changed its orientation due to the neutralization of charge resulting in changed gating rather than a loss of PIP2. Additionally, they document some differences in response to muscarinic stimulation, yet their experimental design does not separate G-protein binding to CaV2.2 from PIP2 metabolism, making data interpretation extremely challenging. Indeed, their WT channel data (Figures 6B and E) don't fit the kinetic changes of CaV2.2 previously observed with Gq mediated PLC activation, indicating that much of what they are examining is not the result of PIP2 metabolism. While Park et al. seems to focus on whether they have changed G-protein mediated modulation. However their data indicate that it is not that b2a is more sensitive to G-protein modulation, but rather it is that b2a-containing channels are insensitive to PLC breakdown of PIP2. Moreover, insufficient data are presented to make a conclusion about the relationship between the VSDII and PIP2 to determine whether this PIP2 mediates one of the previously described actions of PIP2 on CaV2 gating.

Lastly from the crystal structure, the distal I-II linker is close to the VSDII raising the possibility that a single PIP2 residue may interact with both the VSDII and the I-II linker. Amongst all these issues, the authors also bring up PIP2 interaction with the HOOK region of CaVbeta subunits as well as with the N-terminus of CaVbeta2e. Altogether, despite the beautiful current recordings as well as the nice diagrams, it became very difficult to sort out how these observations fit a model of lipid interactions with the CaV2.2 channel complex. While the authors have provided one model to explain their data, they have not ruled out others that could also explain their data.

[Editors' note: further revisions were suggested prior to acceptance, as described below.]

Thank you for resubmitting your work entitled "Molecular basis of the PIP_2_-dependent regulation of Ca_V_2.2 channel and its modulation by Ca_V_ β subunits" for further consideration by *eLife*. Your revised article has been evaluated by Richard Aldrich (Senior Editor) and a Reviewing Editor.

Your revised manuscript has now been seen by two of the original three reviewers. They agree that the manuscript has been greatly improved and presents important new results. However, one of the reviewers had suggestions for editorial changes, which are detailed below. The only significant issue raised by that reviewer was in regard to the data presented in Figure 6 (see below) and that reviewer argued that Figure 6 should be removed and replaced by Figure 6-S1 and Figure 6-S2. If you are willing to make this and the other suggested editorial changes, your manuscript will hopefully be acceptable without further review.

Detailed Reviewer Comments:

– Line 41. Please include Wu et al. 2002 to acknowledge that they were the first to show that PIP2 is necessary for voltage gating of CaV2.1 and 2.2.

– Line 72, please site work from the Colecraft lab on CaVbeta subunits and others in addition to the Suh and Hille labs.

– Line 228. The sentence, "….This suggested that incremental increases in linker length lead to a decrease in channel gating….", may be easier to understand if it is changed to "This suggested that incremental increases in linker length lead to a decreased voltage-sensitivity.

– Page 10 lines 277-279. This is an important conclusion, but the sentence is awkward/not grammatically correct. The sentence may be clearer if expressed as: More importantly, our data indicate that PIP2 interacts with the polybasic motif when CaV2.2 is expressed with cytosolic b-subunits but not when expressed with lipidated membrane-anchored b-subunits.

– Figure 5-S3 D. Seems weird that the VSP has no effect when the model predicts that there is another PIP2 bound to the channel that in theory should be metabolized by activating the phosphatase. The authors are arguing that the 5% inhibition is due to the 2nd PIP2 being metabolized. However, the model by Wu et al. would suggest they should see enhancement with the 2nd PIP2 metabolized, not inhibition. Some explanation of this discrepancy would be helpful.

– Page 11, lines 290-292. Changes in gating with additional RA mutations don't support their conclusion about the nonspecificity of lipid binding to the I-II linker. They didn't probe the effects of binding by other lipids to these mutant channels. Rather, their data further characterize which amino acids may participate in binding phospholipids/PIP2. Thus, this sentence should be removed.

– Figure 6 is fraught with several signal transduction/experimental design issues outlined here. First, muscarinic receptors, when transfected into HEK/TSA cells overexpress, and consequently are notoriously promiscuous in their coupling to signal transduction cascades, complicating data interpretation. In the 1990s the Hille lab published several papers looking at native ca^2+^ channel modulation by various receptor signaling cascades. Notably, Shapiro et al. (1999) presented data showing that the membrane-delimited pathway is carried primarily by M2Rs and that when this receptor is knocked out, no other muscarinic receptors, (e.g, the M1Rs) took their place in neurons.

Whereas in this manuscript a recombinant system seems to exhibit M1R and M2R promiscuity in signaling. Second, Shapiro et al's data show that the diffusible 2nd messenger pathway, now known to be due to PIP2 metabolism, takes ~80 sec to reach full effect. Here, the data in Figure 6 are all collected 10-20 seconds after the application of Oxo-M, which would miss much of the PIP2 modulation of CaV2.2. Third, the previous metabolism of the 2nd PIP2, bound to channels, increases voltage sensitivity causing a negative shift in G/Gmax curves, seen best at negative test potentials, yet negative test potentials were not monitored independently of G-protein effects. Together the issue of promiscuous muscarinic receptor coupling to signaling cascades as well as issues with timing and optimal voltage protocols, make Figure 6 results meaningless. I strongly encourage the removal of Figure 6 in favor of keeping the better-designed expts in Figure 6-S1 and S2.

– Figure 6-S1A. Please show in the schematic of the chimeric which loops and domains are 1C vs 1B with different colors. I may have missed it but, please include a description of the alpha1B-1C chimera either in the Methods or Results section. If this chimera was used previously by you or others, please cite the reference for the reader. If it was a gift, please acknowledge this.

– Figure 6-S2. Lines 242-244. This sentence should be removed. Not enough control experiments were done to make this conclusion. The authors haven't convincingly shown that they have the two pathways isolated from one another. One clear way to do this is simply to raise the [BAPTA] from 0.1 to 10 mM. The data in Figure 7 just scratch the surface of PIP2 interaction with VSDII. In addition to the discussion about charge changes in the voltage-sensor vs PIP2 binding to gating charge residues is necessary for clarity.

Additional overstated results in the discussion need backing off since other explanations may account for the results:

– Page 14 line 381. Please change "demonstrated" to predict.

– Line 421, change "…We found that…" to "…Our data are consistent with…"

– Line 425, "…Our data demonstrate…" to Our data indicate…"

---

## [Author Response]

The work described in the manuscript provides new information about the roles of PIP2/phospholipids and the β subunit in regulating the gating of the calcium channel CaV2.2 and is likely to be of broad interest because CaV2.2 is an important pharmacological target. In particular, the work identifies a previously unknown site in the CaV2.2 loop at which phospholipids regulate channel gating. However, there was consensus among the reviewers that important conclusions of the manuscript are not well supported by the experimental results.In your consideration of a revised version, please take note of the following.1. The title or abstract should explicitly state that the experiments were conducted on transfected tsA201 cells.

We thank the reviewer for constructive comment. As suggested, we input “tsA-201” in the revised abstract (p.2).

2. The Discussion needs to be better organized and more succinct. The Reviewers have made specific suggestions in that regard.

We thank the reviewer for valuable suggestion. As recommended, we reorganized the Discussion section into specific subtopics and inserted separate subtitles for each subtopic in the revised version (p.13-17).

3. Because GTP in the pipette should have promoted G-protein mediated stabilization of the reluctant state, the prepulse depolarization used to activate DR-VSP could have caused the G-protein to dissociate and relieved this tonic inhibition. Thus, the authors need to provide more methodological detail about the prepulse application and describe how the effects of activation of DR-VSP could be experimentally distinguished from G-protein dissociation. Otherwise, the possible effect of G-protein dissociation complicates the interpretation of the experiments with DR-VSP.

We thank the reviewer for raising this concern. To clarify this issue, we have re-performed the Figure 1D experiments with 1 mM concentration of GDPβD in the pipette solution. Since GDPβD inhibits the basal activation of G protein and also the possible Gβγ-mediated suppression of CaV2.2 channels, we can eliminate the voltage-dependent current relief by the prepulse depolarization under the condition. With the same Dr-VSP activation protocol, we found that CaV2.2 channels with 1 mM GDPβD show very similar PI(4,5)P2 sensitivities to those of experiments with 0.1 mM GTP (please see new Author response image 1), suggesting that 0.1 mM GTP concentration in the pipette solution is not sufficient to trigger the G protein activation or suppress the CaV2.2 channels through the Gβγ-dependent pathway. Thus, our data demonstrate that 0.1 mM GTP concentration is close to the normal intracellular GTP level in the basal condition and the +120-mV prepulse changes the PIP2 responses through the activation of Dr-VSP, but not through the dissociation of the Gβγ subunit from the channels.

**Author response image 1. sa2fig1:** Summary of the Ca_v_2. 2 current inhibition by Dr-VSP-mediated PIP_2_ depletion in cells were recorder with pipette solution containing GDP-β-S. (A) Cells received a test pulse (a), then a depolarisation of 120mV, a hyperpolarisation to -150mV, followed by the second test pulse (b). Ca_v_2.2 currents before (a) and after (b) the depolarizing pulse are superimposed in cells without Dr-VSP (Upper) and with DR-VSP (bottom). (B) Summary of the Ca_v_2.2 current inhibition (%) by a depolarizing pulse to +120mV in cells without Dr-VSP (Upper) and with DR-VSP (bottom). Dots indicate the individual data points for each cell (*n*=8-11). Data are mean ± SEM.

4. It should also be acknowledged that DR-VSP may not have equivalent access to the various pools of PIP2 that are important for regulating channel gating.

We appreciate the reviewers for this insightful suggestion. The reviewer’s point was previously tested in the model cell systems (Suh et al., 2010). According to the FRET assay, there was no difference in the maximum effects between M_1_R activation and Dr-VSP activation on Ca_V_2.2 current suppression. Those results suggest that Dr-VSP system can deplete the membrane PIP_2_ in the various pools, like M_1_R activation. However, in the cases where the apparent binding affinity of PIP_2_ to membrane protein is very high and/or the PIP_2_-binding site is isolated (pocket-like structure) and surrounded by other basic and hydrophobic residues in the N- and C-terminus, the Dr-VSP may be not able to access to the site. For instance, some Kir2.1 K^+^ channels with relatively higher PIP_2_ affinity show less sensitivity to receptor stimulation or current rundown, even though they require PIP_2_ for activation (Huang et al., 1998; Zhang et al., 1999). In our results, the weak sensitivity of S site to the Dr-VSP could be partly due to the low accessibility of the protein to the PIP_2_-binding S site. The inequivalent accessibility of Dr-VSP to the PIP_2_ on different binding sites was described in the Discussion section (p.16, the first paragraph).

Suh, B.C., Leal, K., and Hille, B. (2010) Neuron 67, 224-238.

Huang, C.L., Feng, S., and Hilgemann, D.W. (1998) Nature 391,803–806

Zhang, H., He, C., Yan, X., Mirshahi, T., and Logothetis, D.E. (1999) Nat. Cell Biol.1,183–188

5. Features of the model presented by the authors to explain PIP2 actions on channel gating are not constrained by the experimental results and thus too speculative. These include statements about distance of the β subunit from the membrane and movement of the I-II loop, statements about the occupancy of the R and S states in relationship to the experimental results, and statements about PIP2 binding rather than phospholipid binding more generally.

We thank the reviewer for constructive critics. In the middle of our revision period, the human Ca_V_2.2 channel complex was solved by cryo-electron microscopy at a resolution of 3.0Å (Gao et al., 2021). The structure indicates that there is a structured PIP_2_ binding pocket within the S4_II_ domain of Ca_V_2.2 channel and PIP_2_-binding to the site is important in solidifying the VSD_II_ in a down conformation. On the basis of the structural data and our new experimental results, now we can provide the PIP_2_ regulation on Ca_V_2.2 channel activity more clearly.

First, the conformational shift of I-II loop looked impossible since the loop was already attached to the S4_II_ domain of Ca_V_2.2 channel even with cytosolic β3 subunit (Gao et al., 2021). This suggests that the N-terminal length of membrane-anchored Ca_V_ β subunits is important for translocating the hydrophobic fatty acid tails of lipid anchor (e.g. Lyn_11_, PS or PIP_2_) to the channel complex and replacing the PIP_2_ molecule bound to the reluctant R site in the I-II loop. Thus, the differential regulation of Ca_V_2.2 channel activity by the diverse variants Ca_V_ β subunit can be explained by the bidentate model (p.13-14, Discussion section).

Second, Gao group reported that the 5-phosphate group of PIP_2_ interacts with two basic residues (R584 and K587) in S4_II_ domain of Ca_V_2.2 channel. However, the functional role of the site was not tested in their paper. By neutralizing the two amino acids, here we examined the role of the site in Ca_V_2.2 channels with different types of Ca_V_ β subunit. Our results in new Figure 7 show that (1) mutation of the site completely blocks the PIP_2_ sensitivity in channels with β2a, but partially inhibits the PIP_2_ sensitivity in channels with β2c; (2) mutation of both sites including the pocket in S4_II_ domain and the polybasic motif in the I-II loop blocked all the PIP_2_ responsiveness in channels with the β2c; (3) mutation of the site in S4_II_ domain shifts the voltage-dependent activation curves to the right direction. Together, the results suggest that the PIP_2_-binding to the site in S4_II_ domain is important in stabilizing the channels in more conductive willing state (p. 14-16, Discussion section).

With the new data, we think that our results provide sufficient evidence for the presence of two PIP_2_-binding sites in Ca_V_2.2 channels, the S site in S4II domain and the R site in the I-II loop, though the R site seems to interact with more general anionic phospholipids such as phosphatidylserine and PIP_2_ in the plasma membrane (we corrected this point in the revised version). Our new data now sufficiently support our hypothesis that Ca_V_2.2 channel gating is regulated by PIP_2_ through the dual modulatory mechanisms in a Ca_V_ β subunit-dependent manner.

Gao S, Yao X, and Yan N. 2021. Structure of human Ca_V_2.2 channel blocked by the painkiller ziconotide. Nature 596, 143–147.

Reviewer #2 (Recommendations for the authors):[…]The authors might consider the following comments/suggestions to help clarify and strengthen the manuscript.– The introduction could be strengthened by starting with a statement about the question that is being addressed. As it is now, the introduction is used to introduce each key component of the a1, β complex and PIP modulation but a clearer direct leading paragraph would help provide context and rational for the background information.

We thank the reviewer for constructive suggestion. Based on the reviewer’s point, we shortened the first paragraph of the Introduction for leading to the main topic of the story.

– Similarly, the Results section would benefit from a leading sentence or two about the logic of the experiments and their connection to the proposed model.

We thank the reviewer for constructive suggestion. We divided the Discussion section according to the subtopics and added a leading title to each in the Discussion section. We also added some leading sentences in each topic and modified the subtitles in the Results section.

– The authors refer to "distance" from the plasma membrane to the Cavb subunit (e.g. first subheading of the results and throughout) which implies a measurement which is not the case. Why not say that the linker disrupts the effect of PM anchoring on CaV channel inactivation and inhibition by PIP2 depletion? The intermediate linkers 12, 24, 36 aa do not show progressive changes in current inhibition (following PIP2 depletion; supplemental figure 1-3) but do show a transition between the 24aa and 36aa linkers. Thus, there isn't a clear progressive correlation between length/distance and inhibition.

We thank the reviewer for the important point. We also agree with the idea that there isn't a clear progressive correlation between length/distance and inhibition. Thus we changed the sentence to a more correct description of the phenomena (p.6, the first paragraph).

– Define "Lyn" when first appears.

We thank the careful work of reviewer. As suggested, we modified the sentence (p.5).

– "However, the actions of the chimeric Lyn-48aa-β2c in current regulation were similar to those of the channels with the cytosolic β2c subunit, although the chimeric analogue was targeted to the PM." This is not clear, be more explicit. It's an important result.

We thank the reviewer for constructive suggestion. We modified the sentence for better understanding (p.5, bottom line).

– "…these data suggest that the distance from the PM to the CaV β subunit may be more important than PM-anchoring of CaV β in determining the inactivation kinetics and PIP2 sensitivity of CaV2.2 channels." This statement is not clear. The linker disrupts the effect of membrane anchoring.

We thank the careful work of reviewer. We rewrote this sentence based on our new observation. “These data suggest that the distance from the N-terminal lipid anchor to the GK domain of β subunit is crucial in determining the inactivation rate and PIP_2_ sensitivity of Ca_V_2.2 channels” (p.7, the first paragraph).

– Figure 2. The authors generate alanine substituted CaVb subunits in two key regions shown to be required to interact with CaVa1 subunit – SH3 and GK domains. In the untethered version of CaVb2, there is strong inactivation and PIP2 depletion inhibition consistent with data shown in Figure 1. Similarly, membrane targeting of b2 changes the phenotype to slow inactivation and weak inhibition by PIP2 depletion, but the mutant version of the membrane targeted b2 subunit does not exhibit the WT phenotype suggesting that the SH3 and or GK domains are required to prevent PIP2 depletion mediated inhibition. The authors suggest that this is evidence that there is a conformational shift in the I-II linker of Cava1 subunit to decrease both the inactivation rate and PIP2 sensitivity of CaV2.2 channels. This is an attractive model but they do not directly test this hypothesis and all manipulations are of the β subunit.

We thank the reviewer for constructive suggestion. In Figure 4, we confirmed that GK binding to the AID domain is enough for the PIP_2_ sensitive regulation of Ca_V_2.2 channels, suggesting that the SH3 domain is not needed for the PIP_2_ regulation of Ca_V_2.2 channels. In our new model, the proximal interaction of Ca_V_ β subunit with PM is the key for the β regulation of Ca_V_2.2 channels. As the reviewer’s comment, it is still unclear whether there is conformational shift in the I-II linker of Ca_V_ α1 subunit though some evidence supports the possibility (Yang et al., 2013; Subramanyam and Colecraft, 2015). Therefore, as shown in the Discussion section, the effects of N-terminal length of β subunit on channel regulation can be explained by the bidentate model (p.13-14, Discussion section).

Subramanyam, P., and Colecraft, H.M. (2015). J. Mol. Biol. 427, 190–204.

Yang, T., He, L.L., Chen, M., Fang, K., and Colecraft, H.M. 2013. Nat. Commun. 4, 14950–14957.

– "In CaV2.2 channels with Lyn-FRB-HOOK-GK-Linker-FKBP (RCF), where a 194-aa linker was inserted between GK and FKBP, however, rapamycin enhanced the FRETr signal only without causing significant changes in the current amplitude (Figure 3B, right and Figure 3—figure supplement 1)." Rapamycin did affect current amplitude and PIP2 inhibition significantly according to the data shown in figure C-F (right). As above, it's not clear how these experiments "confirm that the inactivation kinetics and PIP2 sensitivity of CaV2.2 channels are mainly determined by a β subunit-dependent conformational shift of the I-II helix."

We thank the reviewer for raising this concern. As reviewers mentioned, there was a slight increase in current amplitude and PIP_2_ sensitivity in the cells (Figure 3B). However, the increase in the responses was much weaker than those of experiments with Lyn-FRB-HOOK-GK-FKBP (RF) construct. Moreover, the average current amplitude was not significantly changed (see Figure 3—figure supplement 1). However, we slight revised the words because of the unclarity (p.7).

– "The effects of rapamycin on inactivation kinetics and PIP2 sensitivity were much weaker in CaV2.2 channels with RCF in comparison to those in channels with RF (Figure 3C-F), suggesting that rapamycin-induced dimerization was not sufficient to shift the I-II helix to the PM in channels with RCF". As per above, this is one possibility but there are many others. The data are not sufficient to make this claim. Furthermore, as noted above, there are measurable differences induced by rapamycin with the RCF construct.

As the reviewer’s comments, there could be other factors that affect the current amplitude and PIP_2_ sensitivity in experiments with rapamycin-induced translocatable constructs. We think that the minor unclear part of the results was from the extended version of the constructs. In normal Ca_V_ β subunit, the SH3 and GK domains combine to form a complex, shortening the N-terminal length. However, in this constructs FRB and FKBP fused to make a FRB-rapamycin-FKBP ternary complex. This complex may not be tightly condensed like the β subunit and the slightly extended form of the complex may explain why the PIP_2_ sensitivity is not completely removed in the cells with RCF construct.

We also modified the sentence based on our new model (please see p.7).

– Is Fig, 3-supplemental 1 mislabeled? In A the largest currents are labelled RCF in panel A but labelled RF in panel B?

We thank for the query. Y-axis of panel A shows absolute current amplitude of Ca_V_2.2 channels obtained before and after rapamycin application. However, Y-axis of panel B indicates that relative current amplitude after rapamycin application; the current amplitude after (red) rapamycin was divided by the current amplitude before (black) rapamycin application. Thus our labelling is correct.

– The experiments in Figure 4 show a strong correlation between linker length and faster inactivation- higher PIP2 sensitivity. However, it is also clear from the recordings that there are effects on the overall rate of activation. This should be mentioned in the Results section.

We thank the reviewer for constructive suggestion. We measured the activation rates and present the data in new Figure 4—figure supplement 2 (p.8, in the middle of the first paragraph).

– The experiments using muscarinic receptor are an important addition but they raise a general question. Why is there strong inhibition by M1 activation in the presence of b2a, and in the presence of b2c with the alanine substituted CaVa1 subunit, when under these conditions, there is very little or no inhibition in response to PIP2 depletion? What component of the receptor-mediated inhibition of the calcium current depends on PIP2 depletion?

We thank the reviewer for raising this constructive question. M_1_ muscarinic modulation of Ca_V_2.2 channels depends not only on the PIP_2_-dependent pathway but also on the Gβγ-dependent pathway (Keum et al., 2014): M_1_R activation strongly inhibits Ca_V_2.2 currents mostly through the Gβγ-dependent (voltage-dependent) pathway in the presence of β2a or through the PIP_2_ depletion-mediated (voltage-independent) pathway in the presence of β2c. Here, we performed more experiments to investigate about the voltage-dependency of Ca_V_2.2 channels with different β subunits (see new Figure 6). In cells expressing the alanine-substituted I-II loop of α1B with β2c subunit, we found that M_1_R inhibits the current mostly through the voltage-dependent pathway (new Figure 6F-H). Our new results support that disruption of PIP_2_ binding to the polybasic motif in I-II loop converts the channels to willing state which has low-affinity to Gβγ (p.11 the first paragraph).

Keum, D., Baek, C., Kim, D.I., Kweon, H.J., and Suh, B.C. (2014) J. Gen. Physiol. 144, 297–309.

– Figure 5F-G. Shows a comparison of Qon and tails currents measured by stepping to the reversal protntal to isolated Qon and then stepping back to a negative membrane potential (probably -80 mV) to measure the instantaneous current. They show that the ratio of Itail to Qon is reduced in WT CaV channels in the presence of b2c relative to b2a and b2c in the presence of the CaValpha alanine mutant. They suggest that this provides evidence that neutralizing the basic amino acids in CaV α within the I-II influences the channel open probability. They conclude that b2a supports channels with higher Po compared to b2c but there is a simpler explanation. Cav channels in the presence of b2c inactivate compared to those expressed with b2a. The authors use a 100 ms depolarization to drive activation which is long enough for CaV channels to inactivate, especially given that the deodorization is to +60 mV. The authors could generate scatter plots comparing Qon vs Itail and compared the slopes and/or use much shorter step depolarizations to limit the degree of inactivation.

Thank you for your interest in this result. We have performed more experiments for defining the status of Ca_V_2.2 channels in willing or reluctant. Especially, through the analysis of the voltage dependent activation curve of each channel, we could provide more clearer information for the variation of the channel state (Please see Figures 5F-G and 7F-G). Therefore, we decide to delete this data from the manuscript.

Reviewer #3 (Recommendations for the authors):[…]1) Introduction. The Introduction does not give credit for "first findings" from other labs, which inadvertently gives misdirected credit for earlier findings.

We thank the reviewer for providing us with valuable feedback. Based on our new results, we revised our hypothetical model and cited the corresponding early studies in the introduction section (p.4).

2) Issue of R and S gating: Figure 1. The imaging data nicely show that the tethered β-subunits do indeed associate with the plasma membrane. However, a major issue with the current tracings is that the working definitions of R and S putative PIP2 binding sites and their relationship to consequential gating patterns is not well defined in the literature. In Figure 1D, the red traces do not appear to show kinetic changes normal associated with R S transitions. The +DVSP experiment should metabolize PIP2 at both the R and S sites. However, no change in either current amplitude or kinetics is observed with palmitoylated beta2a, which is thought to block at least one PIP2 binding site but leave the other site available for PIP2 binding. The authors need to provide an explanation for this if they are going to pursue the R and S model of PIP2's actions.

We thank the reviewer for raising these important questions.

First, our data shown in +Dr-VSP experiments provide just the peak current traces evoked during the 5-ms test pulse because the Ca_V_2.2 channel was easily inactivated by the three-pulse test protocol (see the protocol in Figure 1—figure supplement 2). Because of this short test protocol, we usually cannot see the changes of gating kinetics during the S-R transition. However, our new data in Figure 7 show that mutation of the S site strongly shifts the voltage-dependent activation curve to the right direction. In addition, the current inactivation kinetics was changed by the mutation of the PIP_2_-binding sites in channels with cytosolic β subunit. The channels with β2a subunit did not show any changes in inactivation kinetics among the mutant forms probably because the S-site mutation alone is not enough for changing the inactivation kinetics.

Second, with Dr-VSP, the current suppression in Ca_V_2.2 channels with β2a is little but significantly stronger (~5-10% inhibition) than the control (-Dr-VSP) (Figure 1 and supplementary 1). We found that direct mutation of the S site greatly decreased the current density (Figure 7F). Based on these results, we assume that Dr-VSP cannot completely deplete all the PIP_2_ within the S site because of the high-affinity PIP_2_ binding of the site (this possibility is described in the Discussion section in the first paragraph on page 16). Again, we thank the reviewer for this inspiring comment.

3) GTP in the pipette. GTP was included in the pipette solution, which should tonically promote G-protein-mediated reluctant gating. However, the prepulse given in the DR-VSP experiment (Figure 1D) should knock off any tonic G-protein interaction with CaV2.2 converting them to willing gating. (The authors need to state how long the delay is between the prepulse to 120 mV and the following current recorded at +10 mV.) Consequently, it's possible that all the +Dr-VSP experiments are testing changes in PIP2 binding to the S site because the prepulse will overcome any PIP2 effects at the R site as Wu et al. 2002 has documented. The authors need to address the confounding issue of GTP in the pipette and/or provide data that could rule out some of these confounding issues. The authors should repeat the expt in Figure 1D with GDP-β-S in the pipette. This critical control should reveal if there is a G-protein effect contaminating the data. In revisiting this experiment, the authors should also plot G-V curves to determine whether a shift in activation occurs following prepulses when GDP-β-S is included in the pipette.

We thank the reviewer for raising this concern. As the reviewer suggest, we performed the experiments in the presence of the hydrolysis-resistant GDP analog GDPβD (1 mM) in the pipette solution. Since GDPβD inhibits the basal activation of G protein and the possible Gβγ-mediated suppression of CaV2.2 channels, we can eliminate the voltage-dependent current relief by the prepulse depolarization. With the same Dr-VSP activation protocol, we found that all CaV2.2 channels shows very similar PI(4,5)P2 sensitivities to those of experiments with 0.1 mM GTP (please see Author response image 1). Our new data clearly suggest that 0.1 mM GTP concentration in the pipette solution is not sufficient to suppress the CaV2.2 channels through the Gβγ-dependent pathway and thus the prepulse used to activate Dr-VSP did not change the PIP2 responses by affecting G-protein dissociation from the channels.

Surprisingly, no prepulse facilitation occurs for beta2a (Figure 1-suppl 2A); previously the authors did observe facilitation with a prepulse using the same concentration of GTP in the pipette as used in the current study (Keum et al. 2014; Figure 2+3). The authors need to address this inconsistency with their previous data.

We thank the reviewer for this comment. Keum et al. (2014) showed that current of Ca_V_2.2 channels with β2a was inhibited by a prepulse (Dr-VSP-mediated PIP_2_ depletion) in cells with DrVSP not without Dr-VSP. Our Figure 1-suppl 2A data also show that current amplitude of Ca_V_2.2 channels with β2a was not significantly different before and after the depolarizing pulse in cells without Dr-VSP. Park et al. (2017) previously confirmed that current amplitudes of Ca_V_2.2 channels with several β2c derivatives was not significantly different before and after the depolarizing pulse in cells without Dr-VSP. All these results indicate that the channels are not suppressed by any GTP-induced intracellular signals. (Please also see above #3 Q/A.)

Keum, D., Baek, C., Kim, D.-I., Kweon, H.J., and Suh, B.C. 2014. J. Gen. Physiol. 144, 297–309.

Park, C.G., Park, Y., and Suh, B.C. 2017. J. Gen. Physiol. 149, 261–276.

4) Experimental design for testing M1Rs. Figure 5 suppl2. In a number of the experiments, the currents are sampled 10 sec after Oxo-M application. Previously, the authors (Keum et al. 2014) showed that at 10 sec, inhibition could be overcome by prepulses, suggesting that Gbeta/γ was binding to the channels. The authors need to include traces from the time point when the current amplitude was measured. This experiment should be repeated with prepulses to determine how much of the inhibition is voltage-independent.

We thank the reviewer for constructive suggestion. According to the review’s comments, we performed new experiments to see the involvement of voltage-dependent and voltage independent pathways in the M_1_R-mediated Ca_V_2.2 channel suppression. Our results show that M_1_R inhibits the channels through both VD and VI pathways but with different portions. The results were presented in new Figure 6F-H and described in the Results section (p. 11).

Lastly, the authors should show the current-voltage relationships of control vs Oxo-M for the 4 conditions. This will also reveal the consequences of mutating the polybasic residues on the voltage-sensitivity of the channel gating.

We thank the reviewer for constructive suggestion. According to the review’s comments, we also measured the current-voltage relationships in those channels. The Results were presented in new Figure 6B. As the reviewer mentioned, the voltage-dependent activation curves were shifted to the right direction, suggesting that the voltage-sensitivity of all channel combinations was decreased after PIP_2_ depletion by M_1_R activation (also see the Results section in page 11).

Also, in Figure 5 suppl2 Oxo-M inhibits currents of beta2a-WT and beta2a-4A-mutant channels. From their paper Keum et al. (2014) in Figure 2 using the same conditions of collecting traces 10 sec after Oxo-M application, they found remarkable slowed activation kinetics as if they are looking at Gbeta/gaamma mediated inhibition. They found also that VSP does not decrease the current, suggesting that the phospholipid metabolized by Oxo-M stimulated signaling must somehow be inaccessible to the 5-phosphatase or it is not PIP2. In this manuscript to be consistent, the authors should include individual traces from the time courses shown in Figure 5-suppl 2.

As suggested, we show the current traces at the points in new Figure 6E. The results show that the current activation was similarly slowed down after M_1_R activation regardless of β isotype, suggesting that Gβγ-mediated voltage-dependent inhibition is involved in all the channel combinations during Oxo-M application (also see the Results section in page 11).

5) Data Interpretation. The interpretation of the data seem premature, in particular insufficient data are presented to assign locations for R vs S phospholipid binding sites. It is not obvious that all the experiments are looking at the same PIP2 binding site. For example, the authors contend that neutralizing 4 of the polybasic residues in the I-II loop removes a PIP2 binding site to stabilize CaV2.2 in a willing state. The authors also document the role of SH3-GK interaction and of lipid tethers of CaVbeta2 subunits in promoting lower sensitivity to PIP2 at the putative R site. There are additional basic residues in the α-helix that potentially participate in the binding of a second phospholipid. There are also polybasic regions in the CaVbeta2 N-terminus and HOOK regions that may be disturbed by mutating the 4 basic residues. The authors are best served by focusing on making the case for a phospholipid binding to the polybasic region critical for channel gating.

We thank the reviewer for pointing out new approaches. A recent Nature paper published during our revising period (Gao et al., 2021) indicates that there is a PIP_2_ binding site in S4_II_ domain of α1B. So, by mutating the amino acid residues in the site, here we examined the functional effect of the new PIP_2_-binding site on Ca_V_2.2 channel gating. As shown in new Figure 7, mutation of the new site in S4_II_ domain completely blocked the Dr-VSP-induced current suppression in channels with β2a. In addition, the mutant channel showed a very decreased voltage-sensitivity in the channel gating, suggesting that the PIP_2_-binding site in S4_II_ domain is responsible for maintaining the channels in the willing state. However, in mutant Ca_V_2.2 channels with β2c, Dr-VSP activation still significantly inhibits the current, suggesting that the channels with β2c interact with PIP_2_ through additional binding site. Then, we confirmed that when both the PIP_2_-binding sites including the site in S4_II_ domain and the polybasic motif in I-II loop were simultaneously mutated, the current inhibition by PIP_2_ depletion completely disappeared in the channels with β2c. Through the serious of experiments shown in Figure 5-7, we confirmed that the PIP_2_-binding site in S4_II_-domain is the S site and the polybasic motif in the I-II loop is the R site. Therefore, we confirmed that at least two PIP_2_-binding sites exist in the Ca_V_2.2 channels and they are differentially involved in the regulation of channel gating depending on the coupled Ca_V_ β subunits (please see new Figure 7 and the corresponding Results section on the page 11-12).

Page 10-line 1. The authors suggest that the membrane-anchored β-subunits somehow disrupt phospholipid binding to the polybasic binding motif in the α-helix of CaV2.2's I-II loop. They do not entertain the possibility that in certain subunit configurations, the α-helix may bind PIP2 (or other phospholipids) protecting it from metabolism- in other words unavailable to the voltage-activated membrane associated 5 π phosphatase (Dr-VSP). Lastly, in the discussion the authors suggest that the fatty acid tails of the lipidated Lyn11 constructs may insert into the membrane to compete with PIP2 binding to CaV2.2, similar to the proposed actions of palmitoylated beta2a (Heneghan et al., 2009; Suh et al., 2012). The authors should discuss whether the tether data fit this model. Lastly, the N-terminus region of the I-II loop forms a rigid α-helix which greatly influences channel activity (Vitko et al., 2008). The authors need to include these possibilities in the discussion.

We thank the reviewer for these constructive comments.

1) Previous reports showed that both N- and C-terminal region in the I-II loop of Ca_V_1.2 channels form the α-helix structure (Kaur et al., 2015, Vitko et al., 2008). However, the C-terminus, but not the N-terminus, was essential for the PM translocation of the I-II linker (Kaur et al., 2015). Thus, we think that, though both N- and C-terminal α-helix structures are involved in the regulation of channel activity, the polybasic motif on the C-terminal a-helix only influences the channels through the interaction with anionic phospholipids in the PM. This was described in the Discussion section (p.15, the first paragraph).

2) The structural analysis of Gao et al. (2021) showed that our “N-terminal length” data fit with the tether model rather than the conformational transition model. Thus, we revised all the sentences mentioning the possible model. The functional role of fatty acid tails of Lyn_11_ was also briefly described in the Discussion (p.14, the first paragraph).

Specific CommentsThe authors repeatedly claim PIP2 binding when they have shown that a phospholipid is involved, but not specifically PIP2. In this paper the α-helix phospholipid binding site may indeed be selective for PIP2, however, Kaur et al. (2015) presented data that phosphatidylserine also may bind to the homologous site in CaV1.2. The authors need to use the more conservative term phospholipid in more places rather than PIP2.

We thank the reviewer for constructive comments. As the reviewer suggest, we changed the description through the whole manuscript.

In the introduction the authors cite that Kaur et al. (2015) as demonstrating a homologous polybasic sequence in the I-II loop that is the site of PIP2-mediated reluctant gating. In rereading that paper, Kaur et al. conclude the site stabilizes channel gating (see abstract of their paper). They did probe gating changes but did not make any definitive conclusion about the existence of a reluctant site. Therefore, the authors should remove the last two sentences (found in lines 4-7) on page 4.

We thank the reviewer for constructive comments. We remove the two sentences (now the last part of the second paragraph on page 3).

Page 3; 2nd paragraph, 1st sentence. I believe the authors mean intracellular regulatory signals, not intercellular regulatory signals.

We thank the reviewer for correction. As suggested, we changed the word (p.3).

Page 4. 2nd paragraph. Please also add Richards et al. from the Dolphin lab and Miranda-Laferte et al., 2011, to the list of citations on the function of the HOOK region.

We thank the careful work of reviewer, and we apologize for omitting citations about the function of the HOOK region. As suggested, we added more citations (p.4, the first paragraph).

Figure 1A. Please state in the legend that the location of the β-subunit is based on CryoEM of Cav1.1, the skeletal muscle L-type channel, and β1a locations, but not β2 isoforms or an β-subunit from the CaV2 family. This is important because CaV1.1 gates quite differently than the CaV2 channel family. This may indicate that a β-subunit localizes to a similar but somewhat different position than in CaV2.2. Also please state where the AID region is in the I-II loop, i.e., near the N-terminus end of the I-II loop.

We thank the reviewer for helpful comments. However, in the middle of our revision period, human Ca_V_2.2 channel complex was solved by cryo-electron microscopy at resolution of 3.0Å (Gao et al., 2021). The structure show that Ca_V_ β subunit is located beside the domain II, which is consistent with our schematic diagram. So, the Gao’s paper was cited there instead (see Figure 1 legend).

Figure 1D. Explain the rationale for why the upper traces are measured over a 500 ms test pulse for visualizing inactivation while the lower traces are measured at ~ 10 ms to document changes in current amplitude following Dr-VSP. Why not use a long test pulse in both instances?

As the reviewer mentioned, we used different protocols for measuring the current inactivation and the PIP_2_ sensitivity in Ca_V_2.2 channels. For measuring the current inactivation, we need at least 500-ms test pulse. However, for measuring the PIP_2_ sensitivity, we have to apply three consecutive pulses: 10-ms test pulse, 120-mV depol, and then the second 10-ms test pulse. When we applied 500-ms test pulse in this consecutive protocol, the channels were strongly inactivated and the current generated by second test pulse was dramatically decreased even without Dr-VSP. Since we need the peak current amplitude only for examining the PIP_2_ sensitivity, we applied the short 10-ms test pulse. However, we measured the current inactivation by using the 500-ms test pulse in new Figure 7. Thus, they could know how the inactivation changes depending on the channel compositions now.

Figure 1-Suppl 2B looks as though the Y-axis is mislabeled. It should simply read % Inhibition by a prepulse.

We thank the reviewer for this correction. We change Y-axis in Fig-Suppl 2B to “% Inhibition by a pulse”.

Figure 2A- State which beta2 variant, beta2a, beta2c, or another were mutated and used in the results presented in Figure 2A.

We thank the reviewer for this question. The splicing variants of β2 type present the amino acid sequence differences only in the N-terminus. Therefore, when we deleted the N-terminal sequence in the β2 subunit as shown in Figure 2, there is no more variance in the β2 construct. However, as the reviewer’s suggestion, we specified it to β2a in the Result section only since it was constructed from β2a (p.6, the second paragraph).

Page 8. 3rd line from top of page. Together, our date further confirm…. This is an overstatement. Please change confirm to, "… are consistent with our model where…"

We thank the reviewer for the helpful comment. As suggested, we have revised the sentence (p.7, the last sentrence).

Combine Figure 4 suppl 2 and4 for easier comparison.

We thank the reviewer for this constructive suggestion for a modification of figures. We merged the Figures to new Figure 4-suppl 2.

Figure 4E. The Y-axis looks like the label should be normalized current remaining rather than inactivation. The same for PIP2 depletion curve. Whether this is a correct analysis of the curve, the plot needs some clarification.

As suggested, we modified the Y-axis to “Relative to Lyn-GK”.

Figure 4 suppl 3D. The X-axis is mislabeled and should read V1/2 activation NOT inactivation.

We thank the reviewer for this correction. As suggested, we corrected the X-axis.

Page 11. Last sentence of the 1st paragraph. This is an overinterpretation of their data. If the authors think the channels inactivate when both sites lose their phospholipid, they need to show the channels are stabilized in an inactivated state by plotting voltage-dependent inactivation graphs. If the authors don't have the data, they should state the channels are stabilized in a nonconducting state.

We thank the reviewer for constructive comments. As shown in new Figure 7F-G, the voltage dependent activation graph was greatly shifted to the right direction in the mutant channels regardless of β2 subtypes, suggesting that the channels without PIP_2_ coupling are almost nonconducting at +10 mV. Therefore, as the reviewer suggest, we changed the inactive state to nonconducting state in the manuscript and the Figure 8 (p.13, the first paragraph; p.16, the second paragraph).

[Editors' note: further revisions were suggested prior to acceptance, as described below.]

The manuscript has been improved but there are some remaining issues that need to be addressed, as outlined below:Reviewer #2 (Recommendations for the authors):The authors have addressed most of the comments in this revised manuscript. They have added new data, removed other more preliminary findings, and incorporated findings of recently published cryo-EM structure of hCaV2.2 (Gao et al., 2021). A few remaining comments:

We sincerely thank you for your valuable comments.

– The experimental rigor of this study is strong and the authors are experts in this important field. The manuscript does still lack clarity regarding the overarching goal/specific questions being addressed by the study. The additional first paragraph in this revised manuscript is helpful, but an illustrative model in the introduction could anchor the reader and bring the significance and impact of the studies into sharper focus. This would give important context to the impressive experiments and the results in this manuscript, it would highlight current gaps in knowledge being addressed, and it would also help underscore/appreciate precisely how the authors proposed model (Figure 8) advances our understanding of "the molecular basis of Cavb and PIP2 dependent regulation of CaV2.2 voltage-gated calcium channels".

We thank the reviewer for this constructive comment. According to this comment and Reviewer 3’s comments #1 and 2, we narrowed down the scope of this research from ‘the overall understanding of Ca_V_ β and PIP_2_ regulation of Ca_V_2.2 channel’ to ‘more specific PIP_2_ regulation of Ca_V_2.2 channel and modulation by Ca_V_ β subunits’. We thus changed the title based on our new findings as follow: Molecular basis of the PIP_2_-dependent regulation of Ca_V_2.2 channel and its modulation by Ca_V_ β subunits. As summarized in the Abstract, here we report evidence showing that there two distinct PIP_2_-interacting sites in Ca_V_2.2 and the two PIP_2_ interactions are differentially modulated depending on Ca_V_ β isotypes. Our model does not indicate whether it is S or R state (or domain) anymore, but precisely show how the coupled Ca_V_ β isotypes affect the PIP_2_ interaction with the two different binding sites. These findings provide new information to the Ca_V_2.2 channel studies.

– Could the authors be more explicit about how previously defined channel states that they refer to in the introduction and throughout (e.g. as detailed Wu et al., 2002 propose 6 distinct states which include 2 PIP2 binding domains "R" and "S") are incorporated into the authors final model (Figure 8).

We thank the reviewer for valuable suggestion. However, since the biophysical properties of Ca_V_ channel states are not fully understood yet (please also see reviewer 3-issue #1), we draught back our definition for the states of channel in the revised text. In our new model, we focused how the PIP_2_-interaction with two different binding sites on Ca_V_2.2 channels regulate the PIP_2_ sensitivity of the channels with different Ca_V_ β isotypes. We did not define the state of the channels, but clearly demonstrate why each channel complex shows different PIP_2_ sensitivity.

– The Results section is clearer and the subheadings are helpful. However, the first page of the Results section, in particular, is not effective in framing the rest of the report. Is not obvious why the Results section starts with a relatively long introduction to ".multiple types of CaV b subunit…." (lines 100-107). This could be more effective.

We thank the reviewer for helpful comments. We revised the first starting sentence for clarity and focusing (Line 99).

– As previously recommended. The authors should be careful when referring to distance between protein motifs if they are not actually measuring distance. There are still some instances of this in the Results section.

We thank the reviewer for raising this concern. As commented, we changed the word “distance” to “length” through the text to describe the phenomena correctly.

– Discussion. Consider removing the description of the concluding model in Figure 8, to the start of the Discussion section. Also, importantly, include statements about how the new model differs from, and how it is an advance over previous publications/models. More generally, the discussion could benefit from more explicit statements of what is new in this manuscript and how the author's proposed model helps to advance understanding.

We thank the reviewer for this constructive suggestion. As suggested, we moved the concluding description about our schematic model (Figure 8) to the start of the Discussion section (Line 335-) and also reorganized and revised the rest parts of the Discussion. We also pointed out the advance information in our new model in the Discussion section (Line 374-): actually, the regulatory effects of two distinct PIP_2_-binding sites provide new advance to the Ca_V_2.2 channel study.

– The manuscript should be read thoroughly for grammatical accuracy; this includes the abstract, introduction, results, and discussion.

We thank the careful work of reviewer. We apologize for lack of grammatical accuracy. We have checked the accuracy of the manuscript through the formal English Correction process.

Reviewer #3 (Recommendations for the authors):The revised manuscript by Park et al. is much improved in several ways. The requested details from the 3 Reviewers have added clarity to the manuscript. Additionally, Park et al. gave succinct responses to all of the reviewers' concerns. The description of Lyn11 being lipidated on vicinal amino acids gives new insight to potential mechanistic explanations for some of the data. Removal of the idea of the I-II linker somehow being pressed against the membrane by a tethered β subunit also appears to simplify data interpretation. The added data in figure 6 has greatly added to clarifying Gbeta-γ effects vs phospholipid effects during M1R stimulation. They show that at 10 sec of muscarinic stimulation, the majority of the palmitoylated CaVb2a-containing channels are modulated by a VD mechanism, most likely by Gbeta-γ binding to CaV2.2. The recent publication by Gao et al. 2021 of a CryoEM structure of CaV2.2/beta3/alpha2delta1 with a PIP2 molecule bound to the VSDII of CaV2.2 has contributed structural data that was used to interpret the biophysical data presented here, which also improves the paper. Additionally, in Figure 7, Park et al. have pivoted in their focus from the polybasic segment of the I-II linker to test the phospholipid sensitivity of channels with key residues in the VSD-II identified by Gao et al. as residues that interact with a phosphoinositol headgroup. The quality of the data is excellent. The significance of the findings are timely and extremely important for understanding the unique physiology of CaV2.2 channels.That said, a major shortcoming of the manuscript remains in the interpretation of the results, which all three Reviewers were concerned about in the first version of the manuscript. I remain concerned. These shortcomings fall under three problems and must be fixed, so that if published, the model does not create another unnecessary conflict in the field which then takes years to undo. While it is great that the Suh lab has reinterpreted their model based on their new data in Figure 7, they now have taken on not only the polybasic phospholipid binding site but also the VSDII PIP2 binding site with only scant data to support their definitive conclusions. In particular three issues stand out.Issue #1: Willing (S) and Reluctant (R) gating vs the predicted PIP2 "R" and "S" binding sites. I have reread carefully, Wu et al. 2002; Kauer et al. 2015; Gao et al., 2021, Dong et al. 2021 as well as Heneghan et al. 2009 in addition to work from the Suh lab. What seems clear is that CaV2.2 has multiple lipid binding sites that influence channel gating. The problem for all of us is that we are trying to identify the "R" and "S" binding sites that mediate "reluctant" vs "willing" gating and channel stability, first described by Wu et al. (2002). The movement of different parts of the channel may converge to create a signature pattern of gating. In this manuscript Park et al. are calling VD gating as "Reluctant Gating" and that it is the polybasic segment on the I-II linker that is responsible for VD gating. However, from my reading, Gao et al. 2021 favor a model where the PIP2 binding site on the VSD-II regulates voltage-dependent (VD) gating, not the polybasic segment. I quote from Gao et al's discussion, "The PIP2 binding site observed in this study may account for the voltage-dependent inhibitory modulation as it stabilizes the down conformation of the VSDII. It remains to be investigated whether a separate PIP2-binding site is responsible for the voltage-independent rundown reduction, a mechanism that may involve Gbg proteins." The problem is that Park et al. have come to the OPPOSITE conclusion. If I am reading this paper correctly, Park et al. have assigned the VDII PIP2 binding site as the "S" site and the polybasic site on the I-II linker as the "R" site with the "R" site responsible for VD gating. Intuitively, the Gao et al. and the newer Dong et al. CryoEM structures showing PIP2 stabilized a down state of the VSD-II should make the channel harder to open. Therefore, if that PIP2 is metabolized by M1R signaling, the VSDII should now move more easily with depolarization, which is a key characteristic of the original R site (Wu et al., 2002); Another way to think of this is that if the VSD-II site was responsible for "R" gating, its metabolism is predicted to remove the restraint on the up-movement of the VSDII and should give rise to more rapid activation kinetics and more current – NOT inhibition. Yet other labs that have studied N-current modulation by M1Rs observe current inhibition following activated M1R signaling, suggesting that another site – perhaps the polybasic segment of the I-II linker-- is the "S" site involved in N-current inhibition by Oxo-M. These two CryoEM structures are in major conflict with Park et al's model – if I am understanding it properly.The point I am making may seem long-winded but understanding CaV2.2's structure and gating is an extraordinarily important issue to NOT GET WRONG. What may be smartest for Park et al. to do is to back off of their conclusion and state the pros and cons of which site might be the "R" vs "S" phospholipid binding sites. Notably both the Gao et al. and Dong et al. were very careful about assigning "their PIP2" to a particular gating function, raising the issue that if CaV2.2 does have more than one PIP2 binding site, where is it? Neither CryoEM study found a phospholipid that bound to the polybasic site on the I-II linker. Moreover, the N-terminus of CaV2.2 was clipped off which may also alter channel structure and/or CaVbeta orientation and therefore stabilize/destabilize a particular channel construct. There is no need for Park et al. to assign a gating term (willing or reluctant) to either of the two sites. The paper will be much clearer if they call the I-II linker polybasic phospholipid binding site as just that and the VSDII site, the VSD-II PIP2 binding site, and nothing more: not a willing or reluctant site. At the very least, Park et al., need to clearly point out the different conclusion of Gao et al. with theirs; even better, they may want to reconsider their conclusions of their data. Indeed, my interpretation of the data in Figure 6 suggest that the 4A substitutions in the I-II linker do indeed result in a loss of M1R-mediated current inhibition due to phospholipid metabolism. If so, it seems that this is the "S" site.

We appreciate the reviewer for this in-depth considerations and insightful suggestion. We took the points seriously. We fully understand the reviewer’s concerns and agree to interpret our conclusion without defining the specific or conclusive channel state (e.g. S or R, willing or reluctant). Further studies may solve this important issue comprehensively.

Shortly, in the revised manuscript we simply described the channel states related with their PIP_2_ sensitivity. Since our results demonstrate for the first time that Ca_V_2.2 channel gating is distinctively regulated by two PIP_2_-regulatory sites, we think that the studies will still give an important message to the Ca_V_2.2 channel field.

Issue #2. What lipid(s) bind to the I-II polybasic binding site. Park et al. present a remarkably simplistic and simultaneously aggressive interpretation of their phospholipid data in calling the polybasic I-II linker site "THE other PIP2" binding site. I make this statement for the following conclusions. (1) Gao et al. and Dong et al. did not capture a PIP2 bound to the polybasic region of the I-II linker. (2) Kauer et al. found their data inconclusive as to whether the homologous polybasic segment on CaV1.2's I-II linker is a specific PIP2 binding site or a more general phospholipid binding site; and also were unsure whether the site regulates "S" or "R" gating. (2) Park et al. did not provide control experiments where other basic residues in this segment of the I-II linker (shown in Figure 5 Supplemental 1) are mutated to alanine to determine the specificity of their mutations vs others in controlling gating. Indeed, it's possible that disruption of any part of that segment will alter gating. (3) Furthermore, they have not shown that disrupting this site might alter a PIP2 binding site elsewhere. Imaging data only show low resolution membrane localization of membrane associated fluorescing reporters and don't serve to identify PIP2 binding sites on channel subunits. I point out that the more we learn about the cytoplasmic loops of CaV2.2, the more intimate and complicated their interactions appear to be, such that a point mutation in a charged residue at one location may have profound effects on multiple aspects of channel function far from that residue. Much more data need to be collected before we have a clear understanding of lipid effects on CaV2.2 gating. (4) We don't know even how many critical inner membrane lipid binding sites might exist in the CaV2.2 channel complex; collective findings suggest at least two, but do not rule out more. Therefore, the authors must denote the polybasic I-II linker site not as "the" PIP2 binding site but as a potential PIP2 binding site or simply a phospholipid binding site. Backing off this terminology does not detract in any way from the findings. Indeed historically, calcium channel biophysicists were very careful in interpreting DHP action on L-current with identifying DHP binding sites on L-channels and the same with G-protein interaction sites with CaV2 channels. Please replace "PIP2" binding site with "phospholipid" binding site when discussing the polybasic I-II linker.

We thank the reviewer for in-depth analysis and constructive critics. As you pointed out, we replaced the ‘PIP_2_ binding I-II linker site’ with ‘potential PIP_2_ binding site or (nonspecific) phospholipid binding site’ throughout the manuscript.

Issue #3. The interpretation of what PIP2 molecules/their metabolism may be doing to Gbeta-γ binding following M1Rs is way overstated on page 11. There are certainly other possibilities, for example, it is known that a number of kinases that phosphorylate CaV2.2 are also activated following M1R stimulation and have notable effects on N-channels. Park et al's interpretation could be correct, but they don't have the data to back up their model. The authors need to remove their comments about PIP2 and Gbeta-γ from the end of the paragraph on page 11.

We thank the reviewer for constructive critics. As suggested, we remove the last sentence (end of the 2^nd^ paragraph on page 11).

Other needed changes.1) Please include the GDP-betaS data in Figure 1 Suppl2. They would be helpful there. I am amazed that there is no tonic G-protein modulation, but Figure 6F also clearly supports this conclusion. Just to be a stickler about conclusions, the statement that just because 0.1 mM GTP has no effect on prepulse currents doesn't mean this concentration is physiological as the authors state early on in their rebuttal. Just let the data stand in this case.

We thank the careful work of reviewer. As suggested, we have included the GDP-β-S data in Figure 1 Suppl2 and briefly described the results in Line 129-.

2) Figures1A, 2A, 3A, 4A diagrams show Lyn 11 monolipidated, but in the discussion the authors comment that Lyn 11 is myristoylated (presumably on G3 and palmitoylated on C4). This fact changes dramatically the interpretation of the Lyn11 construct data. Please state this when first using Lyn 11 constructs. Also the authors state that C3 and C4 are of the Lyn11 construct are lipidated. Please correct the C to a G in the discussion text.

We really thank the reviewer for this correction. Lyn_11_ is myristoylated on G2 and palmitoylated on C3. We modified these sentences in Results and Discussion section (Lines 108-109, 368369).

3) For clarity, please add another tail to all Lyn11 schematics in Figures1A, 2A, 3A, 4A.

We thank the careful work of reviewer. As suggested, we added lipid tail to all Lyn_11_ schematics in Figures1A, 2A, 3A, 4A.

4) Also please site a reference that shows this simultaneous dual lipidation of Lyn 11.

As suggested, we cited Resh’s paper (Line 109).

5) Figure 3B needs a sample size.

As suggested, we add sample size in figure 3 source data.

6) The concerns raised by Reviewer 2 do not seem adequately addressed. In particular, Reviewer #2 commented there are many other explanations for the changes in currents from the RF and RFC constructs, yet the authors contend all the changes are due to the β subunit's closeness to the plasma membrane. The authors need to back off of this conclusion.

As the reviewer’s concern, we agree that rapamycin itself might have some non-specific effects on especially ca^2+^ channel current. Therefore, though we think that the RF and RCF results are still supportive to the conclusion, we revised some sentences and deleted the concluding remark from the Results section (1^st^ paragraph of page 8).

In summary, Park et al. have taken on attempting to (1) define CaV2.2's "R" and "S" PIP2/phospholipid binding sites, (2) describe gating associated with different PIP2 binding sites, (3) illustrate the functional effects of vicinal lipidation of CaVbeta subunits on PIP2 binding to CaV2.2 and consequent gating and modulation, and lastly (4) attempt to define the functional relationship between PIP2 and Gbeta-γ with limited numbers of control experiments for each of these 4 important aspects of CaV2.2. I remain uncomfortable with the conclusions in this manuscript without further major changes.

We really thank the reviewer for providing us with very valuable feedback. As described above, we backed down our definition about the point #1. About (4) the Gβγ regulation, we agree with your critic and deleted the definitive conclusion from the section. Actually, we have been working on this voltage-dependent regulation of Ca_V_2.2 channels separately, thus the description was added without much supporting data. Overall, we have tried to narrow down the aims and scopes in the revised version. Thank you for your consideration.

[Editors' note: further revisions were suggested prior to acceptance, as described below.]

The manuscript has been improved but there are some remaining issues that need to be addressed, as outlined below:The reviewers agree that the revised manuscript describes beautifully performed experiments, but that it still suffers from a lack of focus and insufficient framing. Specifically, what mechanistic questions were the experiments designed to address? How do the results bear on those questions? What alternative explanations are also compatible with these results? What future experiments would help to eliminate these alternatives? Below, I have tried to give examples related to these points.1. The authors note that they are not addressing the relationship to the R and S functional states, but the introduction still starts – as it did in previous versions, by discussing R and S states as they relate to PIP2 modulation of CaV2 channels. This sets expectations that the authors will develop a consensus new model.

We thank you for the acute professional comment. So far, the molecular identity of the R and S state of Ca_V_ channels is not yet clearly defined. In our studies with various mutant Ca_V_2.2 channels, we found that the data doesn’t conform to specify the state of channels. According to the reviewers’ suggestion, we deleted the sentence mentioning the two states from the Introduction section. Our study more focused on the functional roles of two potential PIP_2_ interacting sites on Ca_V_2.2 channels and their differential modulatory mechanisms in channels with different Ca_V_ β isoforms. This was summarized in our model figure.

2. A criticism of the previous version of the manuscript was the use of "distance" based on the number of amino acid residues. The revised version has replaced "distance" with "length" (e.g., Figure 4), but the implication is the same: that the length defined in this way is an indication of physical distance. Although this may be correct, there are also other possibilities (e.g., differences in torsional rigidity) which should be discussed.

We appreciate the reviewer’s instructive opinion. We agree with the reviewers that there are uncertainties in the explanation for the differential regulation of channels by linker length. It is challenging to fully address the mechanism by this study. As the reviewer’s suggestion, we have added another possibility in the Discussion section (p.15, the last part).

3. The muscarinic modulation experiments do not demonstrate clear pathway separation. These experiments need to be designed to isolate the effects of PIP2 metabolism from Gbeta-γ binding to CaV2.2.

We thank the reviewers for this constructive criticism. To clarify this point, we have performed two new experiments and the results were described in the Results section (Figure 6. Suppl1 and Figure 6. Suppl2).

1) First, by using the Gβγ-insensitive chimeric Ca_V_2.2 construct α1C-1B we isolated the PIP_2_dependent regulatory pathway from the M_1_R/Gβγ-mediated modulation of Ca_V_2.2 channels as shown in Figure 6 Suppl1. The results showed that muscarinic inhibition was similar in cells expressing either α1C-1B WT or α1C-1B 4A channels with the PM-anchored β2a, but much stronger in α1C-1B WT channels with the cytosolic β2c compared to α1C-1B 4A channels with β2c, which was consistent with the results obtained by M_1_R activation in the WT in α1B channels. These results support that the functional effects of polybasic motif on PIP_2_ sensitivity in the M_1_R-meidated of Ca_V_2.2 modulation were independent from the Gβγ-mediated signaling (p.11, in the second paragraph).

2) We also applied M_2_ (G_i_-coupled) muscarinic receptor (M_2_R) to examine the functional effects of Gβγ-mediated channel regulation on the polybasic motif-mediated modulation of Ca_V_2.2 channels without the possible PIP_2_ depletion-mediated effects as seen in Figure 6 Suppl2. The results showed that M_2_R activation strongly inhibited the currents in both WT and 4A α1B regardless of the coupled β2 isotype. In Ca_V_2.2 channels with the β2a subunit, recovery from Gβγ-mediated inhibition by prepulse (PP) were the same in WT α1B and 4A α1B. In Ca_V_2.2 with the β2c subunit, however, there was much stronger recovery of Gβγmediated inhibition in α1B 4A than α1B WT. Therefore, the recovery pattern from Gβγmediated inhibition was very similar in M_1_R- and M_2_R-induced modulation of Ca_V_2.2 channels (Figure 6 and Figure 6 Suppl2). These results support that the polybasic motif within the I-II loop suppresses the Gβγ-mediated modulation of Ca_V_2.2 channels with the cytosolic β2c subunit (p.12).

4. For the proposed I-II linker phospholipid binding site in the I-II loop C-terminus, it should be determined whether mutation of other arginine residues in the distal region does not alter phospholipid actions on CaV2.2 gating to support the idea that a specific binding site has been identified.

We thank the reviewer for constructive criticism. For this, we have constructed two new mutant α1B DNAs.

First, we removed two arginine residues (R476 and R477) in the distal region of the polybasic motif by replacing with alanine (α1B R476,477A) as shown in new Figure 5 Suppl3. In cells expressing α1B R476,477A with the membrane-anchored β2a, we did not detect any significant differences in current inactivation and PIP_2_ sensitivity compared to WT α1B. However, in cells expressing α1B R476,477A with cytosolic β2c, the inactivation rate was slower and the PIP_2_ sensitivity was weaker than in WT α1B with β2c, which were similar to the responses of 4A α1B with β2c (p.10, the last paragraph).

Second, we also constructed a α1B R465,466A by replacing two arginine residues (R465 and R466) within the polybasic motif (R465, R466, K469, and R472) with alanine as seen in Figure 5 Suppl3. The current inactivation and PIP_2_ sensitivity of α1B R465,466A and WT α1B with membrane-anchored β2a did not significantly differ and were similar to that of α1B R476,477A with β2a. Moreover, α1B R465,466A with cytosolic β2c exhibited slower inactivation and weaker PIP_2_ sensitivity than WT α1B with β2c, which was similar to the responses of α1B R476,477A and 4A α1B with β2c (p.10, the last paragraph).

Both results demonstrate that several basic residues at the end of the I-II loop broadly interacts with membrane PIP_2_ in Ca_V_2.2 channels, supporting that, as the reviewer’s assumptions, the polybasic motif in the I-II loop is more like a nonspecific phospholipid binding site.

5. There also need to be additional control mutations in VSD-II to demonstrate that only those in the proposed binding site for the PIP2 headgroup affect the actions of PIP2.

We thank the reviewer for this constructive comment. As suggested, we removed two other arginine residues (R578 and R581) within the S4_II_ by replacing with alanine (α1B R578,581A) in new Figure 7 Suppl1. Although α1B RA/KA completely abolished the PIP_2_ sensitivity in channels with membrane-anchored β2a, there was no significant difference in PIP_2_ sensitivity between WT α1B and new α1B R578,581A in cells with β2a. In channels with β2c, there was also significant PIP_2_ depletion-mediated inhibition in α1B R578,581A as like α1B. On the basis of these results, we speculate that the arginine residues (R578 and R581) within S4_II_ are not involved in the PIP_2_ regulation of Ca_V_2.2 channels though these residues are adjacent to the proposed PIP_2_ binding site (R584 and K587). These results further demonstrate that Ca_V_2.2 complexed with any β2 isotype interact with membrane PIP_2_ through a specific binding pocket (R584 and K587) within S4_II_ domain (p.13).

6. The previous version of the manuscript suggested that there were two binding sites important for the actions of PIP2 on CaV2.2, whereas the current version suggests there are three. Is it possible that there are only two binding sites, and that PIP2 has two different actions at this one site depending on the state of the channel?

We thank the reviewer for this comment. In this manuscript, we suggested that a single α1B subunit has two binding sites for membrane phospholipids including PIP_2_. In our previous studies, we showed that the HOOK region of Ca_V_ β subunits could interact with membrane lipids through a nonspecific electrostatic interaction (JGP, 2017). We do not exclude the possible extra-interactions between the membrane phospholipids and the auxiliary subunits of Ca_V_ channel, but those interactions are not the aims of the present study. Since the PIP_2_ sensitivity of Ca_V_2.2 is different between channels coupled with β2a and β2c subunits, though both β2 subunits have exactly the same HOOK region, we think that the HOOK interaction does not affect the differential PIP_2_ regulation of α1B subunit.

Your comment for the possibility that ‘PIP_2_ has two different actions at one site depending on the state of the channel’ is very intriguing. However, our data suggest that two distinct PIP_2_ molecules (or one PIP_2_ and one phospholipid) are involved in the coupling with the two binding sites. (1) Cleavage of a single 5-P from three phosphate groups of PIP_2_ triggers the two different actions of PIP_2_ in a single channel with β2c. The possibility that a single 5-P of PIP_2_ simultaneously interacts with two independent binding sites of α1B is very low and not seen in other channels. (2) PIP_2_ sensitivities of the two potential binding sites were always fixed as ~5% for the S4_II_ and ~25% for the polybasic motif, suggesting that there seems no cross-talk in the regulation between the two binding sites. (3) Similarly, if the PIP_2_ binding pocket in S4_II_ is important for a specific PIP_2_ binding, mutation of the site should also abolish or decrease the PIP_2_ sensitivity of I-II loop interaction as well as S4_II_. However, our data showed that the PIP_2_ sensitivity of the polybasic motif was constantly conserved in every experimental condition with mutant S4_II_. Finally, the distal polybasic residues (R578 and R581) of the I-II loop also affects the PIP_2_ sensitivity, supporting the possibility that polybasic region of the I-II loop broadly interacts with membrane phospholipids through nonspecific electrostatic association.

Reviewer #2 (Recommendations for the authors):The new focus of this revised manuscript is to describe the molecular basis of β-subunit control of PIP2 inhibition of CaV2 channels.The authors' main findings are 1. That there are 2 binding sites for PIP2 on I-II linker and in the domain II S4. The authors and others have reported previously that the β subunit can modulate channel sensitivity to PIP2 including showing that membrane-linked and non-membrane-linked β subunits have differential effects on PIP2 action. In this study, the authors extend these findings, motivated by recent cyro-EM studies of PIP2 binding to CaV2.2 channels. They conclude that Cavb subunits that are membrane-anchored prevent the action of PIP2 via I-II site. But β subunits have no effect on PiP2 action via domain II S4.A major issue with the previous versions of this manuscript was the lack of clarity about how the authors' study and models of PIP2 action related to previous functional work that defines (among other findings) R and S channel states. In their response, the authors have removed references to this model (except in the introduction – see below). This narrows the focus of the work and limits the impact/broader relevance of the study.The technical quality of the work is not in question, it is very high and rigorous. The authors have contributed functional data showing the role of the domain II S4 site defined in cryo-EM studies of CaV2.2 and shown its lack of CaVb subunit dependence. They have also refined our understanding of the interaction between Cavb, PIP2 and the I-II domain of CaV2.2.However, it remains difficult to extract the broader impact of this study.

We thank the reviewer for constructive criticism. Our studies have shown that subcellular localization of the Ca_V_ β subunit plays a critical role in determining the PIP_2_ sensitivity of Ca_V_ channels. Ca_V_2.2 channels coexpressed with a membrane-localized β subunit, such as β2a or β2e, exhibit low PIP_2_ sensitivity, whereas channels with a cytosolic β subunit, such as β2b, β2c or β3, exhibit high PIP_2_ sensitivity. However, the underlying mechanisms for the differential regulation of Ca_V_2.2 channel gating by PIP_2_ depending on the subcellular localization of Ca_V_ β subunits remains unclear. Our studies verified here (1) where are the PIP_2_ binding sites on a1B subunit, (2) what are the effects of PIP_2_ binding to the sites on channel gating, and (3) how the β subunit regulates the interaction between binding sites and membrane PIP_2_ depending in different cytoplasmic location.

We think that our studies will give a significant implication to the people working on every Ca_V_ channel and other PIP_2_-sensitive ion channels. We also broaden the knowledge about the modulatory mechanisms of Ca_V_2.2 channels by muscarinic receptor-mediated signaling pathways.

The authors note that they are not addressing the relationship to the R and S functional states, but their introduction still starts – as it did in previous versions, discussing R and S states as they relate to PIP2 modulation of CaV2 channels. This sets expectations that the authors will develop a consensus new model.

We thank you for the constructive comment. As mentioned above, our results with various mutant Ca_V_2.2 channels did not clearly indicate the channel states. Therefore, as the reviewers’ concerns, we removed the sentence mentioning two states of Ca_V_ channel in the introduction section. Here we focused on the functional role of PIP_2_ regulatory sites on Ca_V_2.2 channels with different Ca_V_ β subunit.

Other commentsL.175. See previous review. The authors are still implying distance measurements "…functional role of length between lipid anchor and GK domain…". In this context length and distance are equivalent. Also, see L.197 and several other instances including in the conclusions.

We appreciate the reviewer’s instructive opinion. As mentioned above, we agree with the reviewer that there are uncertainties in the explanation for the differential regulation of channels by linker length. As the comments of reviewer #1, we have described another possibility in the Discussion section (p.15, the last part).

Reviewer #3 (Recommendations for the authors):This is the third version of the Park et al. manuscript entitled, "Molecular basis of the PIP2-dependent regulation of CaV2.2 channel and its modulation by CaV β subunits". Previous studies from several labs have provided data suggesting that PIP2 has 2 functional effects on CaV2.2. These effects are thought to be due to PIP2 binding to two unidentified sites on CaV2.2. The initial manuscript was submitted prior to the publications of Gao et al. (2021) and Dong et al. (2021), both of which have provided high-resolution crystal structures of a CaV2.2/ alpha2delta/ β complex with a PIP2 head group bound to Domain II's voltage-sensing domain (VSDII) and its fatty acid tails interacting with S5 and S6 pore helices. Consequently, Park et al. have pivoted their manuscript to not only clarify their original findings surrounding the role of lipidated β subunits in altering PIP2 sensitivity of CaV2.2 currents and the importance of the polybasic residues in the I-II linker but now also to probe whether a putative PIP2 head group bound to VSDII influences channel gating and/or kinetics, which would indicate that VSDII is one of the functionally important PIP2 binding sites. The authors contend their re-submission has 4 major findings: (1) The length of an amino acid tether, bound to a β subunit, alters CaV2.2 currents only if the tether is short. (2) The vicinal myristylated and palmitoylated residues of the Lyn domain of the tether mimic vicinal palmitoylation of CaVb2a. They hypothesize that in both cases the lipids may compete with PIP2 for binding to the "S" site on CaV2.2, which has not been identified. (3) Polybasic residues in the I-II linker of CaV2.2 may be a nonspecific phospholipid binding site that will bind PIP2 to affect channel gating and is the site of competition with CaVbeta2a's palmitoyl groups. (4) In this re-submission, the authors present new experiments testing whether the recently discovered PIP2 interaction site on the VSDII may be a second PIP2 binding site.

We sincerely thank you for your valuable comments.

While the data in Figures1-4 provide substantial data for a model where vicinally lipidated β subunits compete with PIP2 binding to CaV2.2, the experiments supporting points 3 and 4 have experimental design problems and only superficially probe the putative PIP2 binding sites with few control experiments. For example, in their initial submission, Park et al. probed whether polybasic residues in the channel's I-II linker may be a PIP2 binding site similar to the phospholipid-binding site described for CaV1.2 by Kaur et al. (2015) and also the site disrupted by palmitoylated CaVb2a. They mutated a subset of the basic residues present in the distal region of the I-II linker and showed changed gating. However, no other mutations of basic residues were made, so it is not clear how specific these residues are for PIP2 binding, nor whether PIP2 may still have been bound, but the I-II linker changed its orientation due to the neutralization of charge resulting in changed gating rather than a loss of PIP2. Additionally, they document some differences in response to muscarinic stimulation, yet their experimental design does not separate G-protein binding to CaV2.2 from PIP2 metabolism, making data interpretation extremely challenging. Indeed, their WT channel data (Figures 6B and E) don't fit the kinetic changes of CaV2.2 previously observed with Gq mediated PLC activation, indicating that much of what they are examining is not the result of PIP2 metabolism. While Park et al. seems to focus on whether they have changed G-protein mediated modulation. However their data indicate that it is not that b2a is more sensitive to G-protein modulation, but rather it is that b2a-containing channels are insensitive to PLC breakdown of PIP2. Moreover, insufficient data are presented to make a conclusion about the relationship between the VSDII and PIP2 to determine whether this PIP2 mediates one of the previously described actions of PIP2 on CaV2 gating.

We appreciate the reviewer for these in-depth considerations and insightful suggestion.

1) As reviewer’s comments, we mutated a subset of the basic residues present in the distal region of the I-II linker and measured the channel gating.

First, we removed two arginine residues (R476 and R477) in the distal region of the polybasic motif by replacing with alanine (α1B R476,477A) as shown in new Figure 5 Suppl3. In cells expressing α1B R476,477A with the membrane-anchored β2a, we did not detect any significant differences in current inactivation and PIP_2_ sensitivity compared to WT α1B. However, in cells expressing α1B R476,477A with cytosolic β2c, the inactivation rate was slower and the PIP_2_ sensitivity was weaker than in WT α1B with β2c, which were similar to the responses of 4A α1B with β2c (p.10, the last paragraph).

Second, we also constructed a α1B R465,466A by replacing two arginine residues (R465 and R466) within the polybasic motif (R465, R466, K469, and R472) with alanine as seen in Figure 5 Suppl3. The current inactivation and PIP_2_ sensitivity of α1B R465,466A and WT α1B with membrane-anchored β2a did not significantly differ and were similar to that of α1B R476,477A with β2a. Moreover, α1B R465,466A with cytosolic β2c exhibited slower inactivation and weaker PIP_2_ sensitivity than WT α1B with β2c, which was similar to the responses of α1B R476,477A and 4A α1B with β2c (p.10, the last paragraph).

Both results demonstrate that several basic residues at the end of the I-II loop broadly interacts with membrane PIP_2_ in Ca_V_2.2 channels, supporting that, as the reviewer’s assumptions, the polybasic motif in the I-II loop is more like a nonspecific phospholipid binding site.

2) As reviewer’s comments, our experimental design does not separate G-protein binding to Ca_V_2.2 from PIP_2_ metabolism. To clarify this issue, we perform two different experiments (new Figure 6 Suppl1 and Figure 6 Suppl2).

First, by using the Gβγ-insensitive chimeric Ca_V_2.2 construct α1C-1B we isolated the PIP_2_dependent regulatory pathway from the M_1_R/Gβγ-mediated modulation of Ca_V_2.2 channels as shown in Figure 6 Suppl1. The results showed that muscarinic inhibition was similar in cells expressing either α1C-1B WT or α1C-1B 4A channels with the PM-anchored β2a, but much stronger in α1C-1B WT channels with the cytosolic β2c compared to α1C1B 4A channels with β2c, which was consistent with the results obtained by M_1_R activation in the WT in α1B channels. These results support that the functional effects of polybasic motif on PIP_2_ sensitivity in the M_1_R-meidated of Ca_V_2.2 modulation were independent from the Gβγ-mediated signaling (p.11, in the second paragraph).

We also applied M_2_ (G_i_-coupled) muscarinic receptor (M_2_R) to examine the functional effects of Gβγ-mediated channel regulation on the polybasic motif-mediated modulation of Ca_V_2.2 channels without the possible PIP_2_ depletion-mediated effects as seen in Figure 6

Suppl2. The results showed that M_2_R activation strongly inhibited the currents in both WT and 4A α1B regardless of the coupled β2 isotype. In Ca_V_2.2 channels with the β2a subunit, recovery from Gβγ-mediated inhibition by prepulse (PP) were the same in WT α1B and 4A α1B. In Ca_V_2.2 with the β2c subunit, however, there was much stronger recovery of Gβγmediated inhibition in α1B 4A than α1B WT. Therefore, the recovery pattern from Gβγmediated inhibition was very similar in M_1_R- and M_2_R-induced modulation of Ca_V_2.2 channels (Figure 6 and Figure 6 Suppl2). These results support that the polybasic motif within the I-II loop suppresses the Gβγ-mediated modulation of Ca_V_2.2 channels with the cytosolic β2c subunit (p.12).

3) We agree that insufficient data are presented to make a conclusion about the relationship between the VSD_II_ and PIP_2_ to determine whether this PIP_2_ mediates one of the previously described actions of PIP_2_ on Ca_V_2 gating.

As suggested, we removed two other arginine residues (R578 and R581) within the S4_II_ by replacing with alanine (α1B R578,581A) in new Figure 7 Suppl1. Although α1B RA/KA completely abolished the PIP_2_ sensitivity in channels with membrane-anchored β2a, there was no significant difference in PIP_2_ sensitivity between WT α1B and new α1B R578,581A in cells with β2a. In channels with β2c, there was also significant PIP_2_ depletion-mediated inhibition in α1B R578,581A as like α1B. On the basis of these results, we speculate that the arginine residues (R578 and R581) within S4_II_ are not involved in the PIP_2_ regulation of Ca_V_2.2 channels though these residues are adjacent to the proposed PIP_2_ binding site (R584 and K587). These results further demonstrate that Ca_V_2.2 complexed with any β2 isotype interact with membrane PIP_2_ through a specific binding pocket (R584 and K587) within S4_II_ domain (p.13).

Lastly from the crystal structure, the distal I-II linker is close to the VSDII raising the possibility that a single PIP2 residue may interact with both the VSDII and the I-II linker. Amongst all these issues, the authors also bring up PIP2 interaction with the HOOK region of CaVbeta subunits as well as with the N-terminus of CaVbeta2e. Altogether, despite the beautiful current recordings as well as the nice diagrams, it became very difficult to sort out how these observations fit a model of lipid interactions with the CaV2.2 channel complex. While the authors have provided one model to explain their data, they have not ruled out others that could also explain their data.

We thank the reviewer for this constructive comment. Based on the crystal structure, your comment for the possibility that a single PIP_2_ has two different actions at one site depending on the state of the channel is very intriguing. However, our data suggest that two distinct PIP_2_ molecules (or one PIP_2_ and one phospholipid) are involved in the coupling with the two binding sites. (1) Cleavage of a single 5-P from three phosphate groups of PIP_2_ triggers the two different actions of PIP_2_ in a single channel with β2c. The possibility that a single 5-P of PIP_2_ simultaneously interacts with two independent binding sites of α1B is very low and not seen in other channels. (2) PIP_2_ sensitivities of the two potential binding sites were always fixed as ~5% for the S4_II_ and ~25% for the polybasic motif, suggesting that there seems no cross-talk in the regulation between the two binding sites. (3) Similarly, if the PIP_2_ binding site in S4_II_ is critical for a specific PIP_2_ binding, mutation of the site should also abolish or decrease the PIP_2_ sensitivity of I-II loop interaction as well as S4_II_. However, our data showed that the PIP_2_ sensitivity of the polybasic motif was constantly conserved in every experimental condition with mutant S4_II_. Finally, the distal polybasic residues (R578 and R581) of the I-II loop also affects the PIP_2_ sensitivity, supporting the possibility that polybasic region of the I-II loop broadly interacts with membrane phospholipids through nonspecific electrostatic association.

As your comments, there would be more extra-interactions between the membrane phospholipids and the auxiliary subunits of Ca_V_ channel, but those interactions are not the aims of the present study. Moreover, though all β2 isoforms have exactly the same HOOK region, the PIP_2_ sensitivity is different in Ca_V_2.2 channels coupled with β2a and β2c subunits. Therefore, we think that the HOOK interaction does not contribute a substantial role in the differential PIP_2_ regulation of α1B subunit.

[Editors' note: further revisions were suggested prior to acceptance, as described below.]

Your revised manuscript has now been seen by two of the original three reviewers. They agree that the manuscript has been greatly improved and presents important new results. However, one of the reviewers had suggestions for editorial changes, which are detailed below. The only significant issue raised by that reviewer was in regard to the data presented in Figure 6 (see below) and that reviewer argued that Figure 6 should be removed and replaced by Figure 6-S1 and Figure 6-S2. If you are willing to make this and the other suggested editorial changes, your manuscript will hopefully be acceptable without further review.

We thank the editor for the valuable feedback. As commented, we removed the original Figure 6 and changed the Figure 6-Suppl1 into new Figure 6. Considering the research flow, the previous Figure 6-Suppl2 was changed into new Figure 6-Suppl 1.

Detailed Reviewer Comments:– Line 41. Please include Wu et al. 2002 to acknowledge that they were the first to show that PIP2 is necessary for voltage gating of CaV2.1 and 2.2.

We thank the reviewer for this comment. As suggested, we included Wu’s paper (line 40 on page 3).

– Line 72, please site work from the Colecraft lab on CaVbeta subunits and others in addition to the Suh and Hille labs.

We thank the reviewer for this comment. As suggested, we cited Takahashi’s paper from the Colecraft lab (line 68 on page 4).

– Line 228. The sentence, "…This suggested that incremental increases in linker length lead to a decrease in channel gating…", may be easier to understand if it is changed to "This suggested that incremental increases in linker length lead to a decreased voltage-sensitivity.

We thank the reviewer for the suggestion. We have revised the sentence (line 224 on page 9).

– Page 10 lines 277-279. This is an important conclusion, but the sentence is awkward/not grammatically correct. The sentence may be clearer if expressed as: More importantly, our data indicate that PIP2 interacts with the polybasic motif when CaV2.2 is expressed with cytosolic b-subunits but not when expressed with lipidated membrane-anchored b-subunits.

We thank the reviewer for careful correction. As suggested, we modified the sentence (the last part of the second paragraph on page 10).

– Figure 5-S3 D. Seems weird that the VSP has no effect when the model predicts that there is another PIP2 bound to the channel that in theory should be metabolized by activating the phosphatase. The authors are arguing that the 5% inhibition is due to the 2nd PIP2 being metabolized. However, the model by Wu et al. would suggest they should see enhancement with the 2nd PIP2 metabolized, not inhibition. Some explanation of this discrepancy would be helpful.

We thank the reviewer for the valuable comments. So far, the model by Wu et al. made in oocyte systems is not consistent with our data obtained from tsA201 cells. The biggest difference is that the P/Q-type Ca_V_2.1 current were inhibited by additional 20 μM PIP_2_ application at +10 mV. (Thus, the currents could be enhanced by PIP_2_ depletion as the editor’s comment. The authors did not test the direct effects of PIP_2_ depletion on Ca_V_2.1 channel, instead they showed that the currents were inhibited by ~60% after PLC-coupled LHRH receptor stimulation in the paper; where they did not see any current increase by receptor mediated PIP_2_ hydrolysis even at +10 mV). On the other hand, our N-type Ca_V_2.2 currents were inhibited by endogenous PIP_2_ depletion at +10 mV. We detected only current inhibition by PIP_2_ depletion at every experimental condition. This difference between two channels might be due to the difference in cell systems used (oocytes vs tsA201 cells), experimental approaches (PIP_2_ addition vs PIP_2_ depletion), or phosphorylation status of the Ca_V_ channels (the phosphorylation reaction is relatively higher in tsA-201 cells). As previous reviewers’ comments, we agree that the molecular identity of R and S states of N-types channels were not clearly defined yet and could be slightly different from that of P/Q-type channels. In our original manuscript, we have already omitted mentions about the two states of channels and the different phenotypes between channel types.

– Page 11, lines 290-292. Changes in gating with additional RA mutations don't support their conclusion about the nonspecificity of lipid binding to the I-II linker. They didn't probe the effects of binding by other lipids to these mutant channels. Rather, their data further characterize which amino acids may participate in binding phospholipids/PIP2. Thus, this sentence should be removed.

We appreciate the reviewer’s instructive suggestion. We deleted the sentence for more clarity (end of the first paragraph on page 11).

– Figure 6 is fraught with several signal transduction/experimental design issues outlined here. First, muscarinic receptors, when transfected into HEK/TSA cells overexpress, and consequently are notoriously promiscuous in their coupling to signal transduction cascades, complicating data interpretation. In the 1990s the Hille lab published several papers looking at native ca^2+^ channel modulation by various receptor signaling cascades. Notably, Shapiro et al. (1999) presented data showing that the membrane-delimited pathway is carried primarily by M2Rs and that when this receptor is knocked out, no other muscarinic receptors, (e.g, the M1Rs) took their place in neurons.Whereas in this manuscript a recombinant system seems to exhibit M1R and M2R promiscuity in signaling. Second, Shapiro et al's data show that the diffusible 2nd messenger pathway, now known to be due to PIP2 metabolism, takes ~80 sec to reach full effect. Here, the data in Figure 6 are all collected 10-20 seconds after the application of Oxo-M, which would miss much of the PIP2 modulation of CaV2.2. Third, the previous metabolism of the 2nd PIP2, bound to channels, increases voltage sensitivity causing a negative shift in G/Gmax curves, seen best at negative test potentials, yet negative test potentials were not monitored independently of G-protein effects. Together the issue of promiscuous muscarinic receptor coupling to signaling cascades as well as issues with timing and optimal voltage protocols, make Figure 6 results meaningless. I strongly encourage the removal of Figure 6 in favor of keeping the better-designed expts in Figure 6-S1 and S2.

We thank the reviewer for the valuable comments. As suggested, we remove Figure 6 and change the original Figure 6-Suppl1 into new Figure 6. Considering the research scheme, the original Figure 6Suppl2 was changed into new Figure 6-Suppl 1.

– Figure 6-S1A. Please show in the schematic of the chimeric which loops and domains are 1C vs 1B with different colors. I may have missed it but, please include a description of the alpha1B-1C chimera either in the Methods or Results section. If this chimera was used previously by you or others, please cite the reference for the reader. If it was a gift, please acknowledge this.

We thank the reviewer for the point. As suggested, we modified colors of chimeric α1C-1B loop in schematic diagram (new Figure 6A). We added a description of the chimeric α1C-1B construct and cited Agler’s paper in the Results section (the first paragraph on page 11). We also acknowledge the chimeric construct in the Methods section (page 17).

– Figure 6-S2. Lines 242-244. This sentence should be removed. Not enough control experiments were done to make this conclusion. The authors haven't convincingly shown that they have the two pathways isolated from one another. One clear way to do this is simply to raise the [BAPTA] from 0.1 to 10 mM.

We thank the reviewer for constructive comment. According to the reviewer’s suggestion, we deleted the sentence (end of the first paragraph on page 12).

– The data in Figure 7 just scratch the surface of PIP2 interaction with VSDII. In addition to the discussion about charge changes in the voltage-sensor vs PIP2 binding to gating charge residues is necessary for clarity.

We thank for the instructive comment. Based the comments, we edited the related sentences (middle of the second paragraph on page 12).

Additional overstated results in the discussion need backing off since other explanations may account for the results:– Page 14 line 381. Please change "demonstrated" to predict.

As suggested, we changed the word (the second paragraph on page 13).

– Line 421, change "…We found that…" to "…Our data are consistent with…"

As suggested, we have revised the sentence (the second paragraph on page 14).

– Line 425, "…Our data demonstrate…" to Our data indicate…"

As suggested, we replaced ‘demonstrate’ with ‘indicate’ (the second paragraph on page 14).